# Stochastic Gradient Methods with Compressed Communication for Decentralized Saddle Point Problems

**Chhavi Sharma, Vishnu Narayanan, P. Balamurugan**
IEOR, IIT Bombay, Mumbai, India
{chhavisharma, vishnu, balamurugan.palaniappan}@iitb.ac.in

## Abstract

We develop two compression based stochastic gradient algorithms to solve a class of non-smooth strongly convex-strongly concave saddle-point problems in a decentralized setting (without a central server). Our first algorithm is a Restart-based Decentralized Proximal Stochastic Gradient method with Compression (C-RDPSG) for general stochastic settings. We provide rigorous theoretical guarantees of C-RDPSG with gradient computation complexity and communication complexity of order $\mathcal{O}((1+\delta)^4 \frac{1}{L^2}\kappa_f^2\kappa_g^2\frac{1}{\epsilon})$, to achieve an $\epsilon$-accurate saddle-point solution, where $\delta$ denotes the compression factor, $\kappa_f$ and $\kappa_g$ denote respectively the condition numbers of objective function and communication graph, and $L$ denotes the smoothness parameter of the smooth part of the objective function. Next, we present a Decentralized Proximal Stochastic Variance Reduced Gradient algorithm with Compression (C-DPSVRG) for finite sum setting which exhibits gradient computation complexity and communication complexity of order $\mathcal{O}\left((1+\delta)\max\{\kappa_f^2, \sqrt{\delta}\kappa_f^2\kappa_g, \kappa_g\}\log\left(\frac{1}{\epsilon}\right)\right)$. Extensive numerical experiments show competitive performance of the proposed algorithms and provide support to the theoretical results obtained.

## 1 Introduction

We focus on solving the following saddle point (or mini-max) problem in a fully decentralized setting without a central server:

$$\min_{x\in\mathcal{X}}\max_{y\in\mathcal{Y}}\Psi(x,y) := \frac{1}{m}\sum_{i=1}^{m}(f_i(x,y)+g(x)-r(y)), \quad \text{(SPP)}$$

where $\mathcal{X}\subset\mathbb{R}^{d_x}$, $\mathcal{Y}\subset\mathbb{R}^{d_y}$ are convex and compact sets, $f_i:\mathcal{X}\times\mathcal{Y}\mapsto\mathbb{R}$ for every node $i\in[m]$ is smooth, strongly-convex in $x$ and strongly-concave in $y$, and $g:\mathcal{X}\mapsto\mathbb{R}$ and $r:\mathcal{Y}\mapsto\mathbb{R}$ are proper, continuous, convex functions which might be non-smooth. This class of saddle point problems finds its use in robust classification and regression applications and AUC maximization problems [30, 40]. We assume that the static topology of the decentralized environment is represented using an undirected, connected, simple graph $\mathcal{G} = (\mathcal{V},\mathcal{E})$, where $\mathcal{V} = \{1,2,\dots,m\} =: [m]$ denotes the set of $m$ computing nodes (with similar computing capabilities and memory) and an edge $e_{ij}\in\mathcal{E}$ denotes that nodes $i,j\in\mathcal{V}$ are connected. Also, we assume that the communication is synchronous and at every synchronization step, node $i$ communicates only with its neighbors $\mathcal{N}(i) = \{j\in\mathcal{V}: e_{ij}\in\mathcal{E}\}$. In this work, we design algorithms which can achieve sublinear/linear convergence rates to solve problem (SPP), by using stochastic oracles for efficient gradient computation, and compressed information exchange for efficient communication. In particular, we focus on the following two settings of problem (SPP): (i) general stochastic setting (ii) finite sum setting. In general stochastic setting, we assume the following form for the local function: $f_i(x,y) = E_{\xi^i\sim\mathcal{D}_i}\left[f_i(x,y;\xi^i)\right]$, where

Workshop on Federated Learning: Recent Advances and New Challenges, in Conjunction with NeurIPS 2022 (FL-NeurIPS'22). This workshop does not have official proceedings and this paper is non-archival.

$\mathcal{D}_i$ is the local sample distribution in node $i$, allowing for general heterogeneous data distributions across different nodes. In finite sum setting, we assume that each $f_i(x, y)$ is represented as the average over local batches, hence $f_i(x, y) = \frac{1}{n} \sum_{j=1}^{n} f_{ij}(x, y)$, where $f_{ij}(x, y)$ represents the loss function at $j$th batch of samples at node $i$.

**Our Contributions**: Inspired by algorithms developed for decentralized minimization problems [24, 20], we design two algorithms for the general stochastic setting and finite sum setting. For general stochastic setting, we propose Restart-based Decentralized Proximal Stochastic Gradient method with Compression (C-RDPSG) which achieves gradient computation complexity (measured in terms of number of calls to stochastic gradient oracles (SGO)) and communication complexity (measured in terms of number of communication rounds) of order $\mathcal{O}((1 + \delta)^4 \frac{1}{L^2} \kappa_f^2 \kappa_g^2 \frac{1}{\epsilon})$, to obtain an $\epsilon$-accurate saddle-point solution, where $\delta$ denotes the compression factor, $\kappa_f$ and $\kappa_g$ denote respectively the condition numbers of objective function $f$ and communication graph $\mathcal{G}$, and $L$ denotes the smoothness parameter of smooth part $\frac{1}{m} \sum_i f_i$. In finite sum setting common in machine learning, we propose Decentralized Proximal Stochastic Variance Reduced Gradient algorithm with Compression (C-DPSVRG) which is shown to have gradient computation complexity and communication complexity of order $\mathcal{O}((1+\delta) \max\{\kappa_f^2, \sqrt{\delta}\kappa_f^2\kappa_g, \kappa_g\} \log(\frac{1}{\epsilon}))$. Using extensive experiments, we show empirical evidence supporting the theoretical guarantees obtained in our analysis.

**Notations:** The following notations will be used in the paper. The $k$-tuple $(u^1, u^2, \ldots, u^k)$ of $k$ vectors $u^1, u^2, \ldots, u^k$, each of size $d \times 1$ denotes the vector $\left[(u^1)^\top (u^2)^\top \ldots (u^k)^\top\right]^\top$ of size $kd \times 1$. The notation $z = (x, y) \in \mathbb{R}^{d_x + d_y}$ denotes the pair of primal variable $x$ and dual variable $y$, and $z^\star = (x^\star, y^\star)$ denotes a saddle point of problem (SPP). The communication link between a pair of nodes $(i, j) \in \mathcal{V} \times \mathcal{V}$ is assumed to have associated weight $W_{ij} \in [0, 1]$. Weights $W_{ij}$ are collected into a weight matrix $W$ of size $m \times m$. $I_p$ denotes a $p \times p$ identity matrix, $\mathbf{1}$ denotes a $m \times 1$ column vector of ones and $J = \frac{1}{m}\mathbf{1}\mathbf{1}^\top$ is a $m \times m$ matrix of uniform weights equal to $\frac{1}{m}$. We define $f(z) := f(x, y) := \sum_{i=1}^{m} f_i(x, y)$, $L = \max\{L_{xx}, L_{yy}, L_{xy}, L_{yx}\}$, $\mu = \min\{\mu_x, \mu_y\}$ where $L_{xx}, L_{yy}, L_{xy}, L_{yx}$ are the smoothness parameters of $f_i(x, y)$ (see Appendices E, G) and $\mu_x, \mu_y$ are the strong convexity, strong concavity parameters of $f_i(x, y)$ (see Assumptions 3.1-3.2). The condition number $\kappa_f$ of $f$ is defined as $L/\mu$. The condition number $\kappa_g$ of communication graph $\mathcal{G}$ is defined as the ratio of largest eigenvalue and second smallest eigenvalue of $I - W$. The notation $\langle u, v \rangle$ denotes the inner product between two $d \times 1$ vectors $u$ and $v$. For a $d \times 1$ vector $u$ and for some $d \times d$ symmetric positive semi-definite (p.s.d) matrix $A$, we define $\|u\|_A^2 = u^\top A u$. $\|u\|^2$ denotes $\ell_2$ norm of vector $u$. $A \otimes B$ denotes the Kronecker product of two matrices $A$ and $B$.

**Paper Organization:** We illustrate an inexact primal-dual hybrid algorithm for solving (SPP) in Section 2, followed by a discussion of assumptions used for problem setup (Section 3). Details about C-RDPSG for general stochastic setting and C-DPSVRG for finite sum setting are given respectively in Sections 4 and 5. Due to space constraints, related work is given in Appendix A and experiment details are in Appendix I.

## 2   An Inexact Primal-Dual Hybrid Algorithm to Solve Problem (SPP)

Assuming the local copy of $(x, y)$ in $i$-th node as $(x^i, y^i)$, we collect local primal and dual variables into $\mathbf{x} = (x^1, x^2, \ldots x^m) \in \mathbb{R}^{md_x}$ and $\mathbf{y} = (y^1, y^2, \ldots y^m) \in \mathbb{R}^{md_y}$. Using this notation, the problem (SPP) can be formulated as an optimization problem with consensus constraints on $\mathbf{x}$ and $\mathbf{y}$:

$$\min_{\mathbf{x} \in \mathbb{R}^{md_x}} \max_{\mathbf{y} \in \mathbb{R}^{md_y}} F(\mathbf{x}, \mathbf{y}) + G(\mathbf{x}) - R(\mathbf{y}) \text{ s.t. } (U \otimes I_{d_x})\mathbf{x} = 0, \ (U \otimes I_{d_y})\mathbf{y} = 0, \quad (1)$$

where $F(\mathbf{x}, \mathbf{y}) = \sum_{i=1}^{m} f_i(x^i, y^i)$, $G(\mathbf{x}) = \sum_{i=1}^{m} (g(x^i) + \delta_{\mathcal{X}}(x^i))$, $R(\mathbf{y}) = \sum_{i=1}^{m} (r(y^i) + \delta_{\mathcal{Y}}(y^i))$ and $U = \sqrt{I_m - W}$. Formulating a decentralized optimization problem as an equivalent problem with a consensus constraint on the local variables is well-known [26]. The assumptions on $W$ (to be made later) would imply $I_m - W$ to be symmetric p.s.d. and hence the existence of $\sqrt{I_m - W}$. $\delta_C(u)$ denotes the indicator function of set $C$ which is 0 when $u \in C$ and $+\infty$ otherwise.

We have the following Lagrangian function of problem (1):

$$\mathcal{L}(\mathbf{x}, \mathbf{y}; S^{\mathbf{x}}, S^{\mathbf{y}}) = F(\mathbf{x}, \mathbf{y}) + G(\mathbf{x}) - R(\mathbf{y}) + \langle S^{\mathbf{x}}, (U \otimes I_{d_x})\mathbf{x} \rangle + \langle S^{\mathbf{y}}, (U \otimes I_{d_y})\mathbf{y} \rangle, \quad (2)$$

where $S^{\mathbf{x}} \in \mathbb{R}^{md_x}$ and $S^{\mathbf{y}} \in \mathbb{R}^{md_y}$ denote the Lagrange multipliers associated with consensus constraints on variables $\mathbf{x}$ and $\mathbf{y}$ respectively. We prove that solving constrained problem (1) is equivalent to solving the following problem (see Theorem C.1 in Appendix C):

$$\min_{\mathbf{x} \in \mathbb{R}^{md_x}, S^{\mathbf{y}} \in \mathbb{R}^{md_y}} \max_{\mathbf{y} \in \mathbb{R}^{md_y}, S^{\mathbf{x}} \in \mathbb{R}^{md_x}} \mathcal{L}(\mathbf{x}, \mathbf{y}; S^{\mathbf{x}}, S^{\mathbf{y}}). \quad (3)$$

To solve problem (3), we propose inexact Primal Dual Hybrid Gradient (PDHG) parallel updates for the primal-dual variable pair $\mathbf{x}, S^{\mathbf{x}}$ and dual-primal pair $\mathbf{y}, S^{\mathbf{y}}$, illustrated in eq. (P1) and (D1).

Note that in eq. (P1), $\nu_{t+1}^{\mathbf{x}}$ is found using a prox-linear step involving linearization of $F(\mathbf{x}, \mathbf{y})$ with respect to $\mathbf{x}$ and a penalized cost-to-move term $\frac{1}{2s}\|\mathbf{x} - \mathbf{x}_t\|^2$, followed by an ascent step to update the Lagrange dual variable $S^{\mathbf{x}}$. Then $\hat{\mathbf{x}}_{t+1}$ is found using prox-linear step similar to the first step but using the recent $S_{t+1}^{\mathbf{x}}$. Finally $\mathbf{x}_{t+1}$ is found by a prox step where $\operatorname{prox}_{sG}(x) = \arg\min_{u \in \mathbb{R}^{md_x}} G(u) + \frac{1}{2s}\|u - x\|^2$. Letting $D_t^{\mathbf{x}} = (U \otimes I_{d_x})S_t^{\mathbf{x}}$, and pre-multiplying by $U \otimes I_{d_x}$ in the update step of $S^{\mathbf{x}}$, equations (P1) reduce to those in equations (P2). Similarly, the updates to $\mathbf{y}, S^{\mathbf{y}}$ can be done using appropriate gradient ascent-descent steps which lead to corresponding equations (D1) and (D2).

| Updates to primal-dual pair $\mathbf{x}, S^{\mathbf{x}}$: | | Updates to dual-primal pair $\mathbf{y}, S^{\mathbf{y}}$: | |
|---|---|---|---|
| $\nu_{t+1}^{\mathbf{x}} = \underset{\mathbf{x} \in \mathbb{R}^{md_x}}{\arg\min} \left\{ \begin{array}{l} \langle \mathbf{x}, \nabla_{\mathbf{x}} F(\mathbf{x}_t, \mathbf{y}_t) \rangle \\ + \langle \mathbf{x}, (U \otimes I_{d_x})S_t^{\mathbf{x}} \rangle \\ + \frac{1}{2s}\|\mathbf{x} - \mathbf{x}_t\|^2 \end{array} \right.$ | | $\nu_{t+1}^{\mathbf{y}} = \underset{\mathbf{y} \in \mathbb{R}^{md_y}}{\arg\max} \left\{ \begin{array}{l} \langle \mathbf{y}, \nabla_{\mathbf{y}} F(\mathbf{x}_t, \mathbf{y}_t) \rangle \\ + \langle \mathbf{y}, (U \otimes I_{d_y})S_t^{\mathbf{y}} \rangle \\ - \frac{1}{2s}\|\mathbf{y} - \mathbf{y}_t\|^2 \end{array} \right.$ | |
| $S_{t+1}^{\mathbf{x}} = S_t^{\mathbf{x}} + \frac{\gamma}{2s}(U \otimes I_{d_x})\nu_{t+1}^{\mathbf{x}}$ | (P1) | $S_{t+1}^{\mathbf{y}} = S_t^{\mathbf{y}} - \frac{\gamma}{2s}(U \otimes I_{d_y})\nu_{t+1}^{\mathbf{y}}$ | (D1) |
| $\hat{\mathbf{x}}_{t+1} = \underset{\mathbf{x} \in \mathbb{R}^{md_x}}{\arg\min} \left\{ \begin{array}{l} \langle \mathbf{x}, \nabla_{\mathbf{x}} F(\mathbf{x}_t, \mathbf{y}_t) \rangle \\ + \langle \mathbf{x}, (U \otimes I_{d_x})S_{t+1}^{\mathbf{x}} \rangle \\ + \frac{1}{2s}\|\mathbf{x} - \mathbf{x}_t\|^2 \end{array} \right.$ | | $\hat{\mathbf{y}}_{t+1} = \underset{\mathbf{y} \in \mathbb{R}^{md_y}}{\arg\max} \left\{ \begin{array}{l} \langle \mathbf{y}, \nabla_{\mathbf{y}} F(\mathbf{x}_t, \mathbf{y}_t) \rangle \\ + \langle \mathbf{y}, (U \otimes I_{d_y})S_{t+1}^{\mathbf{y}} \rangle \\ - \frac{1}{2s}\|\mathbf{y} - \mathbf{y}_t\|^2 \end{array} \right.$ | |
| $\mathbf{x}_{t+1} = \underset{sG}{\operatorname{prox}}(\hat{\mathbf{x}}_{t+1}).$ | | $\mathbf{y}_{t+1} = \underset{sR}{\operatorname{prox}}(\hat{\mathbf{y}}_{t+1}).$ | |
| $\Downarrow$ | | $\Downarrow$ | |
| $\nu_{t+1}^{\mathbf{x}} = \mathbf{x}_t - s\nabla_{\mathbf{x}} F(\mathbf{x}_t, \mathbf{y}_t) - sD_t^{\mathbf{x}}$ | | $\nu_{t+1}^{\mathbf{y}} = \mathbf{y}_t + s\nabla_{\mathbf{y}} F(\mathbf{x}_t, \mathbf{y}_t) - sD_t^{\mathbf{y}}$ | |
| $D_{t+1}^{\mathbf{x}} = D_t^{\mathbf{x}} + \frac{\gamma}{2s}((I_m - W) \otimes I_{d_x})\nu_{t+1}^{\mathbf{x}}$ | | $D_{t+1}^{\mathbf{y}} = D_t^{\mathbf{y}} + \frac{\gamma}{2s}((I_m - W) \otimes I_{d_y})\nu_{t+1}^{\mathbf{y}}$ | |
| $\hat{\mathbf{x}}_{t+1} = \mathbf{x}_t - s\nabla_{\mathbf{x}} F(\mathbf{x}_t, \mathbf{y}_t) - sD_{t+1}^{\mathbf{x}}$ | (P2) | $\hat{\mathbf{y}}_{t+1} = \mathbf{y}_t + s\nabla_{\mathbf{y}} F(\mathbf{x}_t, \mathbf{y}_t) - sD_{t+1}^{\mathbf{y}}$ | (D2) |
| $\qquad = \nu_{t+1}^{\mathbf{x}} - \frac{\gamma}{2}((I_m - W) \otimes I_{d_x})\nu_{t+1}^{\mathbf{x}}$ | | $\qquad = \nu_{t+1}^{\mathbf{y}} - \frac{\gamma}{2}((I_m - W) \otimes I_{d_y})\nu_{t+1}^{\mathbf{y}}$ | |
| $\mathbf{x}_{t+1} = \underset{sG}{\operatorname{prox}}(\hat{\mathbf{x}}_{t+1}).$ | | $\mathbf{y}_{t+1} = \underset{sR}{\operatorname{prox}}(\hat{\mathbf{y}}_{t+1}).$ | |

Inexact PDHG type updates are known for solving a Lagrangian function of an underlying convex minimization problem in single machine setting (e.g. proximal alternating predictor-corrector (PAPC) algorithm [7, 25], primal-dual fixed point (PDFP) algorithm [8]), and in decentralized setting [20]. In contrast to PAPC, PDFP, and the algorithm in [20], the type of inexact PDHG updates proposed in our work addresses a Lagrangian function corresponding to an underlying saddle point problem (SPP). To our knowledge, our work is the first to extend inexact PDHG type updates to solve Lagrangian of saddle point problem of form (SPP). For other perspectives and generalizations of inexact PDHG, see [6, 12, 9, 35, 38, 20].

Observe that the term $((I_m - W) \otimes I_{d_x})\nu_{t+1}^{\mathbf{x}}$ in eq. (P2) and $((I_m - W) \otimes I_{d_y})\nu_{t+1}^{\mathbf{y}}$ in eq. (D2) denote the communication of $\nu_{t+1}^{\mathbf{x}}$ and $\nu_{t+1}^{\mathbf{y}}$ across the nodes. Further note that $\nu_{t+1}^{\mathbf{x}}$ and $\nu_{t+1}^{\mathbf{y}}$ need to be communicated only once for updating $D_{t+1}^{\mathbf{x}}, \hat{\mathbf{x}}_{t+1}$ and $D_{t+1}^{\mathbf{y}}, \hat{\mathbf{y}}_{t+1}$ in the inexact PDHG updates for $\mathbf{x}$ and $\mathbf{y}$ respectively. This results in cheaper communication in every iteration when compared to the multiple communications that happen in a single iteration of the algorithms in [5, 36, 23]. To improve the communication efficiency further, we propose to compress $\nu_{t+1}^{\mathbf{x}}$ and $\nu_{t+1}^{\mathbf{y}}$. We recall that compression has not yet been used in existing algorithms to solve (SPP) in decentralized setting without a central server. Compression based algorithms to solve smooth variational inequalities (related to problem (SPP)) are available only for decentralized settings **with** a central server [3] .

We follow [27, 24] to compress a related difference vector instead of directly compressing $\nu_{t+1}^{\mathbf{x}}$ and $\nu_{t+1}^{\mathbf{y}}$. Each node $i$ is assumed to maintain a local vector $H^{i,\mathbf{x}}$ and a stochastic compression operator $Q$ is applied on the difference vector $\nu_{t+1}^{i,\mathbf{x}} - H_t^{i,\mathbf{x}}$. The concise form of these updates is illustrated in Algorithm 4 (COMM procedure) in Appendix B. Algorithm 1 illustrates the proposed inexact PDHG updates with compression. Thus the inexact PDHG update steps to obtain $\nu_{t+1}^{\mathbf{x}}$ and $\nu_{t+1}^{\mathbf{y}}$ involve computing the gradients $\nabla_{\mathbf{x}} F(\mathbf{x}_t, \mathbf{y}_t) = \left( \nabla_x f_1(x^1, y^1), \ldots, \nabla_x f_m(x^m, y^m) \right)$ and $\nabla_{\mathbf{y}} F(\mathbf{x}_t, \mathbf{y}_t) = \left( \nabla_y f_1(x^1, y^1), \ldots, \nabla_y f_m(x^m, y^m) \right)$ using a gradient computation oracle $\mathcal{G}$. We now discuss methods based on two different stochastic gradient oracles to compute the gradients. Before discussing the methods, we state technical assumptions common to both the methods.

**Algorithm 1** Inexact primal dual hybrid algorithm updates using gradient computation oracle $\mathcal{G}$ (IPDHG)

1: **INPUT:** $\mathsf{x}, \mathsf{y}, \mathsf{D^x}, \mathsf{D^y}, \mathsf{H^x}, \mathsf{H^y}, \mathsf{H^{w,x}}, \mathsf{H^{w,y}}, s, \gamma_\mathsf{x}, \gamma_\mathsf{y}, \alpha_\mathsf{x}, \alpha_\mathsf{y}, \mathcal{G}$
2: Compute gradients $\mathcal{G}^\mathsf{x}$ and $\mathcal{G}^\mathsf{y}$ at $(\mathsf{x}, \mathsf{y})$ via oracle $\mathcal{G}$
3: $\nu^\mathsf{x} = \mathsf{x} - s\mathcal{G}^\mathsf{x} - s\mathsf{D^x}$
4: $\hat{\nu}^\mathsf{x}, \hat{\nu}^\mathsf{w,x}, \mathsf{H}^\mathsf{x}_{new}, \mathsf{H}^\mathsf{w,x}_{new} = \text{COMM}\left(\nu^\mathsf{x}, \mathsf{H^x}, \mathsf{H^{w,x}}, \alpha_\mathsf{x}\right)$
5: $\mathsf{D}^\mathsf{x}_{new} = \mathsf{D^x} + \frac{\gamma_\mathsf{x}}{2s}(\hat{\nu}^\mathsf{x} - \hat{\nu}^\mathsf{w,x})$
6: $\hat{\mathsf{x}} = \nu^\mathsf{x} - \frac{\gamma_\mathsf{x}}{2}(\hat{\nu}^\mathsf{x} - \hat{\nu}^\mathsf{w,x})$
7: $\mathsf{x}_{new} = \text{prox}_{sG}(\hat{\mathsf{x}})$
8: $\nu^\mathsf{y} = \mathsf{y} + s\mathcal{G}^\mathsf{y} - s\mathsf{D^y}$
9: $\hat{\nu}^\mathsf{y}, \hat{\nu}^\mathsf{w,y}, \mathsf{H}^\mathsf{y}_{new}, \mathsf{H}^\mathsf{w,y}_{new} = \text{COMM}\left(\nu^\mathsf{y}, \mathsf{H^y}, \mathsf{H^{w,y}}, \alpha_\mathsf{y}\right)$
10: $\mathsf{D}^\mathsf{y}_{new} = \mathsf{D}^y + \frac{\gamma_\mathsf{y}}{2s}(\hat{\nu}^\mathsf{y} - \hat{\nu}^\mathsf{w,y})$
11: $\hat{\mathsf{y}} = \nu^\mathsf{y} - \frac{\gamma_\mathsf{y}}{2}(\hat{\nu}^\mathsf{y} - \hat{\nu}^\mathsf{w,y})$
12: $\mathsf{y}_{new} = \text{prox}_{sR}(\hat{\mathsf{y}})$
13: **RETURN:** $\mathsf{x}_{new}, \mathsf{y}_{new}, \mathsf{D}^\mathsf{x}_{new}, \mathsf{D}^\mathsf{y}_{new}, \mathsf{H}^\mathsf{x}_{new}, \mathsf{H}^\mathsf{y}_{new}, \mathsf{H}^\mathsf{w,x}_{new}, \mathsf{H}^\mathsf{w,y}_{new}$.

## 3 Assumptions

We list below the assumptions to be used throughout this work.

**Assumption 3.1.** *Each $f_i(\cdot, y)$ is $\mu_x$-strongly convex for every $y \in \mathcal{Y}$; hence for any $x_1, x_2 \in \mathcal{X}$ and fixed $y \in \mathcal{Y}$, it holds: $f_i(x_1, y) \geq f_i(x_2, y) + \langle \nabla_x f_i(x_2, y), x_1 - x_2 \rangle + \frac{\mu_x}{2} \|x_1 - x_2\|^2$.*

**Assumption 3.2.** *Each $f_i(x, \cdot)$ is $\mu_y$-strongly concave for every $x \in \mathcal{X}$; hence for any $y_1, y_2 \in \mathcal{Y}$ and fixed $x \in \mathcal{X}$, it holds: $f_i(x, y_1) \leq f_i(x, y_2) + \langle \nabla_y f_i(x, y_2), y_1 - y_2 \rangle - \frac{\mu_y}{2} \|y_1 - y_2\|^2$.*

**Assumption 3.3.** *$g(x)$ and $r(y)$ are proper, convex, continuous and possibly non-smooth functions.*

**Assumption 3.4.** *The compression operator $Q$ satisfies the following for every $u \in \mathbb{R}^d$: (i) $Q(u)$ is an unbiased estimate of $u$: $E[Q(u)] = u$ (ii) $E[\|Q(u) - u\|^2] \leq \delta \|u\|^2$, where the constant $\delta \geq 0$ denotes the amount of compression induced by operator $Q$ and is called a compression factor. When $\delta = 0$, $Q$ achieves no compression.*

**Assumption 3.5.** *The weight matrix $W$ satisfies the following conditions: $W$ is symmetric and row stochastic, $W_{ij} > 0$ if and only if $(i, j) \in \mathcal{E}$ and $W_{ii} > 0$ for all $i \in [m]$. The eigenvalues of $W$ denoted by $\lambda_1, \ldots, \lambda_m$ satisfy: $-1 < \lambda_m \leq \lambda_{m-1} \leq \ldots \leq \lambda_2 < \lambda_1 = 1$.*

Note that Assumptions 3.1-3.3 and Assumption 3.5 are standard in the study of saddle point problems (e.g. [4, 5, 29, 23]). We also note that Assumption 3.4 is standard in existing works (e.g. [1, 20, 24]). For example, $b$-bits quantization operator (see Appendix I ) satisfies Assumption 3.4.

## 4 General Stochastic Setting

In the general stochastic setting, we allow for availability of heterogeneous data distributions in each node and assume that the gradients $\nabla_x f_i(x, y)$ and $\nabla_y f_i(x, y)$ are computed using an oracle $\mathcal{G}$ described below.

> **General Stochastic Gradient Oracle (GSGO):**
> **(1).** Sample a mini-batch of samples $\xi^i \sim D_i$ in each node $i$, where $D_i$ is the data distribution local to node $i$.
> **(2).** Compute stochastic gradients: $\mathcal{G}^{i,x} = \nabla_x f_i(x^i, y^i; \xi^i)$ and $\mathcal{G}^{i,y} = \nabla_y f_i(x^i, y^i; \xi^i)$.

Inspired by restart based schemes in single machine setting [39, 40], we design in this work, a restart based stochastic gradient method illustrated in Algorithm 2, which invokes in every iteration $k$, a sequence of $t_k$ primal and dual variable updates using inexact PDHG with compression (Algorithm 1). However, the restart scheme proposed in Algorithm 2 is simpler than that in [40], where the restart scheme in single machine setting requires the computation of Fenchel conjugate at every restart incurring additional $\mathcal{O}(d_y)$ operations. Our scheme is also different in design compared to other restart based schemes studied for saddle point problems in single machine setting [42, 13, 22]. In Algorithm 2, step length $s_k$ is chosen to decrease geometrically only at every restart step $k$, resulting in better convergence. The details of derivations of parameter $\rho$, step lengths $\gamma_k^x, \gamma_k^y$ and parameters $\alpha_{x,k}, \alpha_{y,k}$ used in COMM procedure are provided in Appendix E.4.

Under appropriate assumptions on unbiasedness and smoothness of the stochastic gradients (see Appendix E), we have the following convergence result of Algorithm 2.

**Theorem 4.1.** *Suppose* $\{x_{k,0}\}_k$ *and* $\{y_{k,0}\}_k$ *are the sequences generated by Algorithm 2. Then with at most*

$$K(\epsilon) = \max\left\{\mathcal{O}\left(\log_2\left(\frac{\|z_0 - \mathbf{1}z^\star\|^2}{\epsilon} + \frac{\sqrt{\delta}}{mL^2\kappa_f^2\epsilon}\right)\right), \mathcal{O}\left(\log_2\left(\frac{(1+\delta)^2 + m\sigma^2\kappa_g(1+\delta)^2}{L^2\epsilon}\right)\right)\right\},$$

*iterations,* $E[\|x_{K(\epsilon),0} - \mathbf{1}x^\star\|^2 + \|y_{K(\epsilon),0} - \mathbf{1}y^\star\|^2] \leq \epsilon$, *where* $\sigma^2$ *is the local variance bound in stochastic gradients (see Assumption E.1 in Appendix E). Moreover the total gradient computation complexity and communication complexity to achieve* $\epsilon$-*accurate saddle point solution in expectation are*

$$T_{grad}(\epsilon) = \mathcal{O}\left(\max\left\{\frac{(1+\delta)^2\|z_0 - \mathbf{1}z^\star\|\kappa_f^2\kappa_g}{\sqrt{\epsilon}} + \frac{(1+\delta)^2\delta^{1/4}\kappa_f\kappa_g}{\sqrt{m}L\sqrt{\epsilon}}, \frac{(1+\delta)^4(\kappa_f^2\kappa_g + m\sigma^2\kappa_f^2\kappa_g^2)}{L^2\epsilon}\right\}\right)$$

*and* $T_{comm}(\epsilon) = T_{grad}(\epsilon) + K(\epsilon)$ *respectively.*

---

**Algorithm 2** **R**estart-based **D**ecentralized **P**roximal **S**tochastic **G**radient method with **C**ompression (C-RDPSG)

---
1: **INPUT:** $x_{0,0}^i = x_0, y_{0,0}^i = y_0, s_0 = \frac{1}{4L\kappa_f}$, $\mathcal{G}$ obtained using GSGO, number of iterations $K$, $\rho$ depending on $\kappa_f, \kappa_g$ and $\delta$ .
2: **for** $k = 0$ to $K - 1$ **do**
3: $\quad s_k = \frac{s_0}{2^{k/2}}, b_{x,k} = \mu_x s_k - 4s_k^2 L_{yx}^2, b_{y,k} = \mu_y s_k - 4s_k^2 L_{xy}^2$
4: $\quad \gamma_k^x = \frac{b_{x,k}}{2(1+\delta)^2\lambda_{\max}(I_m - W)}, \gamma_k^y = \frac{b_{y,k}}{2(1+\delta)^2\lambda_{\max}(I_m - W)}$
5: $\quad \alpha_{x,k} = \frac{b_{x,k}}{1+\delta}, \alpha_{y,k} = \frac{b_{y,k}}{1+\delta}$
6: $\quad M_{x,k} = 1 - \frac{\sqrt{\delta}\alpha_{x,k}}{1 - \frac{\gamma_k^x}{2}\lambda_{\max}(I - W)}, M_{y,k} = 1 - \frac{\sqrt{\delta}\alpha_{y,k}}{1 - \frac{\gamma_k^y}{2}\lambda_{\max}(I - W)}, M_k = \min\{M_{x,k}, M_{y,k}\}$
7: $\quad$ Set $t_k = \frac{1}{-\log\left(1 - \frac{\rho}{2^{k/2}}\right)} \max\{\log\left(\frac{3M_{x,k} + 6\sqrt{\delta}}{M_k}\right), \log\left(\frac{3M_{y,k} + 6\sqrt{\delta}}{M_k}\right), \log\left(\frac{3}{M_k}\right)\}$
8: $\quad D_{k,0}^x = D_{k,0}^y = 0, H_{k,0}^x = x_{k,0}, H_{k,0}^y = y_{k,0}, H_{k,0}^{w,x} = (W \otimes I_{d_x})H_{k,0}^x, H_{k,0}^{w,y} = (W \otimes I_{d_y})H_{k,0}^y$
9: $\quad$ **for** $t = 0$ to $t_k - 1$ **do**
10: $\quad\quad x_{k,t+1}, y_{k,t+1}, D_{k,t+1}^x, D_{k,t+1}^y, H_{k,t+1}^x, H_{k,t+1}^y, H_{k,t+1}^{w,x}, H_{k,t+1}^{w,y}$
$\quad\quad\quad = \text{IPDHG}(x_{k,t}, y_{k,t}, D_{k,t}^x, D_{k,t}^y, H_{k,t}^x, H_{k,t}^y, H_{k,t}^{w,x}, H_{k,t}^{w,y}, s_k, \gamma_k^x, \gamma_k^y, \alpha_{x,k}, \alpha_{y,k}, \mathcal{G})$
11: $\quad$ **end for**
12: $\quad x_{k+1,0} = x_{k,t_k}, y_{k+1,0} = y_{k,t_k}$
13: **end for**
14: **RETURN:** $x_{K,0}, y_{K,0}$.

---

Due to space considerations, we discuss proof details in Appendix F. We note that the number of outer iterates $K(\epsilon)$ in Theorem 4.1 depend logarithmically on $\kappa_g, \delta$ and total variance bound $m\sigma^2$. Larger values of these parameters contribute to the accumulation of consensus, compression and gradient approximation errors. Hence more restarts might be required to reduce these errors accumulated during IPDHG updates with GSGO (see Figures 6, 4, 8 and 10 in Appendix I for empirical evidence of this fact). The computation complexity in Theorem 4.1 depends on compression factor and graph condition number as $\mathcal{O}(\delta^4)$ and $\mathcal{O}(\kappa_g^2)$. The dependence on graph condition number reduces to $\mathcal{O}(\kappa_g)$ when $\sigma = 0$. Therefore, in the deterministic gradients regime, C-RDPSG is less sensitive to change in network topology compared to stochastic gradients regime. We note that [17] develops non-compression based optimal algorithms with complexity $\mathcal{O}(\kappa_f\sqrt{\kappa_g}\log(1/\epsilon))$ for solving decentralized finite-sum variational inequalities. However, Theorem 4.1 explains the complexity results of C-RDPSG in the general stochastic setting.

## 5 Finite Sum Setting

In the finite sum setting, we assume that each local function $f_i(x, y)$ is of the form $\frac{1}{n}\sum_{j=1}^n f_{ij}(x, y)$. For simplicity, we assume that each node $i$ has same number of batches $n$. However, our analysis easily extends to different number of batches $n_i$. Let $N$ denote the total number of samples. Let $B$ and $N_\ell$ respectively denote the batch size and number of local samples at each node $i$. The number of samples in the function component $f_{ij}$ is determined by the batch size $B = N_\ell/n$. Let $\mathcal{P}_i = \{p_{il} : l \in \{1, 2, \ldots, n\}\}$ denote a probability distribution where $p_{il}$ is the probability with

which batch $l$ is sampled at node $i$. Let $p_{\min} := \min_{i,l} p_{il}$. Without loss of generality we assume that $p_{\min} > 0$, hence each batch is chosen with a positive probability. Note that GSGO in Algorithm 2 shows sublinear convergence for solving (SPP). Stochastic variance reduction techniques [14, 18] are known to accelerate convergence of GSGO based methods for decentralized convex minimization problems [20, 37]. Inspired by this success, we propose a Stochastic Variance Reduced Gradient (SVRG) oracle comprising the following steps.

---

**Stochastic Variance Reduced Gradient Oracle (SVRGO):**
**(1). Index sampling:** Sample $l \in \{1, 2, \ldots, n\} \sim \mathcal{P}_i$ for every node $i$.
**(2). Stochastic gradient computation with variance reduction:** For a reference point $\tilde{z}^i = (\tilde{x}^i, \tilde{y}^i)$, compute stochastic gradients at $z^i = (x^i, y^i)$ with respect to $x$ and $y$ as follows:

$$\mathcal{G}^{i,x} = \frac{1}{np_{il}} \left( \nabla_x f_{il}(z^i) - \nabla_x f_{il}(\tilde{z}^i) \right) + \nabla_x f_i(\tilde{z}^i), \tag{4}$$

$$\mathcal{G}^{i,y} = \frac{1}{np_{il}} \left( \nabla_y f_{il}(z^i) - \nabla_y f_{il}(\tilde{z}^i) \right) + \nabla_y f_i(\tilde{z}^i). \tag{5}$$

**(3). Reference point update:** Sample $\omega \in \{0, 1\} \sim \text{Bernoulli}(p)$ and update the reference points as follows:

$$\tilde{x}^i \longleftarrow \omega x^i + (1 - \omega)\tilde{x}^i \tag{6}$$

$$\tilde{y}^i \longleftarrow \omega y^i + (1 - \omega)\tilde{y}^i. \tag{7}$$

---

The SVRGO setup above requires computation of full-batch gradient at the reference point periodically (equations (4)-(5)), but is memory-efficient compared to other variance reduction schemes (e.g. SAGA [10]). Note also that the SVRGO based algorithm for single machine setting in [30] is for a differently structured problem than (SPP). The C-DPSVRG methodology using SVRGO is illustrated in Algorithm 3 with step sizes defined in Appendix G.

---

**Algorithm 3** **D**ecentralized **P**roximal **S**tochastic **V**ariance **R**eduction method with Compression (C-DPSVRG)

---

1: **INPUT:** $x_0, y_0, D_0^x = D_0^y = 0, H_0^x = x_0, H_0^y = y_0, H_0^{w,x} = (W \otimes I_{d_x})x_0, H_0^{w,y} = (W \otimes I_{d_y})y_0, s = \frac{\mu n p_{\min}}{24L^2}, \alpha_x, \alpha_y, \gamma_x, \gamma_y, \mathcal{G}$ defined using SVRGO.
2: **for** $t = 0$ **to** $T - 1$ in parallel for all nodes $i$ **do**
3:     $x_{t+1}, y_{t+1}, D_{t+1}^x, D_{t+1}^y, H_{t+1}^x, H_{t+1}^y, H_{t+1}^{w,x}, H_{t+1}^{w,y}$
           $=\text{IPDHG}(x_t, y_t, D_t^x, D_t^y, H_t^x, H_t^y, H_t^{w,x}, H_t^{w,y}, s, \gamma^x, \gamma^y, \alpha_x, \alpha_y, \mathcal{G})$
4: **end for**
5: **RETURN:** $x_T, y_T$.

---

Under suitable assumptions on smoothness of mini-batch gradients (see Appendix G), the convergence behavior of Algorithm 3 is given in the following result.

**Theorem 5.1.** *Let $\{x_t\}_t, \{y_t\}_t$ be the sequences generated by Algorithm 3. Suppose Assumptions 3.1-3.5 and Assumptions G.1-G.4 hold. Then computational and communication complexity of algorithm 3 for achieving $\epsilon$-accurate saddle point solution in expectation are*

$$T(\epsilon) = \mathcal{O}\left( \max\left\{ \frac{\sqrt{\delta}(1+\delta)\kappa_g\kappa_f^2}{np_{\min}}, (1+\delta)\kappa_g, \frac{(1+\delta)\kappa_f^2}{np_{\min}}, \frac{2}{p} \right\} \log\left( \frac{\tilde{\Phi}_0}{\epsilon} \right) \right)$$

*where $\tilde{\Phi}_0$ denotes the distance of the initial values $x_0, y_0, D_0^x, D_0^y, H_0^x, H_0^y$ from their respective limit points (described in equation (215) in Appendix G).*

Due to space considerations, we have given proof details of Theorem 5.1 in Appendix H. The complexity of C-DPSVRG depends on $\kappa_f, \kappa_g$ and $\delta$ as $\mathcal{O}((1+\delta)\max\{\sqrt{\delta}\kappa_f^2\kappa_g, \kappa_g, \kappa_f^2\})$. However, the optimal complexity for solving decentralized saddle point problems without compression is shown to be of order $\mathcal{O}(\kappa_f\sqrt{\kappa_g})$ in [17]. Such optimal dependence on $\kappa_g$ is obtained at the cost of multiple communication rounds per iterate unlike one communication round per iterate in C-DPSVRG. When there is no compression ($\delta = 0$) in C-DPSVRG, the complexity reduces to $\mathcal{O}\left( \max\{\kappa_g, \frac{\kappa_f^2}{np_{\min}}, \frac{2}{p}\} \log\left( \frac{\tilde{\Phi}_0}{\epsilon} \right) \right)$. We note that the complexity results in Theorem 5.1 are not optimal, however we observe convergence speedup for C-DPSVRG over baseline methods in our empirical study.

The following corollary of Theorem 5.1 gives particular settings of number of batches $n$ and reference probability $p$ yielding factors of the form $\sqrt{N_\ell} + N_\ell$ for total gradient computations per node, which resemble the factors in the corresponding complexity results for optimal algorithms to solve decentralized variational inequalities without compression [17].

**Corollary 5.2.** *Under the setting of Theorem 5.1, choose $n > \sqrt{N_\ell}, p = B/\sqrt{N_\ell}$ and $p_{\min} = \sqrt{N_\ell}/2n^2$. Then the total number of gradient computations per node to achieve $\epsilon$-accurate saddle point solution is of order $\mathcal{O}((\sqrt{N_\ell} + N_\ell) \max\{\sqrt{\bar{\delta}}(1+\delta)\kappa_g\kappa_f^2, \frac{(1+\delta)\kappa_g}{2}, (1+\delta)\kappa_f^2\} \log(\frac{\tilde{\Phi}_0}{\epsilon}))$.*

## 6   Conclusion

We have proposed two stochastic gradient algorithms for decentralized optimization for saddle point problems, with compression. Both the algorithms offer practical advantages and are shown to have rigorous theoretical guarantees. It would be interesting to adapt both C-RDPSG and C-DPSVRG to cases where some of the constants are unknown in the problem setup.

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

# A Related Work

**Decentralized minimization:** We provide here only a very brief survey of literature on decentralized minimization (see *e.g.* [20] for more comprehensive survey). A simple distributed deterministic gradient method with sublinear convergence rate guarantees was proposed in [32]. Distributed gradient methods using gradient tracking [34, 31] are known to converge to an exact optimal value using constant step size. Stochastic gradient oracles for decentralized minimization problems proposed in [28], have recently been extended to design efficient compression based decentralized methods for solving convex minimization problems with provable theoretical guarantees [15, 24, 20]. More recent works are proposed for non-convex optimization in decentralized settings [41].

| | | Algorithm | SG | Non-smooth | Type of functions | Computation Complexity | Communication Complexity |
|---|---|---|---|---|---|---|---|
| **No compression** | **Gossip** | MMDS [5] | ✗ | ✗ | SC-SC | – | $\tilde{\mathcal{O}}(\log^2(\frac{1}{\epsilon}))$ |
| | | DES [4] | ✓ | ✗ | SC-SC | $\tilde{\mathcal{O}}(\frac{\kappa_f^2}{L^2\epsilon})$ | $\tilde{\mathcal{O}}(\kappa_f\sqrt{\kappa_g}\log(1/\epsilon))$ |
| | | Algorithm 1+3 [17] | ✓ | ✓ | SC-SC | $\mathcal{O}\left(\kappa_f\log(1/\epsilon)\right)$ | $\mathcal{O}\left(\kappa_f\sqrt{\kappa_g}\log(1/\epsilon)\right)$ |
| | | DPOSG [23] | ✓ | ✗ | NC-NC | $\mathcal{O}(\frac{1}{\epsilon^{12}})$ | $\mathcal{O}(\frac{1}{\epsilon^{12}})$ |
| | **No Gossip** | GT-EG [29] | ✗ | ✗ | SC-SC | $\mathcal{O}\left(\kappa_f^{4/3}\kappa_g^{4/3}\log\left(\frac{1}{\epsilon}\right)\right)$ | $\mathcal{O}\left(\kappa_f^{4/3}\kappa_g^{4/3}\log\left(\frac{1}{\epsilon}\right)\right)$ |
| | | DSPAwLA[26] | ✗ | ✓ | C-C | $\mathcal{O}(1/\epsilon^2)$ | $\mathcal{O}(1/\epsilon^2)$ |
| | | DMHSGD [36] | ✓ | ✗ | NC-SC | $\mathcal{O}(\frac{\kappa^3}{(1-\lambda_2(W))^2\epsilon^3})$ | $\mathcal{O}(\frac{\kappa^3}{(1-\lambda_2(W))^2\epsilon^3})$ |
| | | C-RDPSG (Theorem 4.1) (GSS) | ✓ | ✓ | SC-SC | $\mathcal{O}(\frac{\kappa_f^2\kappa_g^2}{L^2\epsilon})$ | $\mathcal{O}(\frac{\kappa_f^2\kappa_g^2}{L^2\epsilon})$ |
| | | C-DPSVRG (Theorem 5.1) (FSS) | ✓ | ✓ | SC-SC | $\mathcal{O}(\max\{\kappa_f^2,\kappa_g\}\log(\frac{1}{\epsilon}))$ | $\mathcal{O}(\max\{\kappa_f^2,\kappa_g\}\log(\frac{1}{\epsilon}))$ |
| **Compression** | **No Gossip** | C-RDPSG (Theorem 4.1) (GSS) | ✓ | ✓ | SC-SC | $\mathcal{O}(\frac{(1+\delta)^4\kappa_f^2\kappa_g^2}{L^2\epsilon})$ | $\mathcal{O}(\frac{(1+\delta)^4\kappa_f^2\kappa_g^2}{L^2\epsilon})$ |
| | | C-DPSVRG (Theorem 5.1) (FSS) | ✓ | ✓ | SC-SC | $\mathcal{O}((1+\delta)\max\{\kappa_f^2,\sqrt{\delta}\kappa_f^2\kappa_g,\kappa_g\}\log(\frac{1}{\epsilon}))$ | $\mathcal{O}((1+\delta)\max\{\kappa_f^2,\sqrt{\delta}\kappa_f^2\kappa_g,\kappa_g\}\log(\frac{1}{\epsilon}))$ |

Table 1: Comparison of proposed optimization algorithms for decentralized saddle point problems with state-of-the-art algorithms. SG denotes Stochastic Gradient. Abbreviations SC-SC, C-C, NC-NC respectively denote Strongly convex-Strongly concave, Convex-Concave, Nonconvex-Nonconcave. GSS and FSS stands respectively for general stochastic setting and finite sum setting

**Decentralized saddle-point problems:** A distributed saddle-point algorithm with Laplacian averaging (DSPAwLA) in [26], based on gradient descent ascent updates to solve non-smooth convex-concave saddle point problems achieves $\mathcal{O}(1/\epsilon^2)$ convergence rate. DSPAwLA uses consensus constrained formulation of saddle point problem. DSPAwLA is obtained by incorporating $\ell_2$ norm based penalty of consensus constraints into the objective function and employing gradient descent ascent scheme to the resultant penalized objective. However, our work proposes an equivalent Lagrangian formulation of consensus constrained saddle point problem (1) and updates primal-dual variables using a variant of primal dual hybrid method. An extragradient method with gradient tracking (GT-EG) proposed in [29] is shown to have linear convergence rates for solving decentralized strongly convex strongly concave problems, under a positive lower bound assumption on the gradient difference norm. However such assumptions might not hold for problems without bilinear structure. Both [29] and [26] are based on non-compression based communications and full batch gradient computations which limit their applicability to large scale machine learning problems. Recently, multiple works [23, 36, 4] have proposed using minibatch gradients for solving decentralized saddle point problems. Decentralized extra step (DES) [4] shows linear communication complexity with dependence on the graph condition number as $\sqrt{\kappa_g}$, obtained at the cost of incorporating multiple rounds of communication of primal and dual updates. A near optimal distributed Min-Max data

similarity (MMDS) algorithm is proposed in [5] for saddle point problems with a suitable data similarity assumption. MMDS is based on full batch gradient computations and requires solving an inner saddle point problem at every iteration. MMDS allows for communication efficiency by choosing only one node uniformly at random to update the iterates. However, every node computes the full batch gradient before heading to next gradient based updates. Moreover, this scheme employs accelerated gossip [21] multiple times to propagate the gradients and model updates to the entire network. Communication complexity of MMDS is shown to depend on eigengap of weight matrix $W$, while gradient computation complexity is not investigated. Decentralized parallel optimistic stochastic gradient method (DPOSG) was proposed in [23] for nonconvex-nonconcave saddle point problems. This method involves local model averaging step (multiple communication rounds) to reduce the effect of consensus error. A gradient tracking based algorithm called DMHSGD for solving nonconvex-strongly concave saddle point problems proposed in [36], uses a large mini-batch at the first iteration and requires the nodes to communicate both model and gradient updates, to achieve better aggregates of quantities. Variance reduction based optimal methods to solve strongly convex-strongly concave nonsmooth finite sum variational inequalities are developed in [17]. The improvement of complexity on graph condition number $\kappa_g$ in [17] is achieved using an accelerated gossip scheme. However, C-DPSVRG does not involve any gossip scheme and hence yields cheaper communication per iterate. The complexity of C-DPSVRG and C-RDPSG does not have optimal dependence on $\kappa_f$, $\kappa_g$ and we leave it for future work. Table 1 positions our work in the context of existing methods.

## B  Compression Algorithm of [24]

We follow [24, 27] to compress a related difference vector instead of directly compressing $\nu_{t+1}^{\mathbf{x}}$ and $\nu_{t+1}^{\mathbf{y}}$. We now describe the compression related updates for $\nu_{t+1}^{\mathbf{x}}$. Each node $i$ is assumed to maintain a local vector $H^{i,\mathbf{x}}$ and a stochastic compression operator $Q$ is applied on the difference vector $\nu_{t+1}^{i,\mathbf{x}} - H_t^{i,\mathbf{x}}$. Hence the compressed estimate $\hat{\nu}_{t+1}^{i,\mathbf{x}}$ of $\nu_{t+1}^{i,\mathbf{x}}$ is obtained by adding the local vector and the compressed difference vector using $\hat{\nu}_{t+1}^{i,\mathbf{x}} = H_t^{i,\mathbf{x}} + Q(\nu_{t+1}^{i,\mathbf{x}} - H_t^{i,\mathbf{x}})$. The local vector $H_t^{i,\mathbf{x}}$ is then updated using a convex combination of the previous local vector information and the new estimate $\hat{\nu}_{t+1}^{i,\mathbf{x}}$ using $H_{t+1}^{i,\mathbf{x}} = (1-\alpha)H_t^{i,\mathbf{x}} + \alpha\hat{\nu}_{t+1}^{i,\mathbf{x}}$ for a suitable $\alpha \in [0,1]$. Collecting the quantities in individual nodes into $H_{t+1}^{\mathbf{x}} = (H_{t+1}^{1,\mathbf{x}}, \ldots, H_{t+1}^{m,\mathbf{x}})$ and $\nu_{t+1}^{\mathbf{x}} = (\nu_{t+1}^{1,\mathbf{x}}, \ldots, \nu_{t+1}^{m,\mathbf{x}})$, the update step can be written as $H_{t+1}^{\mathbf{x}} = (1-\alpha)H_t^{\mathbf{x}} + \alpha\hat{\nu}_{t+1}^{\mathbf{x}}$. Pre-multiplying both sides of $H_{t+1}^{\mathbf{x}}$ update by $W \otimes I$, and denoting $(W \otimes I)H_t^{\mathbf{x}}$ by $H_t^{w,\mathbf{x}}$, and $(W \otimes I)\hat{\nu}_t^{\mathbf{x}}$ by $\hat{\nu}_{t+1}^{w,\mathbf{x}}$ we have: $H_{t+1}^{w,\mathbf{x}} = (1-\alpha)H_t^{w,\mathbf{x}} + \alpha\hat{\nu}_{t+1}^{w,\mathbf{x}}$. $\hat{\nu}_{t+1}^{w,\mathbf{x}}$ can be further simplified as $\hat{\nu}_{t+1}^{w,\mathbf{x}} = (W \otimes I)\hat{\nu}_{t+1}^{\mathbf{x}} = (W \otimes I)(H_t^{\mathbf{x}} + Q(\nu_{t+1}^{\mathbf{x}} - H_t^{\mathbf{x}})) = H_t^{w,\mathbf{x}} + (W \otimes I)Q(\nu_{t+1}^{\mathbf{x}} - H_t^{\mathbf{x}})$. A similar update scheme is used for compressing $\nu_{t+1}^{\mathbf{y}}$. The entire procedure is illustrated in Algorithm 4, where we have used $\nu_{t+1} = (\nu_{t+1}^{\mathbf{x}}, \nu_{t+1}^{\mathbf{y}})$, $H_t = (H_t^{\mathbf{x}}, H_t^{\mathbf{y}})$, $H_t^w = (H^{w,\mathbf{x}_t}, H^{w,\mathbf{y}_t})$. Further recall that $\nu_{t+1}^{\mathbf{x}} = (\nu_{t+1}^{1,\mathbf{x}}, \ldots, \nu_{t+1}^{m,\mathbf{x}})$ denotes the collection of the local variables at $m$ nodes. Similar is the case for the other variables $\nu_{t+1}^{\mathbf{y}}, H_t^{\mathbf{x}}, H_t^{\mathbf{y}}, H_t^{w,\mathbf{x}}, H_t^{w,\mathbf{y}}$.

---

**Algorithm 4** Compressed Communication Procedure (COMM) [24]

1: **INPUT:** $\nu_{t+1}, H_t, H_t^w, \alpha$
2: $Q_t^i = Q(\nu_{t+1}^i - H_t^i)$  {(compression)}
3: $\hat{\nu}_{t+1}^i = H_t^i + Q_t^i$ ,
4: $H_{t+1}^i = (1-\alpha)H_t^i + \alpha\hat{\nu}_{t+1}^i$ ,
5: $\hat{\nu}_{t+1}^{i,w} = H_t^{i,w} + \sum_{j=1}^m W_{ij}Q_t^j$ ,  {(communicating compressed vectors)}
6: $H_{t+1}^{i,w} = (1-\alpha)H_t^{i,w} + \alpha\hat{\nu}_{t+1}^{i,w}$ ,
7: **RETURN:** $\hat{\nu}_{t+1}^i, \hat{\nu}_{t+1}^{i,w}, H_{t+1}^i, H_{t+1}^{i,w}$ for each node $i$ .

---

## C  Basic Results and Inequalities

**Equivalence between problems** (1) **and** (3) **.**

**Theorem C.1.** *Under assumptions of compactness of feasible sets, (sub)gradient boundedness and continuity of $f_i(x,y)$, $g(x)$ and $r(y)$ over $\mathcal{X}$ and $\mathcal{Y}$, problem* (1) *is equivalent to problem* (3) *in the sense that for any solution $(\mathbf{x}^\star, \mathbf{y}^\star, \tilde{S}^{\mathbf{x}}, \tilde{S}^y)$ of* (3)*, the point $(\mathbf{x}^\star, \mathbf{y}^\star)$ is a solution to* (1)*.*

*Proof.* The proof is based on the analysis of a similar result in [33]. Let $\tilde{\Psi}(\mathbf{x}, \mathbf{y}) = F(\mathbf{x}, \mathbf{y}) + \sum_{i=1}^{m}(g(x_i) - r(y_i))$. For simplicity of notation, we assume $U \otimes I_{d_x} = U_x$ and $U \otimes I_{d_y} = U_y$. The objective function $\tilde{\Psi}(\mathbf{x}, \mathbf{y})$ is convex-concave and feasible sets $\mathcal{X}^m, \mathcal{Y}^m$ are compact. Then, using Sion-Kakutani Theorem [2],

$$\min_{\substack{\mathbf{x}\in\mathcal{X}^m \\ U_x\mathbf{x}=0}} \max_{\substack{\mathbf{y}\in\mathcal{Y}^m \\ U_y\mathbf{y}=0}} \tilde{\Psi}(\mathbf{x}, \mathbf{y}) = \max_{\substack{\mathbf{y}\in\mathcal{Y}^m \\ U_y\mathbf{y}=0}} \min_{\substack{\mathbf{x}\in\mathcal{X}^m \\ U_x\mathbf{x}=0}} \tilde{\Psi}(\mathbf{x}, \mathbf{y}). \tag{8}$$

Consider the following problem:

$$\min_{\mathbf{x}\in\mathcal{X}^m} \tilde{\Psi}(\mathbf{x}, \mathbf{y}) \text{ such that } U_x\mathbf{x} = 0. \tag{9}$$

We assume that $(\mathbf{x}^\star, \mathbf{y}^\star)$ is the saddle point solution of problem (1). Therefore, constraint qualification $(U_x\mathbf{x}^\star = 0)$ holds. Then using Lagrange strong duality,

$$\min_{\substack{\mathbf{x}\in\mathcal{X}^m \\ U_x\mathbf{x}=0}} \tilde{\Psi}(\mathbf{x}, \mathbf{y}) = \max_{S^\mathbf{x}} \min_{\mathbf{x}\in\mathcal{X}^m} \tilde{\Psi}(\mathbf{x}, \mathbf{y}) + \langle S^\mathbf{x}, U_x\mathbf{x} \rangle. \tag{10}$$

Since $\mathcal{X}$ and $\mathcal{Y}$ are compact sets, the gradients and subgradients respectively of $f_i(x, y)$, $g(x)$ and $r(y)$ with respect to $x$ will be bounded by a suitable constant $B_1$. Therefore using Theorem 2 in [19], there exists an optimal dual multiplier $\tilde{S}^\mathbf{x}$ for r.h.s in (10) such that $\|\tilde{S}^\mathbf{x}\|_2 \leq \frac{\sqrt{m}B_1}{\lambda_{\min}^+(U_x)} =: R_1$, where $\lambda_{\min}^+(U_x)$ denotes the smallest nonzero eigenvalue of $U_x$. Therefore,

$$\max_{S^\mathbf{x}} \min_{\mathbf{x}\in\mathcal{X}^m} \tilde{\Psi}(\mathbf{x}, \mathbf{y}) + \langle S^\mathbf{x}, U_x\mathbf{x} \rangle = \max_{\|S^\mathbf{x}\|_2 \leq R_1} \min_{\mathbf{x}\in\mathcal{X}^m} \tilde{\Psi}(\mathbf{x}, \mathbf{y}) + \langle S^\mathbf{x}, U_\mathbf{x} \rangle. \tag{11}$$

This implies that (10) can be rewritten as

$$\min_{\substack{\mathbf{x}\in\mathcal{X}^m \\ U_x\mathbf{x}=0}} \tilde{\Psi}(\mathbf{x}, \mathbf{y}) = \max_{\|S^\mathbf{x}\|_2 \leq R_1} \min_{\mathbf{x}\in\mathcal{X}^m} \tilde{\Psi}(\mathbf{x}, \mathbf{y}) + \langle S^\mathbf{x}, U_\mathbf{x} \rangle. \tag{12}$$

Plugging above equality into (8) and by repeated application of Sion-Kakutani theorem [2], we have:

$$\min_{\substack{\mathbf{x}\in\mathcal{X}^m \\ U_x\mathbf{x}=0}} \max_{\substack{\mathbf{y}\in\mathcal{Y}^m \\ U_y\mathbf{y}=0}} \tilde{\Psi}(\mathbf{x}, \mathbf{y}) = \max_{\substack{\mathbf{y}\in\mathcal{Y}^m \\ U_y\mathbf{y}=0}} \left[ \max_{\|S^\mathbf{x}\|_2 \leq R_1} \min_{\mathbf{x}\in\mathcal{X}^m} \tilde{\Psi}(\mathbf{x}, \mathbf{y}) + \langle S^\mathbf{x}, U_x\mathbf{x} \rangle \right]$$

$$= \max_{\substack{\mathbf{y}\in\mathcal{Y}^m \\ U_y\mathbf{y}=0}} \left[ \min_{\mathbf{x}\in\mathcal{X}^m} \max_{\|S^\mathbf{x}\|_2 \leq R_1} \tilde{\Psi}(\mathbf{x}, \mathbf{y}) + \langle S^\mathbf{x}, U_x\mathbf{x} \rangle \right]$$

$$= \min_{\mathbf{x}\in\mathcal{X}^m} \max_{\substack{\mathbf{y}\in\mathcal{Y}^m \\ U_y\mathbf{y}=0}} \max_{\|S^\mathbf{x}\|_2 \leq R_1} \tilde{\Psi}(\mathbf{x}, \mathbf{y}) + \langle S^\mathbf{x}, U_x\mathbf{x} \rangle$$

$$= \min_{\mathbf{x}\in\mathcal{X}^m} \max_{\|S^\mathbf{x}\|_2 \leq R_1} \max_{\substack{\mathbf{y}\in\mathcal{Y}^m \\ U_y\mathbf{y}=0}} \tilde{\Psi}(\mathbf{x}, \mathbf{y}) + \langle S^\mathbf{x}, U_x\mathbf{x} \rangle, \tag{13}$$

where the last equality follows from the previous equality due to separability of the objective function in (13) in $\mathbf{y}$ and $S^\mathbf{x}$. Next, we consider a maximization problem associated with the consensus constraint $U_y\mathbf{y} = 0$ to write (13) in the form of (3). Towards this end, consider

$$\max_{\mathbf{y}\in\mathcal{Y}^m} \tilde{\Psi}(\mathbf{x}, \mathbf{y}) + \langle S^\mathbf{x}, U_x\mathbf{x} \rangle \text{ such that } U_y\mathbf{y} = 0. \tag{14}$$

The dual formulation of (14) is given by

$$\min_{S^\mathbf{y}} \max_{\mathbf{y}\in\mathcal{Y}^m} \tilde{\Psi}(\mathbf{x}, \mathbf{y}) + \langle S^\mathbf{x}, U_x\mathbf{x} \rangle + \langle S^\mathbf{y}, U_y\mathbf{y} \rangle.$$

Again using Lagrange strong duality, we have

$$\max_{\substack{\mathbf{y}\in\mathcal{Y}^m \\ U_y\mathbf{y}=0}} \tilde{\Psi}(\mathbf{x}, \mathbf{y}) + \langle S^\mathbf{x}, U_x\mathbf{x} \rangle = \min_{S^\mathbf{y}} \left[ \max_{\mathbf{y}\in\mathcal{Y}^m} \tilde{\Psi}(\mathbf{x}, \mathbf{y}) + \langle S^\mathbf{x}, U_x\mathbf{x} \rangle + \langle S^\mathbf{y}, U_y\mathbf{y} \rangle \right].$$

By following arguments similar to the primal problem with constraint $U_x\mathbf{x} = 0$, we can write

$$\max_{\substack{\mathbf{y}\in\mathcal{Y}^m \\ U_y\mathbf{y}=0}} \tilde{\Psi}(\mathbf{x}, \mathbf{y}) + \langle S^\mathbf{x}, U_x\mathbf{x} \rangle = \min_{\|S^\mathbf{y}\|_2 \leq R_2} \left[ \max_{\mathbf{y}\in\mathcal{Y}^m} \tilde{\Psi}(\mathbf{x}, \mathbf{y}) + \langle S^\mathbf{x}, U_x\mathbf{x} \rangle + \langle S^\mathbf{y}, U_y\mathbf{y} \rangle \right], \tag{15}$$

where $R_2 := \frac{\sqrt{m}B_2}{\lambda_{\min}^+(U_y)}$. By substituting (15) in (13), we obtain

$$
\min_{\substack{\mathbf{x}\in\mathcal{X}^m \ \mathbf{y}\in\mathcal{Y}^m \\ U_x\mathbf{x}=0 \; U_y\mathbf{y}=0}} \max \tilde{\Psi}(\mathbf{x},\mathbf{y}) = \min_{\mathbf{x}\in\mathcal{X}^m} \max_{\|S^{\mathbf{x}}\|_2\le R_1} \left[ \min_{\|S^{\mathbf{y}}\|_2\le R_2} \max_{\mathbf{y}\in\mathcal{Y}^m} \tilde{\Psi}(\mathbf{x},\mathbf{y}) + \langle S^{\mathbf{x}}, U_x\mathbf{x}\rangle + \langle S^{\mathbf{y}}, U_y\mathbf{y}\rangle \right]
$$

$$
= \min_{\mathbf{x}\in\mathcal{X}^m} \max_{\|S^{\mathbf{x}}\|_2\le R_1} \max_{\mathbf{y}\in\mathcal{Y}^m} \min_{\|S^{\mathbf{y}}\|_2\le R_2} \tilde{\Psi}(\mathbf{x},\mathbf{y}) + \langle S^{\mathbf{x}}, U_x\mathbf{x}\rangle + \langle S^{\mathbf{y}}, U_y\mathbf{y}\rangle
$$

$$
= \min_{\mathbf{x}\in\mathcal{X}^m} \max_{\substack{\|S^{\mathbf{x}}\|_2\le R_1 \; \|S^{\mathbf{y}}\|_2\le R_2 \\ \mathbf{y}\in\mathcal{Y}^m}} \tilde{\Psi}(\mathbf{x},\mathbf{y}) + \langle S^{\mathbf{x}}, U_x\mathbf{x}\rangle + \langle S^{\mathbf{y}}, U_y\mathbf{y}\rangle
$$

$$
= \min_{\mathbf{x}\in\mathcal{X}^m} \min_{\|S^{\mathbf{y}}\|_2\le R_2} \max_{\substack{\|S^{\mathbf{x}}\|_2\le R_1 \\ \mathbf{y}\in\mathcal{Y}^m}} \tilde{\Psi}(\mathbf{x},\mathbf{y}) + \langle S^{\mathbf{x}}, U_x\mathbf{x}\rangle + \langle S^{\mathbf{y}}, U_y\mathbf{y}\rangle
$$

$$
= \min_{\substack{\mathbf{x}\in\mathcal{X}^m \\ \|S^{\mathbf{y}}\|_2\le R_2}} \max_{\substack{\|S^{\mathbf{x}}\|_2\le R_1 \\ \mathbf{y}\in\mathcal{Y}^m}} \tilde{\Psi}(\mathbf{x},\mathbf{y}) + \langle S^{\mathbf{x}}, U_x\mathbf{x}\rangle + \langle S^{\mathbf{y}}, U_y\mathbf{y}\rangle. \tag{16}
$$

Since optimal Lagrange dual variables $\tilde{S}^{\mathbf{x}}$ and $\tilde{S}^{\mathbf{y}}$ always lie in $\ell_2$ balls of radius $R_1$ and $R_2$ respectively, equation (16) can be equivalently written as

$$
\min_{\substack{\mathbf{x}\in\mathcal{X}^m \ \mathbf{y}\in\mathcal{Y}^m \\ U_x\mathbf{x}=0 \; U_y\mathbf{y}=0}} \max \tilde{\Psi}(\mathbf{x},\mathbf{y}) = \min_{\mathbf{x}\in\mathcal{X}^m, S^{\mathbf{y}}} \max_{\mathbf{y}\in\mathcal{Y}^m, S^{\mathbf{x}}} \tilde{\Psi}(\mathbf{x},\mathbf{y}) + \langle S^{\mathbf{x}}, U_x\mathbf{x}\rangle + \langle S^{\mathbf{y}}, U_y\mathbf{y}\rangle
$$

$$
= \min_{\substack{\mathbf{x}\in\mathbb{R}^{md_x} \\ S^{\mathbf{y}}\in\mathbb{R}^{md_y}}} \max_{\substack{\mathbf{y}\in\mathbb{R}^{md_y} \\ S^{\mathbf{x}}\in\mathbb{R}^{md_x}}} F(\mathbf{x},\mathbf{y}) + G(\mathbf{x}) - R(\mathbf{y}) + \langle S^{\mathbf{x}}, U_x\mathbf{x}\rangle + \langle S^{\mathbf{y}}, U_y\mathbf{y}\rangle.
$$

This completes the proof of Theorem C.1.

$\square$

We now provide the optimality conditions for the optimization problem (SPP).

**Optimality Conditions of problem** (SPP). Let $f(x,y) := \sum_{i=1}^m f_i(x,y)$. Since $(x^\star, y^\star)$ is the saddle point solution to (SPP), we have $x^\star = \arg\min_{x\in\mathcal{X}} \Psi(x, y^\star)$ and $y^\star = \arg\max_{y\in\mathcal{Y}} \Psi(x^\star, y)$.

$$
0 \in \frac{1}{m}\nabla_x f(x^\star, y^\star) + \partial g(x^\star) + \partial I_{\mathcal{X}}(x^\star) \tag{17}
$$

$$
= \partial g(x^\star) + \partial I_{\mathcal{X}}(x^\star) + \frac{1}{s}\left(x^\star - \left(x^\star - \frac{s}{m}\nabla_x f(x^\star, y^\star)\right)\right) \tag{18}
$$

$$
= \partial \left(g(x) + I_{\mathcal{X}}(x) + \frac{1}{2s}\left\|x - \left(x^\star - \frac{s}{m}\nabla_x f(x^\star, y^\star)\right)\right\|^2\right)_{x=x^\star}. \tag{19}
$$

This implies that

$$
x^\star = \arg\min_{x\in\mathbb{R}^{d_x}} g(x) + I_{\mathcal{X}}(x) + \frac{1}{2s}\left\|x - \left(x^\star - \frac{s}{m}\nabla_x f(x^\star, y^\star)\right)\right\|^2 \tag{20}
$$

$$
= \arg\min_{x\in\mathcal{X}} g(x) + \frac{1}{2s}\left\|x - \left(x^\star - \frac{s}{m}\nabla_x f(x^\star, y^\star)\right)\right\|^2 \tag{21}
$$

$$
= \operatorname*{prox}_{sg}\left(x^\star - \frac{s}{m}\nabla_x f(x^\star, y^\star)\right). \tag{22}
$$

We also have $y^\star = \arg\min_{y\in\mathcal{Y}} -\Psi(x^\star, y) = \arg\min_{y\in\mathbb{R}^{d_y}}(-\Psi(x^\star, y) + I_{\mathcal{Y}}(y))$. Therefore,

$$
0 \in \partial_y(-\Psi(x^\star, y^\star)) + \partial I_{\mathcal{Y}}(y^\star) \tag{23}
$$

$$
= -\frac{1}{m}\nabla_y f(x^\star, y^\star) + \partial r(y^\star) + \partial I_{\mathcal{Y}}(y^\star) \tag{24}
$$

$$
= \partial r(y^\star) + \partial I_{\mathcal{Y}}(y^\star) + \frac{1}{s}\left(y^\star - \left(y^\star + \frac{s}{m}\nabla_y f(x^\star, y^\star)\right)\right) \tag{25}
$$

$$
= \partial \left(r(y) + I_{\mathcal{Y}} + \frac{1}{2s}\left\|y - \left(y^\star + \frac{s}{m}\nabla_y f(x^\star, y^\star)\right)\right\|^2\right)_{y=y^\star}. \tag{26}
$$

Therefore, $y^\star = \operatorname*{prox}_{sr}\left(y^\star + \frac{s}{m}\nabla_y f(x^\star, y^\star)\right).$

**Notations useful for further analysis:**
In the subsequent analysis, we assume $d_x = d_y = 1$ for simplicity of representation. Our analysis still holds for $d_x > 1$ and $d_y > 1$ by incorporating Kronecker product. We define Bregman distance

with respect to each function $f_i(\cdot, y)$ and $-f_i(x, \cdot)$ as

$$V_{f_i,y}(x_1, x_2) = f_i(x_1, y) - f_i(x_2, y) - \langle \nabla_x f_i(x_2, y), x_1 - x_2 \rangle \tag{27}$$

$$V_{-f_i,x}(y_1, y_2) = -f_i(x, y_1) + f_i(x, y_2) - \langle -\nabla_y f_i(x, y_2), y_1 - y_2 \rangle, \tag{28}$$

respectively. Let $L = \max\{L_{xx}, L_{yy}, L_{xy}, L_{yx}\}$ and $\mu = \min\{\mu_x, \mu_y\}$. Suppose $\lambda_{\max}(I - W)$, $\lambda_{m-1}(I - W)$ and $(I - W)^\dagger$ denote the largest eigenvalue, second smallest eigenvalue and pseudo inverse of $I - W$ respectively. Let $\kappa_f = L/\mu$ and $\kappa_g = \lambda_{\max}(I - W)/\lambda_{m-1}(I - W)$ denote the condition number of function $f$ and graph $G$ respectively. We further define $D_x^\star := -(I - J)\nabla_x F(\mathbf{1}z^\star)$, $D_y^\star := (I - J)\nabla_y F(\mathbf{1}z^\star)$, $H_x^\star := \mathbf{1}(x^\star - \frac{s}{m}\nabla_x f(z^\star))$, $H_y^\star := \mathbf{1}(y^\star + \frac{s}{m}\nabla_y f(z^\star))$, $\text{Range}(I - W) := \{(I - W)z : z \in \mathbb{R}^m\}$, $\text{Range}(\mathbf{1}) := \{\eta\mathbf{1} : \eta \in \mathbb{R}\}$ and $\text{Null}(I - W) := \{z : (I - W)z = 0\}$. We now state a few preliminary results that will be used in later sections. These results may be of independent interest as well.

**Proposition C.2.** *Let $W$ be a weight matrix satisfying assumption 3.5. Then $Null(I - W) = Range(\mathbf{1})$.*

*Proof.* We prove this result in two parts. We first show that $\text{Null}(I - W) \subseteq \text{Range}(\mathbf{1})$ and then show that $\text{Range}(\mathbf{1}) \subseteq \text{Null}(I - W)$. In this regard, let $y \in \text{Null}(I - W)$. Then we have $(I - W)y = 0$ which implies that $Wy = y$. Hence $y$ is an eigen vector of $W$ with eigen value 1. We know that algebraic multiplicity of eigenvalue 1 is one using assumption 3.5. Therefore, there is only one linearly independent eigenvector associated with eigenvalue 1. We also know that $\mathbf{1}$ is an eigenvector associated with eigenvalue 1 because $W\mathbf{1} = \mathbf{1}$. Therefore, $y$ must belong to $\text{Range}(\mathbf{1})$. This completes the first part of the proof. To prove the other part, let $y \in \text{Range}(\mathbf{1})$. Then $(I - W)y = (I - W)\eta\mathbf{1} = 0$. This shows that $y \in \text{Null}(I - W)$. By combining both the parts, we get the desired result. $\square$

**Proposition C.3.** *Let $W$ satisfy Assumption 3.5 and let $D_x^\star$ and $D_y^\star$ be as defined in above paragraph. Then, $D_x^\star \in Range(I - W)$ and $D_y^\star \in Range(I - W)$.*

*Proof.* To prove this result, we first show that $\text{Range}(I - W) = (\text{Range}(\mathbf{1}))^\perp$ using Assumption 3.5. Then we prove that both $D_x^\star$ and $D_y^\star$ lie in $(\text{Range}(\mathbf{1}))^\perp$.

$$\text{Range}(I - W) = \{(I - W)z : z \in \mathbb{R}^m\}, \ \text{Range}(\mathbf{1}) = \{\eta\mathbf{1} : \eta \in \mathbb{R}\}, \tag{29}$$

$$\text{Null}(I - W) = \{z : (I - W)z = 0\}, \tag{30}$$

$$(\text{Range}(\mathbf{1}))^\perp = \{x : \langle x, y \rangle = 0 \text{ for all } y \in \text{Range}(\mathbf{1})\}. \tag{31}$$

We first show that $\text{Range}(I - W) \subseteq (\text{Range}(\mathbf{1}^\top))^\perp$. Towards that end, let $y \in \text{Range}(I - W)$. This implies that there exists a $z \in \mathbb{R}^m$ such that $(I - W)z = y$. Therefore,

$$\langle y, \eta\mathbf{1} \rangle = \eta\mathbf{1}^\top y = \eta(\mathbf{1}^\top(I - W)z) = 0 \text{ for all } \eta \in \mathbb{R}. \tag{32}$$

The last step follows from $W\mathbf{1} = \mathbf{1}$. This implies that $y \in (\text{Range}(\mathbf{1}))^\perp$. Therefore, $\text{Range}(I - W) \subseteq (\text{Range}(\mathbf{1}))^\perp$. Next we show that $\dim(\text{Range}(I - W)) = \dim((\text{Range}(\mathbf{1}))^\perp)$. Using Proposition C.2, we have $\text{Null}(I - W) = \text{Range}(\mathbf{1})$. This implies that $\dim(\text{Null}(I - W)) = 1$. Using Rank-Nullity Theorem [11], we get $\dim(\text{Range}(I - W)) = m - 1$. Further $\text{Range}(\mathbf{1})$ is a one-dimensional subspace of $\mathbb{R}^m$ and hence $\dim((\text{Range}(\mathbf{1}))^\perp) = m - 1$. Therefore, $\dim(\text{Range}(I - W)) = \dim((\text{Range}(\mathbf{1}))^\perp)$. Using Theorem 1.11 in [11], we get $\text{Range}(I - W) = (\text{Range}(\mathbf{1}))^\perp$. This completes the first part of the proof. Recall

$$D_x^\star = -(I - J)\nabla_x F(\mathbf{1}z^\star) \tag{33}$$

$$D_y^\star = (I - J)\nabla_y F(\mathbf{1}z^\star). \tag{34}$$

Therefore, $\eta\mathbf{1}^\top D_x^\star = -\eta\mathbf{1}^\top(I - J)\nabla_x F(\mathbf{1}z^\star) = 0$ because $\mathbf{1}^\top J = \mathbf{1}^\top$. Similarly, $\eta\mathbf{1}^\top D_y^\star = \mathbf{1}^\top(I - J)\nabla_y F(\mathbf{1}z^\star) = 0$. Hence, $D_x^\star \in (\text{Range}(\mathbf{1}))^\perp = \text{Range}(I - W)$ and $D_y^\star \in (\text{Range}(\mathbf{1}))^\perp = \text{Range}(I - W)$. $\square$

**Proposition C.4.** *(**Smoothness in** $x$) Assume that $f(x, y)$ is convex and $L_{xx}$-smooth in $x$ for any fixed $y$. Then*

$$\frac{1}{2L_{xx}} \|\nabla_x f(x_1, y) - \nabla_x f(x_2, y)\|^2 \leq V_{f,y}(x_1, x_2) \leq \frac{L_{xx}}{2} \|x_1 - x_2\|^2 \text{ for all } x_1, x_2. \tag{35}$$

*Proof.* Using the smoothness of $f(\cdot, y)$, we have

$$f(x_1, y) \le f(x_2, y) + \langle \nabla_x f(x_2, y), x_1 - x_2 \rangle + \frac{L_{xx}}{2} \|x_1 - x_2\|^2 \tag{36}$$

$$f(x_1, y) - f(x_2, y) - \langle \nabla_x f(x_2, y), x_1 - x_2 \rangle \le \frac{L_{xx}}{2} \|x_1 - x_2\|^2 \tag{37}$$

$$V_{f,y}(x_1, x_2) \le \frac{L_{xx}}{2} \|x_1 - x_2\|^2 . \tag{38}$$

This completes the proof of second inequality. Let $h(x_1) := V_{f,y}(x_1, x_2)$ for a given $y$ and $x_2$. Notice that $h(x_1) = 0$ at $x_1 = x_2$. Using convexity of $f(x, y)$ in $x$, $h(x_1) \ge 0$. Therefore, $h(x_1)$ achieves its minimum value at $x_2$ and the minimum value is 0.

$$\left\| \nabla_x h(x_1) - \nabla_x h(x_1^{'}) \right\| = \left\| \nabla_x f(x_1, y) - \nabla_x f(x_1^{'}, y) \right\| \tag{39}$$

$$\le L_{xx} \left\| x_1 - x_1^{'} \right\| . \tag{40}$$

This implies that $h(x_1)$ is also $L_{xx}$-smooth.

$$h(\bar{x}) \le h(x_1) + \langle \nabla_x h(x_1), \bar{x} - x_1 \rangle + \frac{L_{xx}}{2} \|\bar{x} - x_1\|^2 \tag{41}$$

Take minimization over $\bar{x}$ on both sides.

$$\min h(\bar{x}) \le h(x_1) + \min_{\bar{x}} \left( \langle \nabla_x h(x_1), \bar{x} - x_1 \rangle + \frac{L_{xx}}{2} \|\bar{x} - x_1\|^2 \right) . \tag{42}$$

Let $\delta(\bar{x}) = \langle \nabla_x h(x_1), \bar{x} - x_1 \rangle + \frac{L_{xx}}{2} \|\bar{x} - x_1\|^2$ .

$$\nabla \delta(\bar{x}) = \nabla_x h(x_1) + L_{xx}(\bar{x} - x_1), \quad \nabla^2 \delta(\bar{x}) = L_{xx} I \succ 0. \tag{43}$$

Therefore,

$$\min_{\bar{x}} \delta(\bar{x}) = \left\langle \nabla_x h(x_1), \frac{-\nabla_x h(x_1)}{L_{xx}} + x_1 - x_1 \right\rangle + \frac{L_{xx}}{2} \left\| \frac{-\nabla_x h(x_1)}{L_{xx}} + x_1 - x_1 \right\|^2 \tag{44}$$

$$= \frac{-1}{L_{xx}} \|\nabla_x h(x_1)\|^2 + \frac{\|\nabla_x h(x_1)\|^2}{2L_{xx}} \tag{45}$$

$$= -\frac{\|\nabla_x h(x_1)\|^2}{2L_{xx}} . \tag{46}$$

Plug in above minimum value into (42).

$$\min h(\bar{x}) \le h(x_1) - \frac{\|\nabla_x h(x_1)\|^2}{2L_{xx}} \tag{47}$$

$$0 \le h(x_1) - \frac{\|\nabla_x h(x_1)\|^2}{2L_{xx}} \tag{48}$$

$$= V_{f,y}(x_1, x_2) - \frac{\|\nabla_x f(x_1, y) - \nabla_x f(x_2, y)\|^2}{2L_{xx}} . \tag{49}$$

This gives

$$\frac{\|\nabla_x f(x_1, y) - \nabla_x f(x_2, y)\|^2}{2L_{xx}} \le V_{f,y}(x_1, x_2). \tag{50}$$

$\square$

**Proposition C.5.** *(**Smoothness in** $y$) Assume that $-f(x, y)$ is convex and $L_{yy}$-smooth in $y$ for any fixed $x$. Then*

$$\frac{1}{2L_{yy}} \|-\nabla_y f(x, y_1) + \nabla_y f(x, y_2)\|^2 \le V_{-f,x}(y_1, y_2) \le \frac{L_{yy}}{2} \|y_1 - y_2\|^2 . \tag{51}$$

The proof of Proposition C.5 is similar to that of Proposition C.4, and is omitted.

## D   A recursion relationship useful for further analysis

In the following discussion, we use the shorthand notation $\mathbf{1}u$ to represent $(\mathbf{1} \otimes I_d)u$, for any $u \in \mathbb{R}^d$, and the notation $A^\dagger$ denotes the pseudo-inverse of a square matrix $A$.

**Lemma D.1.** *Let $x_{k,t+1}$, $y_{k,t+1}$, $D_{k,t+1}^x$, $D_{k,t+1}^y$, $H_{k,t+1}^x$, $H_{k,t+1}^y$, $H_{k,t+1}^{w,x}$, $H_{k,t+1}^{w,y}$ be obtained from Algorithm 1 using IPDHG($x_{k,t}$, $y_{k,t}$, $D_{k,t}^x$, $D_{k,t}^y$, $H_{k,t}^x$, $H_{k,t}^y$, $H_{k,t}^{w,x}$, $H_{k,t}^{w,y}$, $s$, $\gamma_x$, $\gamma_y$, $\alpha_x$, $\alpha_y$, $\mathcal{G}$). Let*

Assumption 3.4 and Assumption 3.5 hold. Suppose $\alpha_y \in (0, (1+\delta)^{-1})$ and

$$\gamma_y \in \left(0, \min\left\{\frac{2 - 2\sqrt{\delta}\alpha_y}{\lambda_{\max}(I-W)}, \frac{\alpha_y - (1+\delta)\alpha_y^2}{\sqrt{\delta}\lambda_{\max}(I-W)}\right\}\right) \tag{52}$$

Then the following holds for all $t \geq 0$:

$$M_y E \|y_{k,t+1} - \mathbf{1}y^\star\|^2 + \frac{2s^2}{\gamma_y} E \left\|D^y_{k,t+1} - D^\star_y\right\|^2_{(I-W)^\dagger} + \sqrt{\delta} E \left\|H^y_{k,t+1} - H^\star_y\right\|^2$$

$$\leq \left\|y_{k,t} - \mathbf{1}y^\star + s\mathcal{G}^y_{k,t} - s\nabla_y F(\mathbf{1}x^\star, \mathbf{1}y^\star)\right\|^2 + \frac{2s^2}{\gamma_y}\left(1 - \frac{\gamma_y}{2}\lambda_{m-1}(I-W)\right)\left\|D^y_{k,t} - D^\star_y\right\|^2_{(I-W)^\dagger}$$

$$+ \sqrt{\delta}(1 - \alpha_y)\left\|H^y_{k,t} - H^\star_y\right\|^2, \tag{53}$$

where $E$ denotes the conditional expectation on stochastic compression at $t$-th update step and $M_y = 1 - \frac{\sqrt{\delta}\alpha_y}{1 - \frac{\gamma_y}{2}\lambda_{\max}(I-W)}$ and $H^\star_x = \mathbf{1}(x^\star - \frac{s}{m}\nabla_x f(z^\star))$ and $H^\star_y = \mathbf{1}(y^\star + \frac{s}{m}\nabla_y f(z^\star))$. .

**Proof of Lemma D.1**:
We follow [20] to prove Lemma D.1. We have $H^\star_x = \mathbf{1}(x^\star - \frac{s}{m}\nabla_x f(x^\star, y^\star))$ and $H^\star_y = \mathbf{1}(y^\star + \frac{s}{m}\nabla_y f(x^\star, y^\star))$. First, we bound the terms appearing on the l.h.s. of (53) as

$$M_y E \|y_{k,t+1} - \mathbf{1}y^\star\|^2 \leq \left\|\nu^y_{k,t+1} - H^\star_y\right\|^2_{I - \frac{\gamma_y}{2}(I-W) - \alpha_y\sqrt{\delta}I} + \frac{\gamma_y^2 M_y}{4}E \left\|\hat{\nu}^y_{k,t+1} - \nu^y_{k,t+1}\right\|^2_{(I-W)^2}, \tag{54}$$

and

$$\left(\frac{2s^2}{\gamma_y}E \left\|D^y_{k,t+1} - D^\star_y\right\|^2_{(I-W)^\dagger} + \sqrt{\delta}E \left\|H^y_{k,t+1} - H^\star_y\right\|^2\right) + \left\|\nu^y_{k,t+1} - H^\star_y\right\|^2_{I - \frac{\gamma_y}{2}(I-W) - \alpha_y\sqrt{\delta}I}$$

$$= \frac{s^2}{\gamma_y}\left\|D^y_{k,t} - D^\star_y\right\|^2_{2(I-W)^\dagger - \gamma_y I} + \frac{1}{2}E \left\|\hat{\nu}^y_{k,t+1} - \nu^y_{k,t+1}\right\|^2_{\gamma_y(I-W) + 2\sqrt{\delta}\alpha_y^2} + \left\|y_{k,t} - \mathbf{1}y^\star + s\mathcal{G}^y_{k,t} - s\nabla_y F(\mathbf{1}z^\star)\right\|^2$$

$$+ \sqrt{\delta}(1 - \alpha_y)\left\|H^y_{k,t} - H^\star_y\right\|^2 - \sqrt{\delta}\alpha_y(1 - \alpha_y)\left\|\nu^y_{k,t+1} - H^y_{k,t}\right\|^2. \tag{55}$$

Proofs of (54) and (55) are provided in Sections D.1 and D.2, respectively.
On adding (54) and (55), we obtain

$$M_y E \|y_{k,t+1} - \mathbf{1}y^\star\|^2 + \frac{2s^2}{\gamma_y}E \left\|D^y_{k,t+1} - D^\star_y\right\|^2_{(I-W)^\dagger} + \sqrt{\delta}E \left\|H^y_{k,t+1} - H^\star_y\right\|^2$$

$$\leq \left\|y_{k,t} - \mathbf{1}y^\star + s\mathcal{G}^y_{k,t} - s\nabla_y F(\mathbf{1}z^\star)\right\|^2 + \frac{s^2}{\gamma_y}\left\|D^y_{k,t} - D^\star_y\right\|^2_{2(I-W)^\dagger - \gamma_y I}$$

$$+ \sqrt{\delta}(1 - \alpha_y)\left\|H^y_{k,t} - H^\star_y\right\|^2 - \sqrt{\delta}\alpha_y(1 - \alpha_y)\left\|\nu^y_{k,t+1} - H^y_{k,t}\right\|^2$$

$$+ \frac{1}{2}E \left\|\hat{\nu}^y_{k,t+1} - \nu^y_{k,t+1}\right\|^2_{\gamma_y(I-W) + 2\sqrt{\delta}\alpha_y^2} + \frac{\gamma_y^2 M_y}{4}E \left\|\hat{\nu}^y_{k,t+1} - \nu^y_{k,t+1}\right\|^2_{(I-W)^2}. \tag{56}$$

We now bound the terms on the r.h.s. of (56). First, observe that

$$\frac{1}{2}E \left\|\hat{\nu}^y_{k,t+1} - \nu^y_{k,t+1}\right\|^2_{\gamma_y(I-W) + 2\sqrt{\delta}\alpha_y^2} + \frac{\gamma_y^2 M_y}{4}E \left\|\hat{\nu}^y_{k,t+1} - \nu^y_{k,t+1}\right\|^2_{(I-W)^2}$$

$$= \frac{1}{2}E \left\|\sqrt{\gamma_y(I-W) + 2\sqrt{\delta}\alpha_y^2}(\hat{\nu}^y_{k,t+1} - \nu^y_{k,t+1})\right\|^2 + \frac{\gamma_y^2 M_y}{4}E \left\|(I-W)(\hat{\nu}^y_{k,t+1} - \nu^y_{k,t+1})\right\|^2$$

$$\leq \frac{1}{2}\left\|\sqrt{\gamma_y(I-W) + 2\sqrt{\delta}\alpha_y^2}\right\|^2 E \left\|\hat{\nu}^y_{k,t+1} - \nu^y_{k,t+1}\right\|^2 + \frac{\gamma_y^2 M_y}{4}\|I-W\|^2 E \left\|\hat{\nu}^y_{k,t+1} - \nu^y_{k,t+1}\right\|^2$$

$$\leq \left(\frac{1}{2}\left(\gamma_y\lambda_{\max}(I-W) + 2\sqrt{\delta}\alpha_y^2\right) + \frac{\gamma_y^2 M_y\lambda^2_{\max}(I-W)}{4}\right)E \left\|\hat{\nu}^y_{k,t+1} - \nu^y_{k,t+1}\right\|^2. \tag{57}$$

We also have

$$\hat{\nu}_{k,t+1}^y - \nu_{k,t+1}^y = Q(\nu_{k,t+1}^y - H_{k,t}^y) - \left(\nu_{k,t+1}^y - H_{k,t}^y\right) \tag{58}$$

$$E\left\|\hat{\nu}_{k,t+1}^y - \nu_{k,t+1}^y\right\|^2 = E\left\|Q(\nu_{k,t+1}^y - H_{k,t}^y) - \left(\nu_{k,t+1}^y - H_{k,t}^y\right)\right\|^2$$

$$\leq \delta\left\|\nu_{k,t+1}^y - H_{k,t}^y\right\|^2. \tag{59}$$

Substituting this inequality in (57), we bound the last two terms in the r.h.s of (56) as

$$\frac{1}{2}E\left\|\hat{\nu}_{k,t+1}^y - \nu_{k,t+1}^y\right\|_{\gamma_y(I-W)+2\sqrt{\delta}\alpha_y^2}^2 + \frac{\gamma_y^2 M_y}{4}E\left\|\hat{\nu}_{k,t+1}^y - \nu_{k,t+1}^y\right\|_{(I-W)^2}^2$$

$$\leq \left(\frac{1}{2}\left(\gamma_y\lambda_{\max}(I-W) + 2\sqrt{\delta}\alpha_y^2\right) + \frac{\gamma_y^2 M_y\lambda_{\max}^2(I-W)}{4}\right)\delta\left\|\nu_{k,t+1}^y - H_{k,t}^y\right\|^2. \tag{60}$$

We will now bound the term $\frac{s^2}{\gamma_y}\left\|D_{k,t}^y - D_y^\star\right\|_{2(I-W)^\dagger - \gamma_y I}^2$.

$$\frac{s^2}{\gamma_y}\left\|D_{k,t}^y - D_y^\star\right\|_{2(I-W)^\dagger - \gamma_y I}^2 = \frac{s^2}{\gamma_y}\left\|D_{k,t}^y - D_y^\star\right\|_{2(I-W)^\dagger}^2 - \frac{s^2}{\gamma_y}\left\|D_{k,t}^y - D_y^\star\right\|_{\gamma_y I}^2$$

$$= \frac{s^2}{\gamma_y}\left\|D_{k,t}^y - D_y^\star\right\|_{2(I-W)^\dagger}^2 - \frac{s^2\gamma_y}{\gamma_y}\left\|D_{k,t}^y - D_y^\star\right\|^2$$

$$= \frac{2s^2}{\gamma_y}\left\|D_{k,t}^y - D_y^\star\right\|_{(I-W)^\dagger}^2 - s^2\left\|D_{k,t}^y - D_y^\star\right\|^2$$

$$= \frac{2s^2}{\gamma_y}\left\|D_{k,t}^y - D_y^\star\right\|_{(I-W)^\dagger}^2 - s^2\left\|D_{k,t}^y - D_y^\star\right\|^2 + s^2\lambda_{m-1}(I-W)\left\|D_{k,t}^y - D_y^\star\right\|_{(I-W)^\dagger}^2$$

$$\quad - s^2\lambda_{m-1}(I-W)\left\|D_{k,t}^y - D_y^\star\right\|_{(I-W)^\dagger}^2$$

$$= s^2(D_{k,t}^y - D_y^\star)^\top\left(-I + \lambda_{m-1}(I-W)(I-W)^\dagger\right)(D_{k,t}^y - D_y^\star)$$

$$\quad + \frac{2s^2}{\gamma_y}\left\|D_{k,t}^y - D_y^\star\right\|_{(I-W)^\dagger}^2 - s^2\lambda_{m-1}(I-W)\left\|D_{k,t}^y - D_y^\star\right\|_{(I-W)^\dagger}^2$$

$$\leq \frac{2s^2}{\gamma_y}\left\|D_{k,t}^y - D_y^\star\right\|_{(I-W)^\dagger}^2 - s^2\lambda_{m-1}(I-W)\left\|D_{k,t}^y - D_y^\star\right\|_{(I-W)^\dagger}^2$$

$$= \frac{2s^2}{\gamma_y}\left(1 - \frac{\gamma_y}{2}\lambda_{m-1}(I-W)\right)\left\|D_{k,t}^y - D_y^\star\right\|_{(I-W)^\dagger}^2. \tag{61}$$

By substituting (60) and (61) in (56), we get

$$M_y E\|y_{k,t+1} - \mathbf{1}y^\star\|^2 + \frac{2s^2}{\gamma_y}E\left\|D_{k,t+1}^y - D_y^\star\right\|_{(I-W)^\dagger}^2 + \sqrt{\delta}E\left\|H_{k,t+1}^y - H_y^\star\right\|^2$$

$$\leq \left\|y_{k,t} - \mathbf{1}y^\star + s\mathcal{G}_{k,t}^y - s\nabla_y F(\mathbf{1}z^\star)\right\|^2 + \frac{2s^2}{\gamma_y}\left(1 - \frac{\gamma_y}{2}\lambda_{m-1}(I-W)\right)\left\|D_{k,t}^y - D_y^\star\right\|_{(I-W)^\dagger}^2$$

$$+ \left(\frac{\delta\gamma_y^2 M_y\lambda_{\max}^2(I-W)}{4} + \frac{\gamma_y\delta}{2}\lambda_{\max}(I-W) + \sqrt{\delta}\delta\alpha_y^2 - \sqrt{\delta}\alpha_y(1-\alpha_y)\right)\left\|\nu_{k,t+1}^y - H_{k,t}^y\right\|^2$$

$$+ \sqrt{\delta}(1-\alpha_y)\left\|H_{k,t}^y - H_y^\star\right\|^2. \tag{62}$$

The coefficient of $\left\|\nu_{k,t+1}^y - H_{k,t}^y\right\|^2$ in (62) is

$$
\frac{\delta\gamma_y^2 M_y \lambda_{\max}^2(I-W)}{4} + \frac{\gamma_y\delta}{2}\lambda_{\max}(I-W) + \sqrt{\delta}\delta\alpha_y^2 - \sqrt{\delta}\alpha_y(1-\alpha_y)
$$

$$
< \frac{\delta\gamma_y^2 \lambda_{\max}^2(I-W)}{4} + \frac{\gamma_y\delta}{2}\lambda_{\max}(I-W) + \sqrt{\delta}(1+\delta)\alpha_y^2 - \sqrt{\delta}\alpha_y
$$

$$
= \frac{\delta\gamma_y}{2}\frac{\gamma_y\lambda_{\max}(I-W)}{2}\lambda_{\max}(I-W) + \frac{\gamma_y\delta}{2}\lambda_{\max}(I-W) - \sqrt{\delta}(\alpha_y - (1+\delta)\alpha_y^2)
$$

$$
< \frac{\delta\gamma_y}{2}\lambda_{\max}(I-W) + \frac{\gamma_y\delta}{2}\lambda_{\max}(I-W) - \sqrt{\delta}(\alpha_y - (1+\delta)\alpha_y^2)
$$

$$
= \delta\gamma_y\lambda_{\max}(I-W) - \sqrt{\delta}(\alpha_y - (1+\delta)\alpha_y^2)
$$

$$
< 0, \text{ as } \gamma_y < \frac{\alpha_y - (1+\delta)\alpha_y^2}{\sqrt{\delta}\lambda_{\max}(I-W)}. \tag{63}
$$

In deriving (63), the first inequality follows from $M_y < 1$ (to be proved later in Section E.4) and the second inequality follows from $\frac{\gamma_y\lambda_{\max}(I-W)}{2} < 1$, since from assumption (52), it holds that $0 < \gamma_y < \frac{2-2\sqrt{\delta}\alpha_y}{\lambda_{\max}(I-W)} \leq \frac{2}{\lambda_{\max}(I-W)}$. As a result, (62) can be simplified as

$$
M_y E\left\|y_{k,t+1} - \mathbf{1}y^\star\right\|^2 + \frac{2s^2}{\gamma_y}E\left\|D_{k,t+1}^y - D_y^\star\right\|^2_{(I-W)^\dagger} + \sqrt{\delta}E\left\|H_{k,t+1}^y - H_y^\star\right\|^2
$$

$$
\leq \left\|y_{k,t} - \mathbf{1}y^\star + s\mathcal{G}_{k,t}^y - s\nabla_y F(\mathbf{1}z^\star)\right\|^2 + \frac{2s^2}{\gamma_y}\left(1 - \frac{\gamma_y}{2}\lambda_{m-1}(I-W)\right)\left\|D_{k,t}^y - D_y^\star\right\|^2_{(I-W)^\dagger}
$$

$$
+ \sqrt{\delta}(1-\alpha_y)\left\|H_{k,t}^y - H_y^\star\right\|^2, \tag{64}
$$

proving Lemma D.1. The remainder of this section is devoted to prove (54) and (55).

## D.1 Proof of (54)

First, observe that

$$
\|y_{k,t+1} - \mathbf{1}y^\star\|^2 = \sum_{i=1}^m \left\|y_{k,t+1}^i - y^\star\right\|^2
$$

$$
= \sum_{i=1}^m \left\|\operatorname*{prox}_{sr}(\hat{y}_{k,t+1}^i) - \operatorname*{prox}_{sr}\left(y^\star + \frac{s}{m}\nabla_y f(x^\star, y^\star)\right)\right\|^2
$$

$$
= \sum_{i=1}^m \left\|\operatorname*{prox}_{sr}(\hat{y}_{k,t+1}^i) - \operatorname*{prox}_{sr}\left(H_{i,y}^\star\right)\right\|^2
$$

$$
\leq \sum_{i=1}^m \left\|\hat{y}_{k,t+1}^i - H_{i,y}^\star\right\|^2 \text{ (from non-expansivity of prox)}
$$

$$
= \left\|\hat{y}_{k,t+1} - H_y^\star\right\|^2
$$

$$
= \left\|\nu_{k,t+1}^y - \frac{\gamma_y}{2}(I-W)\hat{\nu}_{k,t+1}^y - H_y^\star\right\|^2
$$

$$
= \left\|\nu_{k,t+1}^y - \frac{\gamma_y}{2}(I-W)(\hat{\nu}_{k,t+1}^y - \nu_{k,t+1}^y + \nu_{k,t+1}^y) - H_y^\star\right\|^2
$$

$$
= \left\|\left(I - \frac{\gamma_y}{2}(I-W)\right)(\nu_{k,t+1}^y - H_y^\star) - \frac{\gamma_y}{2}(I-W)(\hat{\nu}_{k,t+1}^y - \nu_{k,t+1}^y)\right\|^2
$$

$$
= \left\|\left(I - \frac{\gamma_y}{2}(I-W)\right)(\nu_{k,t+1}^y - H_y^\star)\right\|^2 + \frac{\gamma_y^2}{4}\left\|(I-W)(\hat{\nu}_{k,t+1}^y - \nu_{k,t+1}^y)\right\|^2
$$

$$
+ 2\left\langle\left(I - \frac{\gamma_y}{2}(I-W)\right)(\nu_{k,t+1}^y - H_y^\star), -\frac{\gamma_y}{2}(I-W)(\hat{\nu}_{k,t+1}^y - \nu_{k,t+1}^y)\right\rangle. \tag{65}
$$

By taking conditional expectation over stochastic compression at $t$-th iterate, we obtain

$$E \left\| y_{k,t+1} - \mathbf{1}y^\star \right\|^2 \leq \left\| \left( I - \frac{\gamma_y}{2}(I - W) \right) (\nu_{k,t+1}^y - H_y^\star) \right\|^2 + \frac{\gamma_y^2}{4} E \left\| (I - W)(\hat{\nu}_{k,t+1}^y - \nu_{k,t+1}^y) \right\|^2$$
$$- \gamma_y \left\langle \left( I - \frac{\gamma_y}{2}(I - W) \right) (\nu_{k,t+1}^y - H_y^\star), (I - W)E(\hat{\nu}_{k,t+1}^y - \nu_{k,t+1}^y) \right\rangle. \tag{66}$$

We have

$$\hat{\nu}_{k,t+1}^y - \nu_{k,t+1}^y = H_{k,t}^y + Q(\nu_{k,t+1}^y - H_{k,t}^y) - \nu_{k,t+1}^y$$
$$= Q(\nu_{k,t+1}^y - H_{k,t}^y) - \left( \nu_{k,t+1}^y - H_{k,t}^y \right)$$
$$E \left( \hat{\nu}_{k,t+1}^y - \nu_{k,t+1}^y \right) = E \left( Q(\nu_{k,t+1}^y - H_{k,t}^y) \right) - \left( \nu_{k,t+1}^y - H_{k,t}^y \right)$$
$$= 0. \tag{67}$$

The last equality follows from Assumption 3.4. By substituting the above equation in (66), we obtain

$$E \left\| y_{k,t+1} - \mathbf{1}y^\star \right\|^2 \leq \left\| \left( I - \frac{\gamma_y}{2}(I - W) \right) (\nu_{k,t+1}^y - H_y^\star) \right\|^2 + \frac{\gamma_y^2}{4} E \left\| (I - W)(\hat{\nu}_{k,t+1}^y - \nu_{k,t+1}^y) \right\|^2. \tag{68}$$

We will now convert the square norm terms of (68) into matrix-norm based terms. Towards that end, observe that

$$\left( (I - \frac{\gamma_y}{2}(I - W) \right)^2 = I + \frac{\gamma_y^2}{4}(I - W)^2 - \gamma_y(I - W)$$
$$= I - \frac{\gamma_y}{2}(I - W) + \frac{\gamma_y}{2}(I - W) + \frac{\gamma_y^2}{4}(I - W)^2 - \gamma_y(I - W)$$
$$= I - \frac{\gamma_y}{2}(I - W) - \frac{\gamma_y}{2}(I - W) + \frac{\gamma_y^2}{4}(I - W)^2$$
$$= I - \frac{\gamma_y}{2}(I - W) + \frac{\gamma_y}{2}(I - W) \left( -I + \frac{\gamma_y}{2}(I - W) \right)$$
$$= I - \frac{\gamma_y}{2}(I - W) + \frac{\gamma_y}{2}(I - W)^{1/2} \left( -I + \frac{\gamma_y}{2}(I - W) \right) (I - W)^{1/2}. \tag{69}$$

Note that $\left( -I + \frac{\gamma_y}{2}(I - W) \right)$ is a negative semidefinite matrix because $0 < \frac{\gamma_y}{2}\lambda_{\max}(I - W) < 1$ from the choice of $\gamma_y$. Using this fact in equation (69), we get $\forall x \in \mathbb{R}^m$,

$$x^\top \left( I - \frac{\gamma_y}{2}(I - W) \right)^2 x \leq x^\top \left( I - \frac{\gamma_y}{2}(I - W) \right) x. \tag{70}$$

Consider

$$\left\| \left( I - \frac{\gamma_y}{2}(I - W) \right) (\nu_{k,t+1}^y - H_y^\star) \right\|^2 = (\nu_{k,t+1}^y - H_y^\star)^\top \left( I - \frac{\gamma_y}{2}(I - W) \right)^2 (\nu_{k,t+1}^y - H_y^\star)$$
$$\leq (\nu_{k,t+1}^y - H_y^\star)^\top \left( I - \frac{\gamma_y}{2}(I - W) \right) (\nu_{k,t+1}^y - H_y^\star)$$
$$= \left\| \nu_{k,t+1}^y - H_y^\star \right\|_{I - \frac{\gamma_y}{2}(I - W)}^2. \tag{71}$$

Moreover,

$$\left\| (I - W)(\hat{\nu}_{k,t+1}^y - \nu_{k,t+1}^y) \right\|^2 = (\hat{\nu}_{k,t+1}^y - \nu_{k,t+1}^y)^\top (I - W)^2 (\hat{\nu}_{k,t+1}^y - \nu_{k,t+1}^y)$$
$$= \left\| \hat{\nu}_{k,t+1}^y - \nu_{k,t+1}^y \right\|_{(I - W)^2}^2. \tag{72}$$

On substituting above two equalities (71) and (72) in (68), we obtain

$$E \left\| y_{k,t+1} - \mathbf{1}y^\star \right\|^2 \leq \left\| \nu_{k,t+1}^y - H_y^\star \right\|_{I - \frac{\gamma_y}{2}(I - W)}^2 + \frac{\gamma_y^2}{4} E \left\| \hat{\nu}_{k,t+1}^y - \nu_{k,t+1}^y \right\|_{(I - W)^2}^2. \tag{73}$$

To complete the proof of (54), we first show that the first term on the r.h.s. of (73) is at most $M_y^{-1} \left\| \nu_{k,t+1}^y - H_y^\star \right\|_{I - \frac{\gamma_y}{2}(I-W) - \alpha_y \sqrt{\delta}I}^2$, where $M_y = 1 - \frac{\sqrt{\delta}\alpha_y}{1 - \frac{\gamma_y}{2}\lambda_{\max}(I-W)}$ . To this end, consider

$$\sqrt{\delta}\alpha_y I - \frac{\sqrt{\delta}\alpha_y}{1 - \frac{\gamma_y}{2}\lambda_{\max}(I-W)} \left( I - \frac{\gamma_y}{2}(I-W) \right)$$

$$= \frac{\sqrt{\delta}\alpha_y \left( 1 - \frac{\gamma_y}{2}\lambda_{\max}(I-W) \right) I - \sqrt{\delta}\alpha_y \left( I - \frac{\gamma_y}{2}(I-W) \right)}{1 - \frac{\gamma_y}{2}\lambda_{\max}(I-W)}$$

$$= \frac{1}{1 - \frac{\gamma_y}{2}\lambda_{\max}(I-W)} \left( -\frac{\sqrt{\delta}\alpha_y \gamma_y \lambda_{\max}(I-W)}{2}I + \frac{\sqrt{\delta}\alpha_y \gamma_y}{2}(I-W) \right). \qquad (74)$$

The largest eigenvalue of $\sqrt{\delta}\alpha_y I - \frac{\sqrt{\delta}\alpha_y}{1 - \frac{\gamma_y}{2}\lambda_{\max}(I-W)} \left( I - \frac{\gamma_y}{2}(I-W) \right)$ is

$$\frac{1}{1 - \frac{\gamma_y}{2}\lambda_{\max}(I-W)} \left( -\frac{\sqrt{\delta}\alpha_y \gamma_y \lambda_{\max}(I-W)}{2}I + \frac{\sqrt{\delta}\alpha_y \gamma_y}{2}\lambda_{\max}(I-W) \right)$$

$$= 0. \qquad (75)$$

Therefore, $\sqrt{\delta}\alpha_y I - \frac{\sqrt{\delta}\alpha_y}{1 - \frac{\gamma_y}{2}\lambda_{\max}(I-W)} \left( I - \frac{\gamma_y}{2}(I-W) \right)$ is negative semidefinite, and

$$x^\top \left( I - \frac{\gamma_y}{2}(I-W) \right) x = M_y^{-1} x^\top M_y \left( I - \frac{\gamma_y}{2}(I-W) \right) x$$

$$= M_y^{-1} x^\top \left( 1 - \frac{\sqrt{\delta}\alpha_y}{1 - \frac{\gamma_y}{2}\lambda_{\max}(I-W)} \right) \left( I - \frac{\gamma_y}{2}(I-W) \right) x$$

$$= M_y^{-1} x^\top \left( I - \frac{\gamma_y}{2}(I-W) - \frac{\sqrt{\delta}\alpha_y}{1 - \frac{\gamma_y}{2}\lambda_{\max}(I-W)} \left( I - \frac{\gamma_y}{2}(I-W) \right) \right) x$$

$$= M_y^{-1} x^\top \left( I - \frac{\gamma_y}{2}(I-W) - \sqrt{\delta}\alpha_y I \right) x$$

$$\quad + M_y^{-1} x^\top \left( \sqrt{\delta}\alpha_y I - \frac{\sqrt{\delta}\alpha_y}{1 - \frac{\gamma_y}{2}\lambda_{\max}(I-W)} \left( I - \frac{\gamma_y}{2}(I-W) \right) \right) x$$

$$\leq M_y^{-1} x^\top \left( I - \frac{\gamma_y}{2}(I-W) - \sqrt{\delta}\alpha_y I \right) x. \qquad (76)$$

Substituting $x = \nu_{k,t+1}^y - H_y^\star$ into the above inequality and using the definition of $\|x\|_A^2$, we obtain

$$\left\| \nu_{k,t+1}^y - H_y^\star \right\|_{I - \frac{\gamma_y}{2}(I-W)}^2 \leq M_y^{-1} \left\| \nu_{k,t+1}^y - H_y^\star \right\|_{I - \frac{\gamma_y}{2}(I-W) - \alpha_y \sqrt{\delta}I}^2. \qquad (77)$$

From (73) and (77), we see that

$$E\|y_{k,t+1} - \mathbf{1}y^\star\|^2 \leq M_y^{-1} \left\| \nu_{k,t+1}^y - H_y^\star \right\|_{I - \frac{\gamma_y}{2}(I-W) - \alpha_y \sqrt{\delta}I}^2 + \frac{\gamma_y^2}{4} E \left\| \hat{\nu}_{k,t+1}^y - \nu_{k,t+1}^y \right\|_{(I-W)^2}^2.$$

Multiplying throughout by $M_y$, the proof of (54) is complete.

## D.2  Proof of (55)

For every $k$, let $\mathcal{G}_{k,t}^x$ and $\mathcal{G}_{k,t}^y$ denote the stochastic gradient oracles at iterate $t$. For Algorithm 2, $\mathcal{G}_{k,t}^x = \nabla_x F(z_{k,t}; \xi_{k,t})$ and $\mathcal{G}_{k,t}^y = \nabla_y F(z_{k,t}, \xi_{k,t})$ are obtained using general stochastic gradient oracle. In Algorithm 3, $\mathcal{G}_{k,t}^x$ and $\mathcal{G}_{k,t}^y$ are obtained from the SVRG oracle.

**Step 1: Computing** $E \left\| D^y_{k,t+1} - D^\star_y \right\|^2_{(I-W)^\dagger}$

Observe that

$$D^y_{k,t+1} - D^\star_y = D^y_{k,t} + \frac{\gamma_y}{2s}(I-W)\hat{\nu}^y_{k,t+1} - D^\star_y$$

$$= D^y_{k,t} + \frac{\gamma_y}{2s}(I-W)\hat{\nu}^y_{k,t+1} - D^\star_y - \frac{\gamma_y}{2s}(I-W)H^\star_y$$

$$= D^y_{k,t} - D^\star_y + \frac{\gamma_y}{2s}(I-W)(\hat{\nu}^y_{k,t+1} - H^\star_y)$$

$$= D^y_{k,t} - D^\star_y + \frac{\gamma_y}{2s}(I-W)(\hat{\nu}^y_{k,t+1} - \nu^y_{k,t+1}) + \frac{\gamma_y}{2s}(I-W)(\nu^y_{k,t+1} - H^\star_y) \quad (78)$$

On pre-multiplying both sides by $\sqrt{(I-W)^\dagger}$ and taking square norm on the resulting equality, we obtain

$$\left\| \sqrt{(I-W)^\dagger}\left(D^y_{k,t+1} - D^\star_y\right) \right\|^2$$

$$= \left\| \sqrt{(I-W)^\dagger}(D^y_{k,t} - D^\star_y) + \frac{\gamma_y}{2s}\sqrt{(I-W)^\dagger}(I-W)(\nu^y_{k,t+1} - H^\star_y) \right\|^2$$

$$+ \frac{\gamma_y^2}{4s^2}\left\| \sqrt{(I-W)^\dagger}(I-W)(\hat{\nu}^y_{k,t+1} - \nu^y_{k,t+1}) \right\|^2$$

$$+ 2\left\langle \sqrt{(I-W)^\dagger}(D^y_{k,t} - D^\star_y) + \frac{\gamma_y}{2s}\sqrt{(I-W)^\dagger}(I-W)(\nu^y_{k,t+1} - H^\star_y), \frac{\gamma_y}{2s}\sqrt{(I-W)^\dagger}(I-W)(\hat{\nu}^y_{k,t+1} - \nu^y_{k,t+1}) \right\rangle.$$
$$(79)$$

By taking conditional expectation over compression at $t$-th iterate and using the result $E\left(\hat{\nu}^y_{k,t+1} - \nu^y_{k,t+1}\right) = 0$, we obtain

$$E\left\| \sqrt{(I-W)^\dagger}\left(D^y_{k,t+1} - D^\star_y\right) \right\|^2 = \left\| \sqrt{(I-W)^\dagger}(D^y_{k,t} - D^\star_y) + \frac{\gamma_y}{2s}\sqrt{(I-W)^\dagger}(I-W)(\nu^y_{k,t+1} - H^\star_y) \right\|^2$$

$$+ \frac{\gamma_y^2}{4s^2}E\left\| \sqrt{(I-W)^\dagger}(I-W)(\hat{\nu}^y_{k,t+1} - \nu^y_{k,t+1}) \right\|^2$$

$$= \left\| \sqrt{(I-W)^\dagger}(D^y_{k,t} - D^\star_y) \right\|^2 + \frac{\gamma_y^2}{4s^2}\left\| \sqrt{(I-W)^\dagger}(I-W)(\nu^y_{k,t+1} - H^\star_y) \right\|^2$$

$$+ 2\left\langle \sqrt{(I-W)^\dagger}(D^y_{k,t} - D^\star_y), \frac{\gamma_y}{2s}\sqrt{(I-W)^\dagger}(I-W)(\nu^y_{k,t+1} - H^\star_y) \right\rangle$$

$$+ \frac{\gamma_y^2}{4s^2}E\left\| \sqrt{(I-W)^\dagger}(I-W)(\hat{\nu}^y_{k,t+1} - \nu^y_{k,t+1}) \right\|^2. \quad (80)$$

We have

$$\left( \sqrt{(I-W)^\dagger}(I-W) \right)^\top \left( \sqrt{(I-W)^\dagger}(I-W) \right) = (I-W)\sqrt{(I-W)^\dagger}\sqrt{(I-W)^\dagger}(I-W)$$

$$= (I-W)(I-W)^\dagger(I-W)$$

$$= I - W, \quad (81)$$

where the last equality follows from the definition of pseudoinverse. Consider

$$\left\| \sqrt{(I-W)^\dagger}(I-W)(\nu^y_{k,t+1} - H^\star_y) \right\|^2$$

$$= (\nu^y_{k,t+1} - H^\star_y)^\top \left( \sqrt{(I-W)^\dagger}(I-W) \right)^\top \left( \sqrt{(I-W)^\dagger}(I-W) \right)(\nu^y_{k,t+1} - H^\star_y)$$

$$= (\nu^y_{k,t+1} - H^\star_y)^\top (I-W)(\nu^y_{k,t+1} - H^\star_y)$$

$$= \left\| (\nu^y_{k,t+1} - H^\star_y) \right\|^2_{I-W}. \quad (82)$$

Similarly, we see that $\left\|\sqrt{(I-W)^\dagger}(I-W)(\hat{\nu}^y_{k,t+1} - \nu^y_{k,t+1})\right\|^2 = \left\|\hat{\nu}^y_{k,t+1} - \nu^y_{k,t+1}\right\|^2_{I-W}$ Substituting in (80), we obtain

$$E\left\|D^y_{k,t+1} - D^\star_y\right\|^2_{(I-W)^\dagger} = \left\|D^y_{k,t} - D^\star_y\right\|^2_{(I-W)^\dagger} + \frac{\gamma_y^2}{4s^2}\left\|\nu^y_{k,t+1} - H^\star_y\right\|^2_{I-W} + \frac{\gamma_y^2}{4s^2}E\left\|\hat{\nu}^y_{k,t+1} - \nu^y_{k,t+1}\right\|^2_{I-W}$$

$$+ \frac{\gamma_y}{s}\left\langle\sqrt{(I-W)^\dagger}(D^y_{k,t} - D^\star_y), \sqrt{(I-W)^\dagger}(I-W)(\nu^y_{k,t+1} - H^\star_y)\right\rangle.$$
(83)

Now, we will simplify the last term of (83). From the property of adjoints,

$$\left\langle\sqrt{(I-W)^\dagger}(D^y_{k,t} - D^\star_y), \sqrt{(I-W)^\dagger}(I-W)(\nu^y_{k,t+1} - H^\star_y)\right\rangle$$

$$= \left\langle(I-W)\sqrt{(I-W)^\dagger}\sqrt{(I-W)^\dagger}(D^y_{k,t} - D^\star_y), \nu^y_{k,t+1} - H^\star_y\right\rangle$$

$$= \left\langle(I-W)(I-W)^\dagger(D^y_{k,t} - D^\star_y), \nu^y_{k,t+1} - H^\star_y\right\rangle.$$
(84)

Note that $D^\star_y \in \mathrm{Range}(I-W)$ using Proposition C.3. Further note that $D^y_{k,t} \in \mathrm{Range}(I-W)$ because of update process (Step 10) in Algorithm 1. Therefore, there exists $\tilde{D}^y_{k,t}$ and $\tilde{D}_y$ such that $D^y_{k,t} = (I-W)\tilde{D}^y_{k,t}$ and $D^\star_y = (I-W)\tilde{D}_y$.

$$(I-W)(I-W)^\dagger(D^y_{k,t} - D^\star_y) = (I-W)(I-W)^\dagger\left((I-W)\tilde{D}^y_{k,t} - (I-W)\tilde{D}_y\right)$$

$$= (I-W)(I-W)^\dagger(I-W)\left(\tilde{D}^y_{k,t} - \tilde{D}_y\right)$$

$$= (I-W)\left(\tilde{D}^y_{k,t} - \tilde{D}_y\right)$$

$$= (I-W)\tilde{D}^y_{k,t} - (I-W)\tilde{D}_y$$

$$= D^y_{k,t} - D^\star_y.$$
(85)

This gives

$$\left\langle\sqrt{(I-W)^\dagger}(D^y_{k,t} - D^\star_y), \sqrt{(I-W)^\dagger}(I-W)(\nu^y_{k,t+1} - H^\star_y)\right\rangle = \left\langle D^y_{k,t} - D^\star_y, \nu^y_{k,t+1} - H^\star_y\right\rangle.$$
(86)

On substituting above equality in (83), we obtain

$$E\left\|D^y_{k,t+1} - D^\star_y\right\|^2_{(I-W)^\dagger} = \left\|D^y_{k,t} - D^\star_y\right\|^2_{(I-W)^\dagger} + \frac{\gamma_y^2}{4s^2}\left\|\nu^y_{k,t+1} - H^\star_y\right\|^2_{I-W} + \frac{\gamma_y^2}{4s^2}E\left\|\hat{\nu}^y_{k,t+1} - \nu^y_{k,t+1}\right\|^2_{I-W}$$

$$+ \frac{\gamma_y}{s}\left\langle D^y_{k,t} - D^\star_y, \nu^y_{k,t+1} - H^\star_y\right\rangle.$$
(87)

Note that $\frac{1}{m}\mathbf{1}\nabla_y f(z^\star) = \frac{1}{m}\mathbf{1}\sum_{i=1}^m \nabla_y f_i(z^\star) = J\nabla_y F(\mathbf{1}z^\star)$. Then we can write $H^\star_y = \mathbf{1}y^\star + sJ\nabla_y F(\mathbf{1}z^\star)$. We have

$$\nu^y_{k,t+1} - H^\star_y = \nu^y_{k,t+1} - (\mathbf{1}y^\star + sJ\nabla_y F(\mathbf{1}z^\star))$$

$$= \nu^y_{k,t+1} - \left(-sD^\star_y + \mathbf{1}y^\star + s\nabla_y F(\mathbf{1}z^\star)\right)$$

$$= y_{k,t} + s\mathcal{G}^y_{k,t} - sD^y_{k,t} + sD^\star - \mathbf{1}y^\star - s\nabla_y F(\mathbf{1}z^\star)$$

$$= y_{k,t} - \mathbf{1}y^\star + s\left(\mathcal{G}^y_{k,t} - \nabla_y F(\mathbf{1}z^\star)\right) - s\left(D^y_{k,t} - D^\star_y\right).$$
(88)

In deriving equation (88), the second equality follows from the definition of $D^\star_y = (I-J)\nabla_y F(\mathbf{1}z^\star)$, and the third equality follows from the update step of $\nu^y_{k,t+1}$ (Step 9 in Algorithm 1).

Substituting (88) in (87),

$$E\left\|D^y_{k,t+1} - D^\star_y\right\|^2_{(I-W)^\dagger} = \left\|D^y_{k,t} - D^\star_y\right\|^2_{(I-W)^\dagger} + \frac{\gamma_y^2}{4s^2}\left\|\nu^y_{k,t+1} - H^\star_y\right\|^2_{I-W} + \frac{\gamma_y^2}{4s^2}E\left\|\hat{\nu}^y_{k,t+1} - \nu^y_{k,t+1}\right\|^2_{I-W}$$

$$+ \frac{\gamma_y}{s}\left\langle D^y_{k,t} - D^\star_y, y_{k,t} - \mathbf{1}y^\star + s\left(\mathcal{G}^y_{k,t} - \nabla_y F(\mathbf{1}z^\star)\right)\right\rangle - \gamma_y\left\|D^y_{k,t} - D^\star_y\right\|^2.$$
(89)

**Step 2: Computing** $\frac{2s^2}{\gamma_y}E\left\|D^y_{k,t+1} - D^\star_y\right\|^2_{(I-W)^\dagger} + \left\|\nu^y_{k,t+1} - H^\star_y\right\|^2_{I-\frac{\gamma_y}{2}(I-W)}$

Taking square norm on both sides of (88), we obtain .

$$\left\| \nu_{k,t+1}^y - H_y^\star \right\|^2 = \left\| y_{k,t} - \mathbf{1}y^\star + s\left(\mathcal{G}_{k,t}^y - \nabla_y F(\mathbf{1}z^\star)\right) \right\|^2 + s^2 \left\| D_{k,t}^y - D_y^\star \right\|^2$$
$$- 2s\left\langle D_{k,t}^y - D_y^\star, y_{k,t} - \mathbf{1}y^\star + s\mathcal{G}_{k,t}^y - s\nabla_y F(\mathbf{1}z^\star) \right\rangle. \tag{90}$$

Multiply (89) by $\frac{2s^2}{\gamma_y}$ and adding the resulting inequality with (90),

$$\frac{2s^2}{\gamma_y}E\left\| D_{k,t+1}^y - D_y^\star \right\|_{(I-W)^\dagger}^2 + \left\| \nu_{k,t+1}^y - H_y^\star \right\|^2$$

$$= \frac{2s^2}{\gamma_y}\left\| D_{k,t}^y - D_y^\star \right\|_{(I-W)^\dagger}^2 + \frac{\gamma_y}{2}\left\| \nu_{k,t+1}^y - H_y^\star \right\|_{I-W}^2 + \frac{\gamma_y}{2}E\left\| \hat{\nu}_{k,t+1}^y - \nu_{k,t+1}^y \right\|_{I-W}^2 - 2s^2\left\| D_{k,t}^y - D_y^\star \right\|^2$$

$$+ s^2\left\| D_{k,t}^y - D_y^\star \right\|^2 + \left\| y_{k,t} - \mathbf{1}y^\star + s\mathcal{G}_{k,t}^y - s\nabla_y F(\mathbf{1}z^\star) \right\|^2$$

$$= \frac{2s^2}{\gamma_y}\left\| D_{k,t}^y - D_y^\star \right\|_{(I-W)^\dagger}^2 + \frac{\gamma_y}{2}\left\| \nu_{k,t+1}^y - H_y^\star \right\|_{I-W}^2 + \frac{\gamma_y}{2}E\left\| \hat{\nu}_{k,t+1}^y - \nu_{k,t+1}^y \right\|_{I-W}^2 - s^2\left\| D_{k,t}^y - D_y^\star \right\|^2$$

$$+ \left\| y_{k,t} - \mathbf{1}y^\star + s\mathcal{G}_{k,t}^y - s\nabla_y F(\mathbf{1}z^\star) \right\|^2. \tag{91}$$

We have

$$\left\| \nu_{k,t+1}^y - H_y^\star \right\|^2 - \frac{\gamma_y}{2}\left\| \nu_{k,t+1}^y - H_y^\star \right\|_{I-W}^2$$

$$= (\nu_{k,t+1}^y - H_y^\star)^\top I(\nu_{k,t+1}^y - H_y^\star) - (\nu_{k,t+1}^y - H_y^\star)^\top \frac{\gamma_y}{2}(I-W)(\nu_{k,t+1}^y - H_y^\star)$$

$$= (\nu_{k,t+1}^y - H_y^\star)^\top \left( I - \frac{\gamma_y}{2}(I-W) \right)(\nu_{k,t+1}^y - H_y^\star)$$

$$= \left\| \nu_{k,t+1}^y - H_y^\star \right\|_{I-\frac{\gamma_y}{2}(I-W)}^2.$$

Therefore,

$$\frac{\gamma_y}{2}\left\| \nu_{k,t+1}^y - H_y^\star \right\|_{I-W}^2 = \left\| \nu_{k,t+1}^y - H_y^\star \right\|^2 - \left\| \nu_{k,t+1}^y - H_y^\star \right\|_{I-\frac{\gamma_y}{2}(I-W)}^2. \tag{92}$$

By substituting above equality in (91), we obtain

$$\frac{2s^2}{\gamma_y}E\left\| D_{k,t+1}^y - D_y^\star \right\|_{(I-W)^\dagger}^2 + \left\| \nu_{k,t+1}^y - H_y^\star \right\|^2 = \frac{2s^2}{\gamma_y}\left\| D_{k,t}^y - D_y^\star \right\|_{(I-W)^\dagger}^2 + \left\| \nu_{k,t+1}^y - H_y^\star \right\|^2$$

$$- \left\| \nu_{k,t+1}^y - H_y^\star \right\|_{I-\frac{\gamma_y}{2}(I-W)}^2 + \frac{\gamma_y}{2}E\left\| \hat{\nu}_{k,t+1}^y - \nu_{k,t+1}^y \right\|_{I-W}^2$$

$$- s^2\left\| D_{k,t}^y - D_y^\star \right\|^2 + \left\| y_{k,t} - \mathbf{1}y^\star + s\mathcal{G}_{k,t}^y - s\nabla_y F(\mathbf{1}z^\star) \right\|^2$$

Hence we get

$$\frac{2s^2}{\gamma_y}E\left\| D_{k,t+1}^y - D_y^\star \right\|_{(I-W)^\dagger}^2 + \left\| \nu_{k,t+1}^y - H_y^\star \right\|_{I-\frac{\gamma_y}{2}(I-W)}^2$$

$$= \frac{2s^2}{\gamma_y}\left\| D_{k,t}^y - D_y^\star \right\|_{(I-W)^\dagger}^2 + \frac{\gamma_y}{2}E\left\| \hat{\nu}_{k,t+1}^y - \nu_{k,t+1}^y \right\|_{I-W}^2 + \left\| y_{k,t} - \mathbf{1}y^\star + s\mathcal{G}_{k,t}^y - s\nabla_y F(\mathbf{1}z^\star) \right\|^2 - s^2\left\| D_{k,t}^y - D_y^\star \right\|^2. \tag{93}$$

Now we write $\frac{2s^2}{\gamma_y}\left\|D_{k,t}^y - D_y^\star\right\|_{(I-W)^\dagger}^2 - s^2\left\|D_{k,t}^y - D_y^\star\right\|^2$ in terms of $\frac{s^2}{\gamma_y}\left\|D_{k,t}^y - D_y^\star\right\|_{2(I-W)^\dagger-\gamma_y I}^2$.

$$\frac{2s^2}{\gamma_y}\left\|D_{k,t}^y - D_y^\star\right\|_{(I-W)^\dagger}^2 - s^2\left\|D_{k,t}^y - D_y^\star\right\|^2$$

$$= \frac{2s^2}{\gamma_y}(D_{k,t}^y - D_y^\star)^\top (I-W)^\dagger (D_{k,t}^y - D_y^\star) - s^2(D_{k,t}^y - D_y^\star)^\top(D_{k,t}^y - D_y^\star)$$

$$= \frac{s^2}{\gamma_y}\left((D_{k,t}^y - D_y^\star)^\top(2(I-W)^\dagger - \gamma_y I)(D_{k,t}^y - D_y^\star)\right)$$

$$= \frac{s^2}{\gamma_y}\left\|D_{k,t}^y - D_y^\star\right\|_{2(I-W)^\dagger - \gamma_y I}^2. \tag{94}$$

By substituting this equality in (93), we obtain

$$\frac{2s^2}{\gamma_y}E\left\|D_{k,t+1}^y - D_y^\star\right\|_{(I-W)^\dagger}^2 + \left\|\nu_{k,t+1}^y - H_y^\star\right\|_{I-\frac{\gamma_y}{2}(I-W)}^2$$

$$= \frac{s^2}{\gamma_y}\left\|D_{k,t}^y - D_y^\star\right\|_{2(I-W)^\dagger - \gamma_y I}^2 + \frac{\gamma_y}{2}E\left\|\hat{\nu}_{k,t+1}^y - \nu_{k,t+1}^y\right\|_{I-W}^2 + \left\|y_{k,t} - \mathbf{1}y^\star + s\mathcal{G}_{k,t}^y - s\nabla_y F(\mathbf{1}z^\star)\right\|^2. \tag{95}$$

**Step 3: Computing $\sqrt{\delta}E\left\|H_{k,t+1}^y - H_y^\star\right\|^2$ and finishing the proof**

Observe from Step 4 in Algorithm 4 that $H_{k,t+1}^y = (1-\alpha_y)H_{k,t}^y + \alpha_y\hat{\nu}_{k,t+1}^y$, and as a result,

$H_{k,t+1}^y - H_y^\star = (1-\alpha_y)(H_{k,t}^y - H_y^\star) + \alpha_y(\hat{\nu}_{k,t+1}^y - \nu_{k,t+1}^y) + \alpha_y(\nu_{k,t+1}^y - H_y^\star)$, and

$$\left\|H_{k,t+1}^y - H_y^\star\right\|^2 = \left\|(1-\alpha_y)(H_{k,t}^y - H_y^\star) + \alpha_y(\nu_{k,t+1}^y - H_y^\star)\right\|^2 + \alpha_y^2\left\|\hat{\nu}_{k,t+1}^y - \nu_{k,t+1}^y\right\|^2$$

$$+ 2\left\langle (1-\alpha_y)(H_{k,t}^y - H_y^\star) + \alpha_y(\nu_{k,t+1}^y - H_y^\star), \alpha_y(\hat{\nu}_{k,t+1}^y - \nu_{k,t+1}^y)\right\rangle. \tag{96}$$

Taking conditional expectation over compression at $t$-th iterate on both sides and substituting $E\left(\hat{\nu}_{k,t+1}^y - \nu_{k,t+1}^y\right) = 0$, we see that

$$E\left\|H_{k,t+1}^y - H_y^\star\right\|^2 = \left\|(1-\alpha_y)(H_{k,t}^y - H_y^\star) + \alpha_y(\nu_{k,t+1}^y - H_y^\star)\right\|^2 + \alpha_y^2 E\left\|\hat{\nu}_{k,t+1}^y - \nu_{k,t+1}^y\right\|^2$$

$$= (1-\alpha_y)\left\|H_{k,t}^y - H_y^\star\right\|^2 + \alpha_y\left\|\nu_{k,t+1}^y - H_y^\star\right\|^2 - \alpha_y(1-\alpha_{y,k})\left\|\nu_{k,t+1}^y - H_{k,t}^y\right\|^2$$

$$+ \alpha_y^2 E\left\|\hat{\nu}_{k,t+1}^y - \nu_{k,t+1}^y\right\|^2. \tag{97}$$

The last equality follows from the identity $\|(1-\alpha)x + \alpha y\|^2 = (1-\alpha)\|x\|^2 + \alpha\|y\|^2 - \alpha(1-\alpha)\|x-y\|^2$. On multiplying both sides of (97) by $\sqrt{\delta}$, we obtain

$$\sqrt{\delta}E\left\|H_{k,t+1}^y - H_y^\star\right\|^2 = \sqrt{\delta}(1-\alpha_y)\left\|H_{k,t}^y - H_y^\star\right\|^2 + \sqrt{\delta}\alpha_y\left\|\nu_{k,t+1}^y - H_y^\star\right\|^2 + \sqrt{\delta}\alpha_y^2 E\left\|\hat{\nu}_{k,t+1}^y - \nu_{k,t+1}^y\right\|^2$$

$$- \sqrt{\delta}\alpha_y(1-\alpha_y)\left\|\nu_{k,t+1}^y - H_{k,t}^y\right\|^2. \tag{98}$$

We know that

$$\left\|\nu_{k,t+1}^y - H_y^\star\right\|_{I-\frac{\gamma_y}{2}(I-W)-\alpha_y\sqrt{\delta}I}^2 = \left\|\nu_{k,t+1}^y - H_y^\star\right\|_{I-\frac{\gamma_y}{2}(I-W)}^2 - \left\|\nu_{k,t+1}^y - H_y^\star\right\|_{\alpha_y\sqrt{\delta}I}^2. \tag{99}$$

Therefore,

$$\frac{2s^2}{\gamma_y}E\left\|D^y_{k,t+1}-D^\star_y\right\|^2_{(I-W)^\dagger}+\left\|\nu^y_{k,t+1}-H^\star_y\right\|^2_{I-\frac{\gamma_y}{2}(I-W)-\alpha_y\sqrt{\delta}I}$$

$$=\frac{2s^2}{\gamma_y}E\left\|D^y_{k,t+1}-D^\star_y\right\|^2_{(I-W)^\dagger}+\left\|\nu^y_{k,t+1}-H^\star_y\right\|^2_{I-\frac{\gamma_y}{2}(I-W)}-\left\|\nu^y_{k,t+1}-H^\star_y\right\|^2_{\alpha_y\sqrt{\delta}I}$$

$$=\frac{2s^2}{\gamma_y}E\left\|D^y_{k,t+1}-D^\star_y\right\|^2_{(I-W)^\dagger}+\left\|\nu^y_{k,t+1}-H^\star_y\right\|^2_{I-\frac{\gamma_y}{2}(I-W)}-\alpha_y\sqrt{\delta}\left\|\nu^y_{k,t+1}-H^\star_y\right\|^2$$

$$=\frac{s^2}{\gamma_y}\left\|D^y_{k,t}-D^\star_y\right\|^2_{2(I-W)^\dagger-\gamma_yI}+\frac{\gamma_y}{2}E\left\|\hat{\nu}^y_{k,t+1}-\nu^y_{k,t+1}\right\|^2_{I-W}+\left\|y_{k,t}-\mathbf{1}y^\star+s\mathcal{G}^y_{k,t}-s\nabla_yF(\mathbf{1}z^\star)\right\|^2$$

$$-\alpha_y\sqrt{\delta}\left\|\nu^y_{k,t+1}-H^\star_y\right\|^2, \tag{100}$$

where the last equality follows from (95) . Now we add above equality with (98) and obtain the following expression:

$$\frac{2s^2}{\gamma_y}E\left\|D^y_{k,t+1}-D^\star_y\right\|^2_{(I-W)^\dagger}+\left\|\nu^y_{k,t+1}-H^\star_y\right\|^2_{I-\frac{\gamma_y}{2}(I-W)-\alpha_y\sqrt{\delta}I}+\sqrt{\delta}E\left\|H^y_{k,t+1}-H^\star_y\right\|^2$$

$$=\frac{s^2}{\gamma_y}\left\|D^y_{k,t}-D^\star_y\right\|^2_{2(I-W)^\dagger-\gamma_yI}+\frac{\gamma_y}{2}E\left\|\hat{\nu}^y_{k,t+1}-\nu^y_{k,t+1}\right\|^2_{I-W}+\left\|y_{k,t}-\mathbf{1}y^\star+s\mathcal{G}^y_{k,t}-s\nabla_yF(\mathbf{1}z^\star)\right\|^2$$

$$+\sqrt{\delta}(1-\alpha_y)\left\|H^y_{k,t}-H^\star_y\right\|^2+\sqrt{\delta}\alpha_y^2E\left\|\hat{\nu}^y_{k,t+1}-\nu^y_{k,t+1}\right\|^2-\sqrt{\delta}\alpha_y(1-\alpha_y)\left\|\nu^y_{k,t+1}-H^y_{k,t}\right\|^2. \tag{101}$$

Rearranging the r.h.s. of the above equation, we obtain (55).

We have a similar recursion result in terms of $x$.

**Lemma D.2.** *Let* $x_{k,t+1}$, $y_{k,t+1}$, $D^x_{k,t+1}$, $D^y_{k,t+1}$, $H^x_{k,t+1}$, $H^y_{k,t+1}$, $H^{w,x}_{k,t+1}$, $H^{w,y}_{k,t+1}$ *be obtained from Algorithm 1 using IPDHG(*$x_{k,t}$, $y_{k,t}$, $D^x_{k,t}$, $D^y_{k,t}$, $H^x_{k,t}$, $H^y_{k,t}$, $H^{w,x}_{k,t}$, $H^{w,y}_{k,t}$, $s$, $\gamma_x$, $\gamma_y$, $\alpha_x$, $\alpha_y$, $\mathcal{G}$*). Let Assumption 3.4 and Assumption 3.5 hold. Suppose* $\alpha_x\in\left(0,(1+\delta)^{-1}\right)$ *and*

$$\gamma_x\in\left(0,\min\left\{\frac{2-2\sqrt{\delta}\alpha_x}{\lambda_{\max}(I-W)},\frac{\alpha_x-(1+\delta)\alpha_x^2}{\sqrt{\delta}\lambda_{\max}(I-W)}\right\}\right). \tag{102}$$

*Then the following holds for all* $t\geq0$*:*

$$M_xE\left\|x_{k,t+1}-\mathbf{1}x^\star\right\|^2+\frac{2s^2}{\gamma_x}E\left\|D^x_{k,t+1}-D^\star_x\right\|^2_{(I-W)^\dagger}+\sqrt{\delta}E\left\|H^x_{k,t+1}-H^\star_x\right\|^2$$

$$\leq\left\|x_{k,t}-\mathbf{1}x^\star-s\mathcal{G}^x_{k,t}+s\nabla_xF(\mathbf{1}x^\star,\mathbf{1}y^\star)\right\|^2+\frac{2s^2}{\gamma_x}\left(1-\frac{\gamma_x}{2}\lambda_{m-1}(I-W)\right)\left\|D^x_{k,t}-D^\star_y\right\|^2_{(I-W)^\dagger}$$

$$+\sqrt{\delta}(1-\alpha_x)\left\|H^x_{k,t}-H^\star_x\right\|^2, \tag{103}$$

*where E denotes the conditional expectation on stochastic compression at t-th update step and* $M_x=1-\frac{\sqrt{\delta}\alpha_x}{1-\frac{\gamma_x}{2}\lambda_{\max}(I-W)}$*.*

We omit the proof of Lemma D.2 as it is similar to the proof of Lemma D.1

# E Proofs for General Stochastic Setting

In this section, we state and present proofs of the results for general stochastic setting discussed in Section 4.

We now make the following assumptions on the stochastic gradients.

**Assumption E.1. (Unbiasedness and local bounded variance of stochastic gradients)** Stochastic gradients $\nabla_xf_i(x,y;\xi^i)$ and $\nabla_yf_i(x,y;\xi^i)$ for each node $i$ are unbiased estimates of the respective true gradients: $E\left[\nabla_xf_i(x,y;\xi^i)\right]=\nabla_xf_i(x,y),E\left[\nabla_yf_i(x,y;\xi^i)\right]=\nabla_yf_i(x,y)$ and have bounded variance:

$$E\left[\left\|\nabla_xf_i(x^\star,y^\star;\xi^i)-\nabla_xf_i(x^\star,y^\star)\right\|^2\right]\leq\sigma_x^2,$$

$$E\left[\left\|\nabla_yf_i(x^\star,y^\star;\xi^i)-\nabla_yf_i(x^\star,y^\star)\right\|^2\right]\leq\sigma_y^2, \tag{104}$$

where $(x^\star,y^\star)$ is the saddle point of (SPP). Define $\sigma^2:=\sigma_x^2+\sigma_y^2$.

**Assumption E.2. (Smoothness)**

1. Each $f_i(\cdot, y; \xi^i)$ is $L_{xx}$ smooth in expectation; for all $x_1, x_2 \in \mathbb{R}^{d_x}$,
$$E\left\|\nabla_x f_i(x_1, y; \xi^i) - \nabla_x f_i(x_2, y; \xi^i)\right\|^2 \leq 2L_{xx}\left(f_i(x_1, y) - f_i(x_2, y) - \langle\nabla_x f_i(x_2, y), x_1 - x_2\rangle\right).$$

2. Each $-f_i(x, \cdot; \xi^i)$ is $L_{yy}$ smooth in expectation; for all $y_1, y_2 \in \mathbb{R}^{d_y}$,
$$E\left\|-\nabla_y f_i(x, y_1; \xi^i) + \nabla_y f_i(x, y_2; \xi^i)\right\|^2 \leq 2L_{yy}\left(-f_i(x, y_1) + f_i(x, y_2) - \langle-\nabla_y f_i(x, y_2), y_1 - y_2\rangle\right)$$

3. For every $x \in \mathbb{R}^{d_x}$, the following holds: $E\left\|\nabla_x f_i(x, y_1; \xi^i) - \nabla_x f_i(x, y_2; \xi^i)\right\|^2 \leq L_{xy}^2\left\|y_1 - y_2\right\|^2$ for all $y_2, y_2 \in \mathbb{R}^{d_y}$.

4. For every $y \in \mathbb{R}^{d_y}$, the following holds: $E\left\|\nabla_y f_i(x_1, y; \xi^i) - \nabla_y f_i(x_2, y; \xi^i)\right\|^2 \leq L_{yx}^2\left\|x_1 - x_2\right\|^2$ for all $x_1, x_2 \in \mathbb{R}^{d_x}$,

We define a quantity $\Phi_{k,t}$ consisting of primal and dual updates which is instrumental in deriving the convergence rate of Algorithm 2.

$$\Phi_{k,t} = M_{x,k}\left\|x_{k,t} - \mathbf{1}x^\star\right\|^2 + \frac{2s_k^2}{\gamma_{x,k}}\left\|D_{k,t}^x - D_x^\star\right\|_{(I-W)^\dagger}^2 + \sqrt{\delta}\left\|H_{k,t}^x - H_{x,k}^\star\right\|^2$$

$$+ M_{y,k}\left\|y_{k,t} - \mathbf{1}y^\star\right\|^2 + \frac{2s_k^2}{\gamma_{y,k}}\left\|D_{k,t}^y - D_y^\star\right\|_{(I-W)^\dagger}^2 + \sqrt{\delta}\left\|H_{k,t}^y - H_{y,k}^\star\right\|^2 \tag{105}$$

for all $t \geq 0$ and $k \geq 0$.

**Lemma E.3.** *Suppose $\{x_{k,t}\}_t$ and $\{y_{k,t}\}_t$ are the sequences generated by Algorithm 2 with $\mathcal{G} = [\nabla_x F(\cdot; \xi); -\nabla_y F(\cdot; \xi)]$. Then, under Assumption 3.1, Assumption 3.2, Assumption E.1 and Assumption E.2, the following hold for all $t \geq 0$:*

$$E\left\|x_{k,t} - \mathbf{1}x^\star - s_k\mathcal{G}_{k,t}^x + s_k\nabla_x F(\mathbf{1}x^\star, \mathbf{1}y^\star)\right\|^2$$

$$\leq (1 - \mu_x s_k)\left\|x_{k,t} - \mathbf{1}x^\star\right\|^2 + 4s_k^2 L_{xy}^2\left\|y_{k,t} - \mathbf{1}y^\star\right\|^2 - (2s_k - 8s_k^2 L_{xx})\sum_{i=1}^m V_{f_i, y_{k,t}^i}(x^\star, x_{k,t}^i)$$

$$+ 2s_k\left(F(x_{k,t}, \mathbf{1}y^\star) - F(\mathbf{1}x^\star, \mathbf{1}y^\star) + F(\mathbf{1}x^\star, y_{k,t}) - F(z_{k,t})\right) + 2ms_k^2\sigma_x^2, \tag{106}$$

$$E\left\|y_{k,t} - \mathbf{1}y^\star + s_k\mathcal{G}_{k,t}^y - s_k\nabla_y F(\mathbf{1}x^\star, \mathbf{1}y^\star)\right\|^2$$

$$\leq (1 - \mu_y s_k)\left\|y_{k,t} - \mathbf{1}y^\star\right\|^2 + 4s_k^2 L_{yx}^2\left\|x_{k,t} - \mathbf{1}x^\star\right\|^2 - (2s_k - 8s_k^2 L_{yy})\sum_{i=1}^m V_{-f_i, x_{k,t}^i}(y^\star, y_{k,t}^i)$$

$$+ 2s_k\left(-F(x_{k,t}, \mathbf{1}y^\star) + F(\mathbf{1}x^\star, \mathbf{1}y^\star) - F(\mathbf{1}x^\star, y_{k,t}) + F(z_{k,t})\right) + 2ms_k^2\sigma_y^2, \tag{107}$$

*where $E$ denotes the conditional expectation on stochastic gradient at $t$-th update step.*

### E.1 Proof of Lemma E.3

We will derive inequality (106) here. The proof of inequality (107) is similar and is omitted.
In the general stochastic setting, we have $\mathcal{G}_{k,t}^{i,x} = \nabla_x f_i(z_{k,t}^i; \xi_{k,t}^i)$, $\mathcal{G}_{k,t}^{i,y} = \nabla_y f_i(z_{k,t}^i; \xi_{k,t}^i)$ and step size is $s_k$. We now have

$$\begin{aligned} E\left\|x_{k,t} - \mathbf{1}x^\star - s_k\mathcal{G}_{k,t}^x + s_k\nabla_x F(\mathbf{1}x^\star, \mathbf{1}y^\star)\right\|^2 &= \|x_{k,t} - \mathbf{1}x^\star\|^2 + s_k^2 E\|\nabla_x F(\mathbf{1}x^\star, \mathbf{1}y^\star) - \nabla_x F(z_{k,t}; \xi_{k,t})\|^2 \\ &\quad + 2s_k E\langle x_{k,t} - \mathbf{1}x^\star, \nabla_x F(\mathbf{1}x^\star, \mathbf{1}y^\star) - \nabla_x F(z_{k,t}; \xi_{k,t})\rangle \\ &= \|x_{k,t} - \mathbf{1}x^\star\|^2 + s_k^2 E\|\nabla_x F(\mathbf{1}x^\star, \mathbf{1}y^\star) - \nabla_x F(z_{k,t}; \xi_{k,t})\|^2 \\ &\quad + 2s_k\langle x_{k,t} - \mathbf{1}x^\star, \nabla_x F(\mathbf{1}x^\star, \mathbf{1}y^\star) - \nabla_x F(z_{k,t})\rangle \\ &\leq \|x_{k,t} - \mathbf{1}x^\star\|^2 + 2s_k^2 E\|\nabla_x F(\mathbf{1}x^\star, \mathbf{1}y^\star) - \nabla_x F(\mathbf{1}x^\star, \mathbf{1}y^\star; \xi_{k,t})\|^2 \\ &\quad + 2s_k^2 E\|\nabla_x F(\mathbf{1}x^\star, \mathbf{1}y^\star; \xi_{k,t}) - \nabla_x F(z_{k,t}; \xi_{k,t})\|^2 \\ &\quad + 2s_k\langle x_{k,t} - \mathbf{1}x^\star, \nabla_x F(\mathbf{1}x^\star, \mathbf{1}y^\star) - \nabla_x F(z_{k,t})\rangle \\ &\leq \|x_{k,t} - \mathbf{1}x^\star\|^2 + 2s_k^2 E\|\nabla_x F(\mathbf{1}x^\star, \mathbf{1}y^\star; \xi_{k,t}) - \nabla_x F(z_{k,t}; \xi_{k,t})\|^2 \\ &\quad + 2s_k\langle x_{k,t} - \mathbf{1}x^\star, \nabla_x F(\mathbf{1}x^\star, \mathbf{1}y^\star) - \nabla_x F(z_{k,t})\rangle + 2ms_k^2\sigma_x^2. \end{aligned} \tag{108}$$

Also,
$$E\left\|\nabla_x F(\mathbf{1}x^\star, \mathbf{1}y^\star; \xi_{k,t}) - \nabla_x F(z_{k,t}; \xi_{k,t})\right\|^2$$

$$= \sum_{i=1}^m E\left\|\nabla_x f_i(z^\star; \xi_{k,t}^i) - \nabla_x f_i(z_{k,t}^i; \xi_{k,t}^i)\right\|^2$$

$$= \sum_{i=1}^m E\left\|\nabla_x f_i(z^\star; \xi_{k,t}^i) - \nabla_x f_i(x^\star, y_{k,t}^i; \xi_{k,t}^i) + \nabla_x f_i(x^\star, y_{k,t}^i; \xi_{k,t}^i) - \nabla_x f_i(z_{k,t}^i; \xi_{k,t}^i)\right\|^2$$

$$\leq \sum_{i=1}^m \left(2E\left\|\nabla_x f_i(z^\star; \xi_{k,t}^i) - \nabla_x f_i(x^\star, y_{k,t}^i; \xi_{k,t}^i)\right\|^2 + 2E\left\|\nabla_x f_i(x^\star, y_{k,t}^i; \xi_{k,t}^i) - \nabla_x f_i(z_{k,t}^i; \xi_{k,t}^i)\right\|^2\right)$$

$$\leq \sum_{i=1}^m \left(2L_{xy}^2\left\|y_{k,t}^i - y^\star\right\|^2 + 4L_{xx} V_{f_i, y_{k,t}^i}(x^\star, x_{k,t}^i)\right) \quad \text{(from Assumption E.2 and equation (27))}$$

$$= 2L_{xy}^2\left\|y_{k,t} - \mathbf{1}y^\star\right\|^2 + 4L_{xx}\sum_{i=1}^m V_{f_i, y_{k,t}^i}(x^\star, x_{k,t}^i). \tag{109}$$

We now simplify the inner product $\langle x_{k,t} - \mathbf{1}x^\star, \nabla_x F(\mathbf{1}x^\star, \mathbf{1}y^\star) - \nabla_x F(z_{k,t})\rangle$ term present in the r.h.s of (108). Recall the definition of Bregman distance $V_{f_i, y}(x_1, x_2)$:

$$V_{f_i, y_{k,t}^i}(x^\star, x_{k,t}^i) = f_i(x^\star, y_{k,t}^i) - f_i(x_{k,t}^i, y_{k,t}^i) - \left\langle\nabla_x f_i(x_{k,t}^i, y_{k,t}^i), x^\star - x_{k,t}^i\right\rangle \tag{110}$$

$$\left\langle\nabla_x f_i(z_{k,t}^i), -x^\star + x_{k,t}^i\right\rangle = -f_i(x^\star, y_{k,t}^i) + f_i(z_{k,t}^i) + V_{f_i, y_{k,t}^i}(x^\star, x_{k,t}^i). \tag{111}$$

Using $\mu_x$ strong convexity of $f_i(\cdot, y)$, we have

$$f_i(x_{k,t}^i, y^\star) \geq f_i(z^\star) + \left\langle\nabla_x f_i(z^\star), x_{k,t}^i - x^\star\right\rangle + \frac{\mu_x}{2}\left\|x_{k,t}^i - x^\star\right\|^2 \tag{112}$$

$$\left\langle\nabla_x f_i(z^\star), x_{k,t}^i - x^\star\right\rangle \leq f_i(x_{k,t}^i, y^\star) - f_i(z^\star) - \frac{\mu_x}{2}\left\|x_{k,t}^i - x^\star\right\|^2. \tag{113}$$

We now compute

$$\left\langle x_{k,t} - \mathbf{1}x^\star, \nabla_x F(\mathbf{1}x^\star, \mathbf{1}y^\star) - \nabla_x F(z_{k,t})\right\rangle = \sum_{i=1}^m \left\langle x_{k,t}^i - x^\star, \nabla_x f_i(z^\star) - \nabla_x f_i(z_{k,t}^i)\right\rangle$$

$$\leq \sum_{i=1}^m \left(f_i(x_{k,t}^i, y^\star) - f_i(z^\star) - \frac{\mu_x}{2}\left\|x_{k,t}^i - x^\star\right\|^2 + f_i(x^\star, y_{k,t}^i) - f_i(z_{k,t}^i) - V_{f_i, y_{k,t}^i}(x^\star, x_{k,t}^i)\right)$$

$$= F(x_{k,t}, \mathbf{1}y^\star) - F(\mathbf{1}x^\star, \mathbf{1}y^\star) + F(\mathbf{1}x^\star, y_{k,t}) - F(z_{k,t}) - \frac{\mu_x}{2}\left\|x_{k,t} - \mathbf{1}x^\star\right\|^2 \tag{114}$$

$$- \sum_{i=1}^m V_{f_i, y_{k,t}^i}(x^\star, x_{k,t}^i), \tag{115}$$

where the second last step follows from (111) and (113). On substituting (109) and (115) in (108), we obtain

$$E\left\|x_{k,t} - \mathbf{1}x^\star - s_k \mathcal{G}_{k,t}^x + s_k\nabla_x F(\mathbf{1}x^\star, \mathbf{1}y^\star)\right\|^2$$

$$\leq \left\|x_{k,t} - \mathbf{1}x^\star\right\|^2 + 2ms_k^2\sigma_x^2 + 2s_k^2\left(2L_{xy}^2\left\|y_{k,t} - \mathbf{1}y^\star\right\|^2 + 4L_{xx}\sum_{i=1}^m V_{f_i, y_{k,t}^i}(x^\star, x_{k,t}^i)\right)$$

$$+ 2s_k\left(F(x_{k,t}, \mathbf{1}y^\star) - F(\mathbf{1}x^\star, \mathbf{1}y^\star) + F(\mathbf{1}x^\star, y_{k,t}) - F(z_{k,t}) - \frac{\mu_x}{2}\left\|x_{k,t} - \mathbf{1}x^\star\right\|^2 - \sum_{i=1}^m V_{f_i, y_{k,t}^i}(x^\star, x_{k,t}^i)\right)$$

$$= (1 - \mu_x s_k)\left\|x_{k,t} - \mathbf{1}x^\star\right\|^2 + 4s_k^2 L_{xy}^2\left\|y_{k,t} - \mathbf{1}y^\star\right\|^2 - (2s_k - 8s_k^2 L_{xx})\sum_{i=1}^m V_{f_i, y_{k,t}^i}(x^\star, x_{k,t}^i)$$

$$+ 2s_k\left(F(x_{k,t}, \mathbf{1}y^\star) - F(\mathbf{1}x^\star, \mathbf{1}y^\star) + F(\mathbf{1}x^\star, y_{k,t}) - F(z_{k,t})\right) + 2ms_k^2\sigma_x^2, \tag{106}$$

completing the proof.
We have the following corollary.

**Corollary E.4.** *Let* $s_k = \frac{1}{4\kappa_f L 2^{k/2}}$ *for every* $k \geq 0$. *Then, under the setting of Lemma E.3 ,*

$$E \left\| x_{k,t} - \mathbf{1}x^\star - s_k \mathcal{G}_{k,t}^x + s_k \nabla_x F(\mathbf{1}x^\star, \mathbf{1}y^\star) \right\|^2 + E \left\| y_{k,t} - \mathbf{1}y^\star + s_k \mathcal{G}_{k,t}^y - s_k \nabla_y F(\mathbf{1}x^\star, \mathbf{1}y^\star) \right\|^2$$

$$\leq (1 - b_{x,k}) \left\| x_{k,t} - \mathbf{1}x^\star \right\|^2 + (1 - b_{y,k}) \left\| y_{k,t} - \mathbf{1}y^\star \right\|^2 + 2ms_k^2(\sigma_x^2 + \sigma_y^2)$$

*for all* $t \geq 1$, *where* $b_{x,k} = \mu_x s_k - 4s_k^2 L_{yx}^2 =$, $b_{y,k} = \mu_y s_k - 4s_k^2 L_{xy}^2$.

## E.2 Proof of Corollary E.4

As the step size $s_k = \frac{1}{4L\kappa_f 2^{k/2}}$, we have $s_k \leq \frac{1}{4L}$. We now show that the terms $2s_k - 8s_k^2 L_{xx}$ and $2s_k - 8s_k^2 L_{yy}$ appearing in (106) and (107) are non-negative:

$$\begin{aligned}
2s_k - 8s_k^2 L_{xx} &= \frac{2}{4L\kappa_f 2^{k/2}} - 8L_{xx}\frac{1}{16L^2\kappa_f^2 2^k} \\
&\geq \frac{1}{2L\kappa_f 2^{k/2}} - 8L\frac{1}{16L^2\kappa_f^2 2^k} = \frac{1}{2L\kappa_f 2^{k/2}} - \frac{1}{2L\kappa_f^2 2^k} \\
&= \frac{\kappa_f 2^{k/2} - 1}{2L\kappa_f^2 2^k} \geq 0.
\end{aligned} \tag{116}$$

Similarly, we get $2s_k - 8s_k^2 L_{yy} \geq 0$. Recall

$$\begin{aligned}
b_{x,k} &= \mu_x s_k - 4s_k^2 L_{yx}^2 \\
&= \frac{\mu_x}{4L\kappa_f 2^{k/2}} - \frac{4L_{yx}^2}{16L^2\kappa_f^2 2^k} \\
&\geq \frac{\mu}{4L\kappa_f 2^{k/2}} - \frac{4L^2}{16L^2\kappa_f^2 2^k} \\
&= \frac{1}{4\kappa_f^2 2^{k/2}} - \frac{1}{4\kappa_f^2 2^k} \\
&\geq \frac{1}{4\kappa_f^2 2^{k/2}} - \frac{1}{4\kappa_f^2 \sqrt{2} 2^{k/2}} \\
&= \left(1 - \frac{1}{\sqrt{2}}\right)\frac{1}{4\kappa_f^2 2^{k/2}}.
\end{aligned} \tag{117}$$

We now show that $b_{x,k} < 1$.

$$b_{x,k} < \mu_x s_k = \frac{\mu_x}{4L\kappa_f 2^{k/2}} \leq \frac{\mu_x}{4L_{xx}\kappa_f 2^{k/2}} = \frac{1}{4\kappa_x\kappa_f 2^{k/2}} < 1. \tag{118}$$

Therefore, $b_{x,k} \in (0, 1)$ for every $k \geq 0$. In a similar fashion, we obtain $b_{y,k} \in (0, 1)$ for every $k \geq 0$. On adding (106) and (107), we obtain

$$E \left\| x_{k,t} - \mathbf{1}x^\star - s_k \mathcal{G}_{k,t}^x + s_k \nabla_x F(\mathbf{1}z^\star) \right\|^2 + E \left\| y_{k,t} - \mathbf{1}y^\star + s_k \mathcal{G}_{k,t}^y - s_k \nabla_y F(\mathbf{1}z^\star) \right\|^2$$

$$\leq (1 - \mu_x s_k + 4s_k^2 L_{yx}^2) \left\| x_{k,t} - \mathbf{1}x^\star \right\|^2 + (1 - \mu_y s_k + 4s_k^2 L_{xy}^2) \left\| y_{k,t} - \mathbf{1}y^\star \right\|^2$$

$$- (2s_k - 8s_k^2 L_{xx}) \sum_{i=1}^m V_{f_i, y_{k,t}^i}(x^\star, x_{k,t}^i) - (2s_k - 8s_k^2 L_{yy}) \sum_{i=1}^m V_{-f_i, x_{k,t}^i}(y^\star, y_{k,t}^i) + 2ms_k^2(\sigma_x^2 + \sigma_y^2)$$

$$\leq (1 - b_{x,k}) \left\| x_{k,t} - \mathbf{1}x^\star \right\|^2 + (1 - b_{y,k}) \left\| y_{k,t} - \mathbf{1}y^\star \right\|^2 + 2ms_k^2(\sigma_x^2 + \sigma_y^2). \tag{119}$$

The last inequality follows from non-negativity of $V_{f_i, y_{k,t}^i}(x^\star, x_{k,t}^i), V_{-f_i, x_{k,t}^i}(y^\star, y_{k,t}^i), 2s_k - 8s_k^2 L_{xx}$ and $2s_k - 8s_k^2 L_{yy}$.

We now establish a recursion for $E[\Phi_{k,t}]$.

**Lemma E.5.** *Suppose* $\{x_{k,t}\}_t$ *and* $\{y_{k,t}\}_t$ *are the sequences generated by algorithm 2 with* $\mathcal{G} = [\nabla_x F(\cdot; \xi); -\nabla_y F(\cdot; \xi)]$. *Suppose Assumptions 3.1-3.5 and Assumptions E.1-E.2 hold. Let step size* $s_k$ *is chosen according to Corollary E.4. Then, for every* $k \geq 0$, *the following holds in total expectation:*

$$E[\Phi_{k,t+1}] \leq \rho_k E[\Phi_{k,t}] + 2ms_k^2(\sigma_x^2 + \sigma_y^2), \tag{120}$$

*for all* $t \geq 0$, *where* $\rho_k$ *is defined in equation (128).*

### E.3 Proof of Lemma E.5

On adding (53) and (103) , we have

$$M_{x,k} E \left\| x_{k,t+1} - \mathbf{1}x^\star \right\|^2 + \frac{2s_k^2}{\gamma_{x,k}} E \left\| D_{k,t+1}^x - D_x^\star \right\|_{(I-W)^\dagger}^2 + \sqrt{\delta} E \left\| H_{k,t+1}^x - H_{x,k}^\star \right\|^2$$

$$+ M_{y,k} E \left\| y_{k,t+1} - \mathbf{1}y^\star \right\|^2 + \frac{2s_k^2}{\gamma_{y,k}} E \left\| D_{k,t+1}^y - D_y^\star \right\|_{(I-W)^\dagger}^2 + \sqrt{\delta} E \left\| H_{k,t+1}^y - H_{y,k}^\star \right\|^2$$

$$\leq \left\| x_{k,t} - \mathbf{1}x^\star - s_k \mathcal{G}_{k,t}^x + s_k \nabla_x F(\mathbf{1}z^\star) \right\|^2 + \frac{2s_k^2}{\gamma_{x,k}} \left( 1 - \frac{\gamma_{x,k}}{2} \lambda_{m-1}(I-W) \right) \left\| D_{k,t}^x - D_y^\star \right\|_{(I-W)^\dagger}^2$$

$$+ \sqrt{\delta}(1 - \alpha_{x,k}) \left\| H_{k,t}^x - H_{x,k}^\star \right\|^2 + \left\| y_{k,t} - \mathbf{1}y^\star + s_k \mathcal{G}_{k,t}^y - s_k \nabla_y F(\mathbf{1}z^\star) \right\|^2$$

$$+ \frac{2s_k^2}{\gamma_{y,k}} \left( 1 - \frac{\gamma_{y,k}}{2} \lambda_{m-1}(I-W) \right) \left\| D_{k,t}^y - D_y^\star \right\|_{(I-W)^\dagger}^2 + \sqrt{\delta}(1 - \alpha_{y,k}) \left\| H_{k,t}^y - H_{y,k}^\star \right\|^2 . \tag{121}$$

By taking conditional expectation on stochastic gradient at $t$-th step on both sides of above inequality and applying Tower property, we obtain

$$M_{x,k} E \left\| x_{k,t+1} - \mathbf{1}x^\star \right\|^2 + \frac{2s_k^2}{\gamma_{x,k}} E \left\| D_{k,t+1}^x - D_x^\star \right\|_{(I-W)^\dagger}^2 + \sqrt{\delta} E \left\| H_{k,t+1}^x - H_{x,k}^\star \right\|^2$$

$$+ M_{y,k} E \left\| y_{k,t+1} - \mathbf{1}y^\star \right\|^2 + \frac{2s_k^2}{\gamma_{y,k}} E \left\| D_{k,t+1}^y - D_y^\star \right\|_{(I-W)^\dagger}^2 + \sqrt{\delta} E \left\| H_{k,t+1}^y - H_{y,k}^\star \right\|^2$$

$$\leq E \left\| x_{k,t} - \mathbf{1}x^\star - s_k \mathcal{G}_{k,t}^x + s_k \nabla_x F(\mathbf{1}x^\star), \mathbf{1}y^\star) \right\|^2 + \frac{2s_k^2}{\gamma_{x,k}} \left( 1 - \frac{\gamma_{x,k}}{2} \lambda_{m-1}(I-W) \right) \left\| D_{k,t}^x - D_x^\star \right\|_{(I-W)^\dagger}^2$$

$$+ \sqrt{\delta}(1 - \alpha_{x,k}) \left\| H_{k,t}^x - H_{x,k}^\star \right\|^2 + E \left\| y_{k,t} - \mathbf{1}y^\star + s_k \mathcal{G}_{k,t}^y - s_k \nabla_y F(\mathbf{1}x^\star, \mathbf{1}y^\star)) \right\|^2$$

$$+ \frac{2s_k^2}{\gamma_{y,k}} \left( 1 - \frac{\gamma_{y,k}}{2} \lambda_{m-1}(I-W) \right) \left\| D_{k,t}^y - D_y^\star \right\|_{(I-W)^\dagger}^2 + \sqrt{\delta}(1 - \alpha_{y,k}) \left\| H_{k,t}^y - H_{y,k}^\star \right\|^2$$

$$\leq (1 - b_{x,k}) \left\| x_{k,t} - \mathbf{1}x^\star \right\|^2 + (1 - b_{y,k}) \left\| y_{k,t} - \mathbf{1}y^\star \right\|^2$$

$$+ \frac{2s_k^2}{\gamma_{x,k}} \left( 1 - \frac{\gamma_{x,k}}{2} \lambda_{m-1}(I-W) \right) \left\| D_{k,t}^x - D_x^\star \right\|_{(I-W)^\dagger}^2 + \frac{2s_k^2}{\gamma_{y,k}} \left( 1 - \frac{\gamma_{y,k}}{2} \lambda_{m-1}(I-W) \right) \left\| D_{k,t}^y - D_y^\star \right\|_{(I-W)^\dagger}^2$$

$$+ \sqrt{\delta}(1 - \alpha_{x,k}) \left\| H_{k,t}^x - H_{x,k}^\star \right\|^2 + \sqrt{\delta}(1 - \alpha_{y,k}) \left\| H_{k,t}^y - H_{y,k}^\star \right\|^2 + 2ms_k^2(\sigma_x^2 + \sigma_y^2) \tag{122}$$

where the last inequality follows from inequality (119).

By taking total expectation on both sides of above inequality and using the definition of $\Phi_{k,t}$, we obtain

$$E\left[\Phi_{k,t+1}\right]$$

$$\leq (1 - b_{x,k})E\left\|x_{k,t} - \mathbf{1}x^\star\right\|^2 + (1 - b_{y,k})E\left\|y_{k,t} - \mathbf{1}y^\star\right\|^2$$

$$+ \frac{2s_k^2}{\gamma_{x,k}}\left(1 - \frac{\gamma_{x,k}}{2}\lambda_{m-1}(I - W)\right)E\left\|D_{k,t}^x - D_x^\star\right\|_{(I-W)^\dagger}^2 + \frac{2s_k^2}{\gamma_{y,k}}\left(1 - \frac{\gamma_{y,k}}{2}\lambda_{m-1}(I - W)\right)E\left\|D_{k,t}^y - D_y^\star\right\|_{(I-W)^\dagger}^2$$

$$+ \sqrt{\delta}(1 - \alpha_{x,k})E\left\|H_{k,t}^x - H_{x,k}^\star\right\|^2 + \sqrt{\delta}(1 - \alpha_{y,k})E\left\|H_{k,t}^y - H_{y,k}^\star\right\|^2 + 2ms_k^2(\sigma_x^2 + \sigma_y^2)$$

$$= \frac{(1 - b_{x,k})}{M_{x,k}}M_{x,k}E\left\|x_{k,t} - \mathbf{1}x^\star\right\|^2 + \frac{(1 - b_{y,k})}{M_{y,k}}M_{y,k}E\left\|y_{k,t} - \mathbf{1}y^\star\right\|^2$$

$$+ \frac{2s_k^2}{\gamma_{x,k}}\left(1 - \frac{\gamma_{x,k}}{2}\lambda_{m-1}(I - W)\right)E\left\|D_{k,t}^x - D_x^\star\right\|_{(I-W)^\dagger}^2 + \frac{2s_k^2}{\gamma_{y,k}}\left(1 - \frac{\gamma_{y,k}}{2}\lambda_{m-1}(I - W)\right)E\left\|D_{k,t}^y - D_y^\star\right\|_{(I-W)^\dagger}^2$$

$$+ \sqrt{\delta}(1 - \alpha_{x,k})E\left\|H_{k,t}^x - H_{x,k}^\star\right\|^2 + \sqrt{\delta}(1 - \alpha_{y,k})E\left\|H_{k,t}^y - H_{y,k}^\star\right\|^2 + 2ms_k^2(\sigma_x^2 + \sigma_y^2)$$

$$\leq \max\left\{\frac{1 - b_{x,k}}{M_{x,k}}, \frac{1 - b_{y,k}}{M_{y,k}}, 1 - \frac{\gamma_{x,k}}{2}\lambda_{m-1}(I - W), 1 - \frac{\gamma_{y,k}}{2}\lambda_{m-1}(I - W), 1 - \alpha_{x,k}, 1 - \alpha_{y,k}\right\} \times E\left[\Phi_{k,t}\right]$$

$$+ 2ms_k^2(\sigma_x^2 + \sigma_y^2)$$

$$= \rho_k E\left[\Phi_{k,t}\right] + 2ms_k^2(\sigma_x^2 + \sigma_y^2), \tag{123}$$

where

$$\rho_k := \max\left\{\frac{1 - b_{x,k}}{M_{x,k}}, \frac{1 - b_{y,k}}{M_{y,k}}, 1 - \frac{\gamma_{x,k}}{2}\lambda_{m-1}(I - W), 1 - \frac{\gamma_{y,k}}{2}\lambda_{m-1}(I - W), 1 - \alpha_{x,k}, 1 - \alpha_{y,k}\right\}$$

$$\tag{124}$$

### E.4 Feasibility of Parameters for Algorithm 2

**Parameters setting:** From Corollary E.4, the step size used in Algorithm 2 is $s_k = \frac{1}{4\kappa_f L 2^{k/2}}$ for every $k \geq 0$. We choose the parameters involved in **COMM** procedure and other parameters $\gamma_{x,k}, \gamma_{y,k}$ as follows:

$$\alpha_{x,k} = \frac{b_{x,k}}{1 + \delta}, \ \alpha_{y,k} = \frac{b_{y,k}}{1 + \delta} \tag{125}$$

$$\gamma_{x,k} = \frac{b_{x,k}}{2(1 + \delta)^2 \lambda_{\max}(I - W)}, \ \gamma_{y,k} = \frac{b_{y,k}}{2(1 + \delta)^2 \lambda_{\max}(I - W)} \tag{126}$$

$$M_{x,k} = 1 - \frac{\sqrt{\delta}\alpha_{x,k}}{1 - \frac{\gamma_{x,k}}{2}\lambda_{\max}(I - W)}, \ M_{y,k} = 1 - \frac{\sqrt{\delta}\alpha_{y,k}}{1 - \frac{\gamma_{y,k}}{2}\lambda_{\max}(I - W)} \tag{127}$$

$$\rho_k = \max\left\{\frac{1 - b_{x,k}}{M_{x,k}}, \frac{1 - b_{y,k}}{M_{y,k}}, 1 - \frac{\gamma_{x,k}}{2}\lambda_{m-1}(I - W), 1 - \frac{\gamma_{y,k}}{2}\lambda_{m-1}(I - W), 1 - \alpha_{x,k}, 1 - \alpha_{y,k}\right\}$$

$$\tag{128}$$

$$\rho = \min\left\{\left(1 - \frac{1}{\sqrt{2}}\right)\frac{1}{8\kappa_f^2}, \left(1 - \frac{1}{\sqrt{2}}\right)\frac{1}{16(1 + \delta)^2}\frac{1}{\kappa_f^2 \kappa_g}, \frac{1}{1 + \delta}\left(1 - \frac{1}{\sqrt{2}}\right)\frac{1}{4\kappa_f^2}\right\}. \tag{129}$$

**Feasibility of parameters:** Above choice of parameters should satisfy the following conditions:

$$\alpha_{x,k} < \min\left\{\frac{b_{x,k}}{\sqrt{\delta}}, \frac{1}{1+\delta}\right\}, \ \alpha_{y,k} < \min\left\{\frac{b_{y,k}}{\sqrt{\delta}}, \frac{1}{1+\delta}\right\} \tag{130}$$

$$\gamma_{x,k} \in \left(0, \min\left\{\frac{2 - 2\sqrt{\delta}\alpha_{x,k}}{\lambda_{\max}(I - W)}, \frac{\alpha_{x,k} - (1+\delta)\alpha_{x,k}^2}{\sqrt{\delta}\lambda_{\max}(I - W)}\right\}\right), \tag{131}$$

$$\gamma_{y,k} \in \left(0, \min\left\{\frac{2 - 2\sqrt{\delta}\alpha_{y,k}}{\lambda_{\max}(I - W)}, \frac{\alpha_{y,k} - (1+\delta)\alpha_{y,k}^2}{\sqrt{\delta}\lambda_{\max}(I - W)}\right\}\right), \tag{132}$$

$$\frac{\gamma_{x,k}}{2}\lambda_{m-1}(I - W) \in (0,1), \ \frac{\gamma_{y,k}}{2}\lambda_{m-1}(I - W) \in (0,1), \tag{133}$$

$$M_{x,k} \in (0,1), \ M_{y,k} \in (0,1), \tag{134}$$

$$\frac{1 - b_{x,k}}{M_{x,k}} \in (0,1), \ \frac{1 - b_{y,k}}{M_{y,k}} \in (0,1). \tag{135}$$

In this section, we show that all parameters specified in (126) satisfy all requirements of (130)-(135).

**Feasibility of $\alpha_{x,k}$ and $\alpha_{y,k}$.** From (117) and (118), we have $0 < b_{x,k} < 1$. Therefore, $\alpha_{x,k} < \frac{1}{1+\delta}$. Moreover, $\frac{\sqrt{\delta}}{1+\delta} \leq 1/2$. Therefore, $\alpha_{x,k} \leq \frac{b_{x,k}}{2\sqrt{\delta}} < b_{x,k}/\sqrt{\delta}$. Hence, $\alpha_{x,k} < \min\left\{\frac{b_{x,k}}{\sqrt{\delta}}, \frac{1}{1+\delta}\right\}$. Similarly, $\alpha_{y,k} < \min\left\{\frac{b_{y,k}}{\sqrt{\delta}}, \frac{1}{1+\delta}\right\}$ because $b_{y,k} \in (0,1)$.

Consider

$$\frac{\alpha_{x,k} - (1+\delta)\alpha_{x,k}^2}{\sqrt{\delta}\lambda_{\max}(I - W)} = \frac{b_{x,k} - b_{x,k}^2}{\sqrt{\delta}(1+\delta)\lambda_{\max}(I - W)}$$

$$\geq \frac{2(b_{x,k} - b_{x,k}^2)}{(1+\delta)(1+\delta)\lambda_{\max}(I - W)}. \tag{136}$$

The last inequality uses the relation $\frac{\sqrt{\delta}}{1+\delta} \leq \frac{1}{2}$. Using (118), we have $b_{x,k} \leq \frac{1}{4\kappa_x \kappa_f 2^{k/2}} < 0.25$. This allows us to use the inequality $2x - 2x^2 \geq x/2$ for all $0 \leq x \leq 0.75$. Therefore,

$$\frac{\alpha_{x,k} - (1+\delta)\alpha_{x,k}^2}{\sqrt{\delta}\lambda_{\max}(I - W)} > \frac{b_{x,k}}{2(1+\delta)^2\lambda_{\max}(I - W)}$$

$$= \gamma_{x,k}. \tag{137}$$

Consider

$$\frac{2 - 2\sqrt{\delta}\alpha_{x,k}}{\lambda_{\max}(I - W)} = \left(2 - \frac{2\sqrt{\delta}b_{x,k}}{1+\delta}\right)\frac{1}{\lambda_{\max}(I - W)}$$

$$\geq \left(2 - \frac{2\sqrt{\delta}}{1+\delta}\right)\frac{1}{\lambda_{\max}(I - W)}$$

$$\geq \frac{1}{\lambda_{\max}(I - W)}$$

$$> \frac{b_{x,k}}{2(1+\delta)^2\lambda_{\max}(I - W)}$$

$$= \gamma_{x,k}, \tag{138}$$

where the third inequality uses $\frac{\sqrt{\delta}}{1+\delta} \leq \frac{1}{2}$ and fourth inequality uses $\frac{b_{x,k}}{2(1+\delta)^2} < 1$. We know that $b_{y,k} \in (0,1)$. Therefore, by following similar steps, the chosen $\gamma_{y,k}$ is also feasible. As $\gamma_{x,k} < \frac{2 - 2\sqrt{\delta}\alpha_{x,k}}{\lambda_{\max}(I - W)} < \frac{2}{\lambda_{\max}(I - W)}$. Notice that $\lambda_{m-1}(I - W) < \lambda_{\max}(I - W)$ Therefore,

$$\frac{\gamma_{x,k}}{2}\lambda_{m-1}(I - W) < \frac{\gamma_{x,k}}{2}\lambda_{\max}(I - W) < 1. \tag{139}$$

Similarly, $\frac{\gamma_{y,k}}{2}\lambda_{m-1}(I - W) < 1$.

**Feasibility of $M_{x,k}$ and $M_{y,k}$.** Recall $M_{x,k} = 1 - \frac{\sqrt{\delta}\alpha_{x,k}}{1-\frac{\gamma_{x,k}}{2}\lambda_{\max}(I-W)}$ and $M_{y,k} = 1 - \frac{\sqrt{\delta}\alpha_{y,k}}{1-\frac{\gamma_{y,k}}{2}\lambda_{\max}(I-W)}$. We have

$$\gamma_{x,k} < \frac{2 - 2\sqrt{\delta}\alpha_{x,k}}{\lambda_{\max}(I - W)}$$

$$\Rightarrow \frac{\gamma_{x,k}\lambda_{\max}(I - W)}{2} < 1 - \sqrt{\delta}\alpha_{x,k}$$

$$\Rightarrow 1 - \frac{\gamma_{x,k}\lambda_{\max}(I - W)}{2} > \sqrt{\delta}\alpha_{x,k}$$

$$\Rightarrow \frac{\sqrt{\delta}\alpha_{x,k}}{1 - \frac{\gamma_{x,k}\lambda_{\max}(I-W)}{2}} < 1. \tag{140}$$

Moreover, $\frac{\sqrt{\delta}\alpha_{x,k}}{1-\frac{\gamma_{x,k}\lambda_{\max}(I-W)}{2}} > 0$. Therefore, $M_{x,k} \in (0,1)$. The feasibility of $M_{y,k}$ can be proved similarly.

**Feasibility of $\frac{1-b_{x,k}}{M_{x,k}}$ and $\frac{1-b_{y,k}}{M_{y,k}}$.** Observe that

$$\frac{1 - b_{x,k}}{M_{x,k}} = \frac{1 - b_{x,k}}{1 - \frac{\sqrt{\delta}\alpha_{x,k}}{1-\frac{\gamma_{x,k}}{2}\lambda_{\max}(I-W)}}$$

$$= \frac{(1 - b_{x,k})\left(1 - \frac{\gamma_{x,k}}{2}\lambda_{\max}(I - W)\right)}{1 - \frac{\gamma_{x,k}}{2}\lambda_{\max}(I - W) - \sqrt{\delta}\alpha_{x,k}}$$

$$= \frac{(1 - b_{x,k})\left(1 - \frac{b_{x,k}}{4(1+\delta)^2}\right)}{1 - \frac{b_{x,k}}{4(1+\delta)^2} - \frac{\sqrt{\delta}b_{x,k}}{1+\delta}}. \tag{141}$$

Let $\theta = 1 - b_{x,k}$, $\sigma = 1 - \frac{b_{x,k}}{4(1+\delta)^2}$ and $\chi = \frac{\sqrt{\delta}b_{x,k}}{1+\delta}$. We now show that $\chi \leq b - a$.

$$\chi = \frac{\sqrt{\delta}b_{x,k}}{1 + \delta}$$

$$\leq \frac{b_{x,k}}{2}$$

$$= \frac{b_{x,k}}{2} + b_{x,k}\left(1 - \frac{1}{4(1 + \delta)^2}\right) - b_{x,k}\left(1 - \frac{1}{4(1 + \delta)^2}\right)$$

$$= b_{x,k}\left(1 - \frac{1}{4(1 + \delta)^2}\right) + \frac{b_{x,k}}{2} - b_{x,k} + \frac{b_{x,k}}{4(1 + \delta)^2}$$

$$\leq b_{x,k}\left(1 - \frac{1}{4(1 + \delta)^2}\right) + \frac{b_{x,k}}{2} - b_{x,k} + \frac{b_{x,k}}{2}$$

$$= b_{x,k}\left(1 - \frac{1}{4(1 + \delta)^2}\right)$$

$$= \sigma - \theta. \tag{142}$$

Notice that $0 \leq \chi \leq \sigma - \theta$, and $\sigma\theta \leq (\theta + \chi)(\sigma - \chi)$, i.e.,

$$(1 - b_{x,k})\left(1 - \frac{b_{x,k}}{4(1 + \delta)^2}\right) \leq \left(1 - b_{x,k} + \frac{\sqrt{\delta}b_{x,k}}{1 + \delta}\right)\left(1 - \frac{b_{x,k}}{4(1 + \delta)^2} - \frac{\sqrt{\delta}b_{x,k}}{1 + \delta}\right). \tag{143}$$

Substituting the above inequality in (141), we obtain

$$
\frac{1 - b_{x,k}}{M_{x,k}} \le \frac{\left(1 - b_{x,k} + \frac{\sqrt{\delta}b_{x,k}}{1+\delta}\right)\left(1 - \frac{b_{x,k}}{4(1+\delta)^2} - \frac{\sqrt{\delta}b_{x,k}}{1+\delta}\right)}{1 - \frac{b_{x,k}}{4(1+\delta)^2} - \frac{\sqrt{\delta}b_{x,k}}{1+\delta}}
$$

$$
= 1 - b_{x,k} + \frac{\sqrt{\delta}b_{x,k}}{1+\delta}
$$

$$
= 1 - \left(1 - \frac{\sqrt{\delta}}{1+\delta}\right)b_{x,k}
$$

$$
\le 1 - \frac{b_{x,k}}{2}
$$

$$
\le 1 - \left(1 - \frac{1}{\sqrt{2}}\right)\frac{1}{8\kappa_f^2}\frac{1}{2^{k/2}}, \tag{144}
$$

where the last step follows from (117) . Notice that $\left(1 - \frac{1}{\sqrt{2}}\right)\frac{1}{8\kappa_f^2}\frac{1}{2^{k/2}} \in (0,1)$ and hence $\frac{1-b_{x,k}}{M_{x,k}} \in (0,1)$. Similarly, we obtain

$$
\frac{1 - b_{y,k}}{M_{y,k}} \le 1 - \left(1 - \frac{1}{\sqrt{2}}\right)\frac{1}{8\kappa_f^2}\frac{1}{2^{k/2}}. \tag{145}
$$

We now prove another intermediate result that shows the convergence behavior of $E\left[\Phi_{k,t+1}\right]$.

**Lemma E.6.** *Let* $\Phi_{k,t+1}$ *and* $\rho$ *be as defined in* (129). *Under the settings of Lemma E.5,*

$$
E\left[\Phi_{k,t+1}\right] \le \left(1 - \frac{\rho}{2^{k/2}}\right)^{t+1} E\left[\Phi_{k,0}\right] + \frac{m(\sigma_x^2 + \sigma_y^2)}{8\rho L^2 \kappa_f^2}\frac{1}{2^{k/2}}, \tag{146}
$$

*for all* $t \ge 0$.

### E.5   Proof of Lemma E.6

First, we show that $\rho_k \le 1 - \frac{\rho}{2^{k/2}}$, where these quantities are defined in (129). To prove this relation, we first simplify the terms $1 - \frac{\gamma_{x,k}}{2}\lambda_{m-1}(I - W)$ and $1 - \alpha_{x,k}$ appearing in the definition of $\rho_k$:

$$
1 - \frac{\gamma_{x,k}}{2}\lambda_{m-1}(I - W) = 1 - \frac{b_{x,k}}{4(1+\delta)^2}\frac{\lambda_{m-1}(I - W)}{\lambda_{\max}(I - W)}
$$

$$
= 1 - \frac{b_{x,k}}{4(1+\delta)^2}\frac{1}{\kappa_g}
$$

$$
\le 1 - \frac{1}{4(1+\delta)^2}\frac{1}{\kappa_g}\left(1 - \frac{1}{\sqrt{2}}\right)\frac{1}{4\kappa_f^2}\frac{1}{2^{k/2}}
$$

$$
= 1 - \left(1 - \frac{1}{\sqrt{2}}\right)\frac{1}{16(1+\delta)^2}\frac{1}{\kappa_f^2\kappa_g}\frac{1}{2^{k/2}}. \tag{147}
$$

Similarly, we obtain

$$
1 - \frac{\gamma_{y,k}}{2}\lambda_{m-1}(I - W) \le 1 - \left(1 - \frac{1}{\sqrt{2}}\right)\frac{1}{16(1+\delta)^2}\frac{1}{\kappa_f^2\kappa_g}\frac{1}{2^{k/2}}. \tag{148}
$$

Consider

$$
1 - \alpha_{x,k} = 1 - \frac{b_{x,k}}{1+\delta} \le 1 - \frac{1}{1+\delta}\left(1 - \frac{1}{\sqrt{2}}\right)\frac{1}{4\kappa_f^2}\frac{1}{2^{k/2}} \tag{149}
$$

$$
1 - \alpha_{y,k} = 1 - \frac{b_{y,k}}{1+\delta} \le 1 - \frac{1}{1+\delta}\left(1 - \frac{1}{\sqrt{2}}\right)\frac{1}{4\kappa_f^2}\frac{1}{2^{k/2}}. \tag{150}
$$

Let us recall $\rho_k$:

$$\rho_k = \max\left\{\frac{1-b_{x,k}}{M_{x,k}}, \frac{1-b_{y,k}}{M_{y,k}}, 1-\frac{\gamma_{x,k}}{2}\lambda_{m-1}(I-W), 1-\frac{\gamma_{y,k}}{2}\lambda_{m-1}(I-W), 1-\alpha_{x,k}, 1-\alpha_{y,k}\right\}$$

$$\leq \max\left\{1-\left(1-\frac{1}{\sqrt{2}}\right)\frac{1}{8\kappa_f^2}\frac{1}{2^{k/2}}, 1-\left(1-\frac{1}{\sqrt{2}}\right)\frac{1}{16(1+\delta)^2}\frac{1}{\kappa_f^2\kappa_g}\frac{1}{2^{k/2}}, 1-\frac{1}{1+\delta}\left(1-\frac{1}{\sqrt{2}}\right)\frac{1}{4\kappa_f^2}\frac{1}{2^{k/2}}\right\}$$

$$= 1-\min\left\{\left(1-\frac{1}{\sqrt{2}}\right)\frac{1}{8\kappa_f^2}\frac{1}{2^{k/2}}, \left(1-\frac{1}{\sqrt{2}}\right)\frac{1}{16(1+\delta)^2}\frac{1}{\kappa_f^2\kappa_g}\frac{1}{2^{k/2}}, \frac{1}{1+\delta}\left(1-\frac{1}{\sqrt{2}}\right)\frac{1}{4\kappa_f^2}\frac{1}{2^{k/2}}\right\}$$

$$= 1-\min\left\{\left(1-\frac{1}{\sqrt{2}}\right)\frac{1}{8\kappa_f^2}, \left(1-\frac{1}{\sqrt{2}}\right)\frac{1}{16(1+\delta)^2}\frac{1}{\kappa_f^2\kappa_g}, \frac{1}{1+\delta}\left(1-\frac{1}{\sqrt{2}}\right)\frac{1}{4\kappa_f^2}\right\}\frac{1}{2^{k/2}}$$

$$= 1-\frac{\rho}{2^{k/2}}, \tag{151}$$

where $\rho$ is defined in equation (129). Using Lemma E.5, we have

$$E\left[\Phi_{k,t+1}\right] \leq \rho_k E\left[\Phi_{k,t}\right] + 2ms_k^2(\sigma_x^2+\sigma_y^2) \leq \left(1-\frac{\rho}{2^{k/2}}\right)E\left[\Phi_{k,t}\right] + 2ms_k^2(\sigma_x^2+\sigma_y^2)$$

$$=: a_k E\left[\Phi_{k,t}\right] + C_1 s_k^2, \tag{152}$$

where $a_k = 1-\frac{\rho}{2^{k/2}}$ and $C_1 = 2m(\sigma_x^2+\sigma_y^2)$. We now unroll the above recursion to obtain

$$E\left[\Phi_{k,t+1}\right] \leq a_k^{t+1}E\left[\Phi_{k,0}\right] + \sum_{l=0}^{t}a_k^{t-l}C_1 s_k^2 = a_k^{t+1}E\left[\Phi_{k,0}\right] + C_1 s_k^2 a_k^t\sum_{l=0}^{t}a_k^{-l}$$

$$= a_k^{t+1}E\left[\Phi_{k,0}\right] + C_1 s_k^2 a_k^t\frac{a_k^{-(t+1)}-1}{a_k^{-1}-1}$$

$$\leq a_k^{t+1}E\left[\Phi_{k,0}\right] + C_1 s_k^2 a_k^t\frac{a_k^{-(t+1)}}{a_k^{-1}-1} = a_k^{t+1}E\left[\Phi_{k,0}\right] + C_1 s_k^2 a_k^{-1}\frac{a_k}{1-a_k}$$

$$= a_k^{t+1}E\left[\Phi_{k,0}\right] + C_1 s_k^2\frac{1}{1-a_k} = \left(1-\frac{\rho}{2^{k/2}}\right)^{t+1}E\left[\Phi_{k,0}\right] + C_1 s_k^2 2^{k/2}\frac{1}{\rho}$$

Using $C_1 = 2m(\sigma_x^2+\sigma_y^2)$ we have

$$E\left[\Phi_{k,t+1}\right] \leq \left(1-\frac{\rho}{2^{k/2}}\right)^{t+1}E\left[\Phi_{k,0}\right] + 2m(\sigma_x^2+\sigma_y^2)\frac{1}{16L^2\kappa_f^2 2^k}2^{k/2}\frac{1}{\rho}$$

$$= \left(1-\frac{\rho}{2^{k/2}}\right)^{t+1}E\left[\Phi_{k,0}\right] + \frac{m(\sigma_x^2+\sigma_y^2)}{8\rho L^2\kappa_f^2}\frac{1}{2^{k/2}}. \tag{153}$$

# F  Proof of Theorem 4.1

This proof is based on several intermediate results proved in Appendices C-E. Hence it would be useful to refer to those results in order to appreciate the proof of Theorem 4.1.

We divide the proof of Theorem 4.1 into two parts. We first find the total number outer iterations required by Algorithm 2 to achieve target accuracy $\epsilon$. Then we derive the total gradient computation complexity of Algorithm 2.

### F.0.1  Total Number of Outer Iterations

We have the following initializations at every $k+1$ outer iterate:

$$x_{k+1,0} = x_{k,t_k}, \; y_{k+1,0} = y_{k,t_k}. \tag{154}$$

Therefore,

$$\|x_{k+1,0} - \mathbf{1}x^\star\|^2 + \|y_{k+1,0} - \mathbf{1}y^\star\|^2 = \|x_{k,t_k} - \mathbf{1}x^\star\|^2 + \|y_{k,t_k} - \mathbf{1}y^\star\|^2$$

$$\leq \frac{1}{\min\{M_{x,k}, M_{y,k}\}}\left(M_{x,k}\|x_{k,t_k} - \mathbf{1}x^\star\|^2 + M_{y,k}\|y_{k,t_k} - \mathbf{1}y^\star\|^2\right)$$

$$\leq \frac{1}{M_k}\Phi_{k,t_k}. \tag{155}$$

By taking total expectation on both sides and using Lemma E.6, we obtain

$$E \|x_{k+1,0} - \mathbf{1}x^\star\|^2 + E \|y_{k+1,0} - \mathbf{1}y^\star\|^2 \leq \frac{1}{M_k} \left( \left(1 - \frac{\rho}{2^{k/2}}\right)^{t_k} E[\Phi_{k,0}] + \frac{m(\sigma_x^2 + \sigma_y^2)}{8\rho L^2 \kappa_f^2} \frac{1}{2^{k/2}} \right)$$

$$\leq \frac{E[\Phi_{k,0}]}{M_k} \left(1 - \frac{\rho}{2^{k/2}}\right)^{t_k} + \frac{m(\sigma_x^2 + \sigma_y^2)}{8 M_k \rho L^2 \kappa_f^2} \frac{1}{2^{k/2}}.$$

(156)

We now focus on bounding $\Phi_{k,0}$ in terms of $\|x_{k,0} - \mathbf{1}x^\star\|^2$ and $\|y_{k,0} - \mathbf{1}y^\star\|^2$. Recall from (129) that

$$\Phi_{k,0} = M_{x,k} \|x_{k,0} - \mathbf{1}x^\star\|^2 + \frac{2s_k^2}{\gamma_{x,k}} \left\|D_{k,0}^x - D_x^\star\right\|_{(I-W)^\dagger}^2 + \sqrt{\delta} \left\|H_{k,0}^x - H_{x,k}^\star\right\|^2 \tag{157}$$

$$+ M_{y,k} \|y_{k,0} - \mathbf{1}y^\star\|^2 + \frac{2s_k^2}{\gamma_{y,k}} \left\|D_{k,0}^y - D_y^\star\right\|_{(I-W)^\dagger}^2 + \sqrt{\delta} \left\|H_{k,0}^y - H_{y,k}^\star\right\|^2. \tag{158}$$

We now bound the various terms appearing on the r.h.s. of the above equation. First,

$$\left\|H_{k,0}^x - H_{x,k}^\star\right\|^2 = \left\|x_{k,0} - \mathbf{1}(x^\star - \frac{s_k}{m}\nabla_x f(z^\star))\right\|^2$$

$$\leq 2 \|x_{k,0} - \mathbf{1}x^\star\|^2 + \frac{2s_k^2}{m} \|\nabla_x f(z^\star))\|^2. \tag{159}$$

Moreover,

$$\left\|H_{k,0}^y - H_{y,k}^\star\right\|^2 = \left\|y_{k,0} - \mathbf{1}(y^\star + \frac{s_k}{m}\nabla_y f(z^\star))\right\|^2$$

$$\leq 2 \|y_{k,0} - \mathbf{1}y^\star\|^2 + \frac{2s_k^2}{m} \|\nabla_y f(z^\star))\|^2. \tag{160}$$

We now have

$$\left\|D_{k,0}^x - D_x^\star\right\|_{(I-W)^\dagger}^2 = \|(I - J)\nabla_x F(\mathbf{1}z^\star)\|_{(I-W)^\dagger}^2$$

$$=: C_2. \tag{161}$$

$$\left\|D_{k,0}^y - D_y^\star\right\|_{(I-W)^\dagger}^2 = \|(I - J)\nabla_y F(\mathbf{1}z^\star)\|_{(I-W)^\dagger}^2 \tag{162}$$

$$=: C_3. \tag{163}$$

Therefore,

$$\frac{2s_k^2}{\gamma_{x,k}} \left\|D_{k,0}^x - D_x^\star\right\|_{(I-W)^\dagger}^2 = \frac{4(1+\delta)^2 \lambda_{\max}(I-W)s_k^2}{b_{x,k}} C_2$$

$$= \frac{4(1+\delta)^2 \lambda_{\max}(I-W)}{b_{x,k} 16 L^2 \kappa_f^2 2^k} C_2$$

$$= \frac{(1+\delta)^2 \lambda_{\max}(I-W)}{b_{x,k} 4 L^2 \kappa_f^2 2^k} C_2$$

$$\leq \frac{(1+\delta)^2 2}{4 L^2 \kappa_f^2 2^k} C_2 \frac{4\kappa_f^2 2^{k/2}}{\left(1 - \frac{1}{\sqrt{2}}\right)}$$

$$= \frac{2(1+\delta)^2 C_2}{L^2 2^{k/2} \left(1 - \frac{1}{\sqrt{2}}\right)}. \tag{164}$$

Similarly, we have

$$\frac{2s_k^2}{\gamma_{y,k}} \left\|D_{k,0}^y - D_y^\star\right\|_{(I-W)^\dagger}^2 \leq \frac{2(1+\delta)^2 C_3}{L^2 2^{k/2} \left(1 - \frac{1}{\sqrt{2}}\right)}. \tag{165}$$

Substituting the above bounds in (158) gives

$$\Phi_{k,0} \leq (M_{x,k} + 2\sqrt{\delta}) \|x_{k,0} - \mathbf{1}x^\star\|^2 + (M_{y,k} + 2\sqrt{\delta}) \|y_{k,0} - \mathbf{1}y^\star\|^2$$

$$+ \frac{\sqrt{\delta}}{8 m L^2 \kappa_f^2 2^k} \left(\|\nabla_x f(z^\star))\|^2 + \|\nabla_y f(z^\star))\|^2\right) + \frac{2(1+\delta)^2(C_2 + C_3)}{L^2 2^{k/2} \left(1 - \frac{1}{\sqrt{2}}\right)}. \tag{166}$$

On substituting the above inequality in (156), we obtain

$$E \left\| z_{k+1,0} - \mathbf{1} z^\star \right\|^2$$

$$\leq \frac{1}{M_k} \left(1 - \frac{\rho}{2^{k/2}}\right)^{t_k} \left( (M_{x,k} + 2\sqrt{\delta}) E \left\| x_{k,0} - \mathbf{1} x^\star \right\|^2 + (M_{y,k} + 2\sqrt{\delta}) E \left\| y_{k,0} - \mathbf{1} y^\star \right\|^2 \right)$$

$$+ \frac{1}{M_k} \left(1 - \frac{\rho}{2^{k/2}}\right)^{t_k} \left( \frac{\sqrt{\delta}}{8 m L^2 \kappa_f^2 2^k} \left( \|\nabla_x f(z^\star))\|^2 + \|\nabla_y f(z^\star))\|^2 \right) + \frac{2(1+\delta)^2(C_2 + C_3)}{L^2 2^{k/2} \left(1 - \frac{1}{\sqrt{2}}\right)} \right)$$

$$+ \frac{m(\sigma_x^2 + \sigma_y^2)}{8 M_k \rho L^2 \kappa_f^2} \frac{1}{2^{k/2}}. \tag{167}$$

We have

$$t_k = \frac{1}{-\log\left(1 - \frac{\rho}{2^{k/2}}\right)} \max \left\{ \log\left( \frac{3 M_{x,k} + 6\sqrt{\delta}}{M_k} \right), \log\left( \frac{3 M_{y,k} + 6\sqrt{\delta}}{M_k} \right), \log\left( \frac{3}{M_k} \right) \right\}. \tag{168}$$

Therefore,

$$t_k \geq \frac{1}{-\log\left(1 - \frac{\rho}{2^{k/2}}\right)} \log\left( \frac{3 M_{x,k} + 6\sqrt{\delta}}{M_k} \right)$$

$$\Rightarrow -t_k \log\left(1 - \frac{\rho}{2^{k/2}}\right) \geq \log\left( \frac{3 M_{x,k} + 6\sqrt{\delta}}{M_k} \right) = -\log\left( \frac{M_k}{3 M_{x,k} + 6\sqrt{\delta}} \right)$$

$$\Rightarrow t_k \log\left(1 - \frac{\rho}{2^{k/2}}\right) \leq \log\left( \frac{M_k}{3 M_{x,k} + 6\sqrt{\delta}} \right)$$

$$\Rightarrow \left(1 - \frac{\rho}{2^{k/2}}\right)^{t_k} \leq \frac{M_k}{3(M_{x,k} + 2\sqrt{\delta})}. \tag{169}$$

Moreover,

$$t_k \geq \frac{1}{-\log\left(1 - \frac{\rho}{2^{k/2}}\right)} \log\left( \frac{3}{M_k} \right)$$

$$\Rightarrow -t_k \log\left(1 - \frac{\rho}{2^{k/2}}\right) \geq \log\left( \frac{3}{M_k} \right)$$

$$\Rightarrow t_k \log\left(1 - \frac{\rho}{2^{k/2}}\right) \leq \log\left( \frac{M_k}{3} \right)$$

$$\Rightarrow \left(1 - \frac{\rho}{2^{k/2}}\right)^{t_k} \leq \frac{M_k}{3}. \tag{170}$$

By using above inequalities into (167), we obtain ,

$$E \left\| z_{k+1,0} - \mathbf{1} z^\star \right\|^2$$

$$\leq \frac{1}{3} E \left\| x_{k,0} - \mathbf{1} x^\star \right\|^2 + \frac{1}{3} E \left\| y_{k,0} - \mathbf{1} y^\star \right\|^2$$

$$+ \frac{1}{3} \left( \frac{\sqrt{\delta}}{8 m L^2 \kappa_f^2 2^k} \left( \|\nabla_x f(z^\star))\|^2 + \|\nabla_y f(z^\star))\|^2 \right) + \frac{2(1+\delta)^2(C_2 + C_3)}{L^2 2^{k/2} \left(1 - \frac{1}{\sqrt{2}}\right)} \right) + \frac{m(\sigma_x^2 + \sigma_y^2)}{8 M_k \rho L^2 \kappa_f^2} \frac{1}{2^{k/2}}$$

$$= \frac{1}{3} E \left\| x_{k,0} - \mathbf{1} x^\star \right\|^2 + \frac{1}{3} E \left\| y_{k,0} - \mathbf{1} y^\star \right\|^2 + \frac{A_1}{2^k} + \frac{A_2}{2^{k/2}} + \frac{m(\sigma_x^2 + \sigma_y^2)}{8 M_k \rho L^2 \kappa_f^2} \frac{1}{2^{k/2}}, \tag{171}$$

where $A_1 = \frac{\sqrt{\delta}}{24 m L^2 \kappa_f^2} \left( \|\nabla_x f(z^\star))\|^2 + \|\nabla_y f(z^\star))\|^2 \right)$, $A_2 = \frac{2(1+\delta)^2(C_2 + C_3)}{3 L^2 \left(1 - \frac{1}{\sqrt{2}}\right)}$. To proceed further, we derive lower bounds on $M_{x,k}$ and $M_{y,k}$.

**Lower bound on $M_{x,k}$.**    Using (118), we have

$$b_{x,k} \leq \frac{1}{4\kappa_x \kappa_f} \tag{172}$$

$$\Rightarrow \frac{b_{x,k}}{4(1+\delta)^2} \leq \frac{1}{16(1+\delta)^2 \kappa_x \kappa_f}$$

$$\Rightarrow 1 - \frac{b_{x,k}}{4(1+\delta)^2} \geq 1 - \frac{1}{16(1+\delta)^2 \kappa_x \kappa_f}$$

$$\Rightarrow \frac{1}{1 - \frac{b_{x,k}}{4(1+\delta)^2}} \leq \frac{1}{1 - \frac{1}{16(1+\delta)^2 \kappa_x \kappa_f}} \tag{173}$$

We also have

$$\frac{\alpha_{x,k}\sqrt{\delta}}{1 - \frac{\gamma_{x,k}\lambda_{\max}(I-W)}{2}} \leq \frac{\alpha_{x,k}\sqrt{\delta}}{1 - \frac{1}{16(1+\delta)^2 \kappa_x \kappa_f}} \tag{174}$$

$$= \frac{\frac{b_{x,k}}{1+\delta}\sqrt{\delta}}{1 - \frac{1}{16(1+\delta)^2 \kappa_x \kappa_f}} = \frac{\sqrt{\delta} b_{x,k} 16(1+\delta)^2 \kappa_x \kappa_f}{(1+\delta)(16(1+\delta)^2 \kappa_x \kappa_f - 1)}$$

$$= \frac{16\sqrt{\delta}(1+\delta)\kappa_x \kappa_f}{(16(1+\delta)^2 \kappa_x \kappa_f - 1)} \frac{1}{4\kappa_x \kappa_f} = \frac{4\sqrt{\delta}(1+\delta)}{(16(1+\delta)^2 \kappa_x \kappa_f - 1)}$$

$$= \frac{4\sqrt{\delta}(1+\delta)}{15(1+\delta)^2 \kappa_x \kappa_f + (1+\delta)^2 \kappa_x \kappa_f - 1}$$

$$\leq \frac{4\sqrt{\delta}(1+\delta)}{15(1+\delta)^2 \kappa_x \kappa_f} = \frac{4\sqrt{\delta}}{15(1+\delta)\kappa_x \kappa_f}. \tag{175}$$

Therefore, $M_{x,k}$ is lower bounded by $1 - \frac{4\sqrt{\delta}}{15(1+\delta)\kappa_x \kappa_f}$ because

$$M_{x,k} = 1 - \frac{\alpha_{x,k}\sqrt{\delta}}{1 - \frac{\gamma_{x,k}\lambda_{\max}(I-W)}{2}} \geq 1 - \frac{4\sqrt{\delta}}{15(1+\delta)\kappa_x \kappa_f} =: \tilde{M}. \tag{176}$$

Consider

$$\frac{m(\sigma_x^2 + \sigma_y^2)}{8M_k \rho L^2 \kappa_f^2} \frac{1}{2^{k/2}} \leq \frac{m(\sigma_x^2 + \sigma_y^2)}{8\tilde{M}\rho L^2 \kappa_f^2} \frac{1}{2^{k/2}}$$

$$=: \frac{A_3}{2^{k/2}}, \tag{177}$$

where $A_3 = \frac{m(\sigma_x^2 + \sigma_y^2)}{8\tilde{M}\rho L^2 \kappa_f^2}$. On substituting these bounds in (167), we obtain

$$E\|z_{k+1,0} - \mathbf{1}z^\star\|^2 \leq \frac{1}{3}E\|z_{k,0} - \mathbf{1}z^\star\|^2 + \frac{A_1}{2^k} + \frac{(A_2 + A_3)}{2^{k/2}}$$

$$= \frac{1}{3}E\|z_{k,0} - \mathbf{1}z^\star\|^2 + e_k, \tag{178}$$

where $e_k = \frac{A_1}{2^k} + \frac{(A_2+A_3)}{2^{k/2}}$. Using (178) recursively yields

$$E\left\|z_{k+1,0} - \mathbf{1}z^\star\right\|^2 \leq \frac{1}{3^{k+1}} E\left\|z_0 - \mathbf{1}z^\star\right\|^2 + \sum_{l=0}^{k} \frac{1}{3^{k-l}} e_l$$

$$= \frac{1}{3^{k+1}} E\left\|z_0 - \mathbf{1}z^\star\right\|^2 + \frac{1}{3^k} \sum_{l=0}^{k} 3^l \left(\frac{A_1}{2^l} + \frac{(A_2+A_3)}{2^{l/2}}\right)$$

$$= \frac{E\left\|z_0 - \mathbf{1}z^\star\right\|^2}{3^{k+1}} + \frac{A_1}{3^k} \sum_{l=0}^{k} \left(\frac{3}{2}\right)^l + \frac{(A_2+A_3)}{3^k} \sum_{l=0}^{k} \left(\frac{3}{\sqrt{2}}\right)^l$$

$$= \frac{E\left\|z_0 - \mathbf{1}z^\star\right\|^2}{3^{k+1}} + \frac{A_1}{3^k} \left(\frac{(3/2)^{k+1} - 1}{\frac{3}{2} - 1}\right) + \frac{(A_2+A_3)}{3^k} \left(\frac{(3/\sqrt{2})^{k+1} - 1}{\frac{3}{\sqrt{2}} - 1}\right)$$

$$\leq \frac{E\left\|z_0 - \mathbf{1}z^\star\right\|^2}{3^{k+1}} + \frac{2A_1}{3^k} \frac{3^{k+1}}{2^{k+1}} + \frac{(A_2+A_3)}{3^k} \frac{\sqrt{2}}{3 - \sqrt{2}} \frac{3^{k+1}}{\sqrt{2}^{k+1}}$$

$$= \frac{E\left\|z_0 - \mathbf{1}z^\star\right\|^2}{3^{k+1}} + \frac{3A_1}{2^k} + \frac{3}{3 - \sqrt{2}} \frac{(A_2+A_3)}{2^{k/2}}$$

$$\leq \frac{E\left\|z_0 - \mathbf{1}z^\star\right\|^2}{2^k} + \frac{3A_1}{2^k} + \frac{3}{3 - \sqrt{2}} \frac{(A_2+A_3)}{2^{k/2}}$$

$$= \frac{\left\|z_0 - \mathbf{1}z^\star\right\|^2 + 3A_1}{2^k} + \frac{3}{3 - \sqrt{2}} \frac{(A_2+A_3)}{2^{k/2}}. \tag{179}$$

By choosing $k = K(\epsilon) = \max\{\log_2\left(\frac{2\|z_0 - \mathbf{1}z^\star\|^2 + 6A_1}{\epsilon}\right), 2\log_2\left(\frac{6(A_2+A_3)}{(3-\sqrt{2})\epsilon}\right)\}$, we obtain

$$E\left\|z_{K(\epsilon)+1,0} - \mathbf{1}z^\star\right\|^2 \leq \epsilon. \tag{180}$$

Above choice of $K(\epsilon)$ involves constants $A_1, A_2, A_3$. We write each term in $K(\epsilon)$ in terms of parameters $m, L, \kappa_f, \kappa_g, \delta$ and $\sigma$ as follows:

We have $\log_2\left(\frac{2\left\|z_0 - \mathbf{1}z^\star\right\|^2 + 6A_1}{\epsilon}\right) = \log_2\left(\frac{2\left\|z_0 - \mathbf{1}z^\star\right\|^2}{\epsilon} + \frac{\sqrt{\delta}(\|\nabla_x f(z^\star)\|^2 + \|\nabla_y f(z^\star)\|^2)}{4mL^2\kappa_f^2\epsilon}\right)$

$$= \mathcal{O}\left(\log_2\left(\frac{\left\|z_0 - \mathbf{1}z^\star\right\|^2}{\epsilon} + \frac{\sqrt{\delta}}{mL^2\kappa_f^2\epsilon}\right)\right)$$

We also have $2\log_2\left(\frac{6(A_2+A_3)}{(3-\sqrt{2})\epsilon}\right) = 2\log_2\left(\frac{6}{(3-\sqrt{2})\epsilon}\left(\frac{2(1+\delta)^2(C_2+C_3)}{3L^2(1-1/\sqrt{2})} + \frac{m\sigma^2}{8\tilde{M}\rho L^2\kappa_f^2}\right)\right)$

$$= \mathcal{O}\left(\log_2\left(\frac{(1+\delta)^2 + m\sigma^2\kappa_g(1+\delta)^2}{L^2\epsilon}\right)\right).$$

The last equality follows because $1/\rho = \mathcal{O}\left((1+\delta)^2\kappa_f^2\kappa_g\right)$ and $\tilde{M} = \mathcal{O}(1)$.

### F.0.2 Gradient Computation Complexity

The gradient computation complexity $T_{\mathrm{grad}}(\epsilon)$ is bounded by the following computation

$$
T_{\mathrm{grad}}(\epsilon) = \sum_{k=0}^{K(\epsilon)-1} t_k
$$

$$
= \sum_{k=0}^{K(\epsilon)-1} \frac{1}{-\log\left(1 - \frac{\rho}{2^{k/2}}\right)} \max\left\{ \log\left(\frac{3M_{x,k} + 6\sqrt{\delta}}{M_k}\right), \log\left(\frac{3M_{y,k} + 6\sqrt{\delta}}{M_k}\right), \log\left(\frac{3}{M_k}\right) \right\}
$$

$$
\leq \sum_{k=0}^{K(\epsilon)-1} \frac{1}{-\log\left(1 - \frac{\rho}{2^{k/2}}\right)} \max\left\{ \log\left(\frac{3 + 6\sqrt{\delta}}{M_k}\right), \log\left(\frac{3 + 6\sqrt{\delta}}{M_k}\right), \log\left(\frac{3}{M_k}\right) \right\}
$$

$$
\leq \log\left(\frac{3 + 6\sqrt{\delta}}{\tilde{M}}\right) \sum_{k=0}^{K(\epsilon)-1} \frac{1}{-\log\left(1 - \frac{\rho}{2^{k/2}}\right)}
$$

$$
\leq \log\left(\frac{3 + 6\sqrt{\delta}}{\tilde{M}}\right) \sum_{k=0}^{K(\epsilon)-1} \frac{2^{k/2}5}{\rho}
$$

$$
= 5\log\left(\frac{3 + 6\sqrt{\delta}}{\tilde{M}}\right) \frac{2^{K(\epsilon)/2} - 1}{\rho(\sqrt{2} - 1)}
$$

$$
\leq 5\log\left(\frac{3 + 6\sqrt{\delta}}{\tilde{M}}\right) \frac{2^{K(\epsilon)/2}}{\rho(\sqrt{2} - 1)}
$$

$$
= \frac{5}{\rho(\sqrt{2} - 1)} \log\left(\frac{3 + 6\sqrt{\delta}}{\tilde{M}}\right) \sqrt{2}^{\max\left\{\log_2\left(\frac{2\|z_0 - \mathbf{1}z^\star\|^2 + 6A_1}{\epsilon}\right), 2\log_2\left(\frac{6(A_2 + A_3)}{(3 - \sqrt{2})\epsilon}\right)\right\}}
$$

$$
= \frac{5}{\rho(\sqrt{2} - 1)} \log\left(\frac{3 + 6\sqrt{\delta}}{\tilde{M}}\right) \max\left\{ (\sqrt{2})^{\log_2\left(\frac{2\|z_0 - \mathbf{1}z^\star\|^2 + 6A_1}{\epsilon}\right)}, (\sqrt{2})^{2\log_2\left(\frac{6(A_2 + A_3)}{(3 - \sqrt{2})\epsilon}\right)} \right\}
$$

$$
= \frac{5}{\rho(\sqrt{2} - 1)} \log\left(\frac{3 + 6\sqrt{\delta}}{\tilde{M}}\right) \max\left\{ \sqrt{\frac{2\|z_0 - \mathbf{1}z^\star\|^2 + 6A_1}{\epsilon}}, \frac{6(A_2 + A_3)}{(3 - \sqrt{2})\epsilon} \right\}
$$

$$
= \frac{5}{\sqrt{2} - 1} \log\left(\frac{3 + 6\sqrt{\delta}}{\tilde{M}}\right)
$$

$$
\times \left( \min\left\{ \left(1 - \frac{1}{\sqrt{2}}\right) \frac{1}{8\kappa_f^2}, \left(1 - \frac{1}{\sqrt{2}}\right) \frac{1}{16(1 + \delta)^2} \frac{1}{\kappa_f^2\kappa_g}, \frac{1}{1 + \delta} \left(1 - \frac{1}{\sqrt{2}}\right) \frac{1}{4\kappa_f^2} \right\} \right)^{-1} \times
$$

$$
\max\left\{ \sqrt{\frac{2\|z_0 - \mathbf{1}z^\star\|^2}{\epsilon} + \frac{\sqrt{\delta}\left(\|\nabla_x f(z^\star))\|^2 + \|\nabla_y f(z^\star))\|^2\right)}{4mL^2\kappa_f^2\epsilon}}, \mathcal{O}\left( \left(\frac{(1 + \delta)^2(C_2 + C_3)}{L^2} + \frac{m\sigma^2}{\tilde{M}\rho L^2\kappa_f^2}\right) \frac{1}{\epsilon} \right) \right\}
$$

$$
= \mathcal{O}\left( \max\left\{ \frac{(1 + \delta)^2 \|z_0 - \mathbf{1}z^\star\| \kappa_f^2\kappa_g}{\sqrt{\epsilon}} + \frac{(1 + \delta)^2\delta^{1/4}\kappa_f\kappa_g}{\sqrt{m}L\sqrt{\epsilon}}, \frac{(1 + \delta)^4(\kappa_f^2\kappa_g + m\sigma^2\kappa_f^2\kappa_g^2)}{L^2\epsilon} \right\} \right).
$$
(181)

Notice that above complexity does not contain $1/\tilde{M}$ term because $\tilde{M} \in \left[\frac{11}{15}, 1\right]$.

### F.0.3   Communication Complexity

We finish the proof of Theorem 4.1 by computing the communication complexity as follows.

$$T_{\text{comm}}(\epsilon) = \sum_{k=0}^{K(\epsilon)-1} (t_k + 1)$$

$$= T_{\text{grad}}(\epsilon) + K(\epsilon)$$

$$= \mathcal{O}\left( \max\left\{ \frac{(1+\delta)^2(\|z_0 - \mathbf{1}z^\star\| \kappa_f^2\kappa_g + \kappa_f\kappa_g)}{\sqrt{\epsilon}}, \frac{(1+\delta)^4\kappa_f^2\kappa_g}{L^2\epsilon} + \frac{(1+\delta)^4\sigma^2\kappa_f^2\kappa_g^2}{L^2\epsilon} \right\} \right.$$

$$\left. + \log_2\left( \frac{(1+\delta)^2}{L^2\sqrt{\delta}\epsilon} + \frac{m\sigma^2}{\kappa_f^4\kappa_g(1+\delta)^2L^2\epsilon} \right) \right). \tag{182}$$

### F.1   Algorithm 2 behavior in deterministic setting

In this section, we briefly discuss that Algorithm 2 converges to the saddle point solution with linear rates when $\sigma_x = \sigma_y = 0$, where $\sigma_x$ and $\sigma_y$ are the bounds on the variances of stochastic gradients of GSGO (see Assumption E.1). Recall the recursive relation in Lemma E.5:

$$E\left[\Phi_{k,t+1}\right] \leq \rho_k E\left[\Phi_{k,t}\right] + 2ms_k^2(\sigma_x^2 + \sigma_y^2).$$

On substituting $k = 0$ and $\sigma_x = \sigma_y = 0$ in above inequality, we obtain

$$E\left[\Phi_{0,t+1}\right] \leq \rho_0 E\left[\Phi_{0,t}\right], \tag{183}$$

where $\rho_0$ is defined in (128). By unrolling above recursion in $t$, we get

$$E\left[\Phi_{0,t+1}\right] \leq \rho_0^{t+1}\Phi_{0,0}. \tag{184}$$

Note that $M_{x,0}\|x_{0,t} - \mathbf{1}x^\star\|^2 + M_{y,0}\|y_{0,t} - \mathbf{1}y^\star\|^2 \leq \Phi_{0,t}$ from (105). Therefore, $E\|x_{0,t} - \mathbf{1}x^\star\|^2 + E\|y_{0,t} - \mathbf{1}y^\star\|^2 \leq \frac{E[\Phi_{0,t}]}{\min\{M_{x,0},M_{y,0}\}}$. Under the above settings, Algorithm 2 needs $T_{grad}(\epsilon) = \frac{1}{\log(1/\rho_0)}\log\left(\frac{\Phi_{0,0}}{\epsilon\min\{M_{x,0},M_{y,0}\}}\right)$ gradient computations and communications to achieve $E\|x_{0,T_{grad}(\epsilon)} - \mathbf{1}x^\star\|^2 + E\|y_{0,T_{grad}(\epsilon)} - \mathbf{1}y^\star\|^2 \leq \epsilon$. We now write $T_{grad}(\epsilon)$ in terms of $\kappa_f, \kappa_g$ and $\delta$. Using (151), we have $\rho_0 \leq 1 - \rho$, where $\rho$ is defined in (129). Notice that $\frac{1}{\log(1/\rho_0)} \leq \frac{1}{-\log(1-\rho)} \leq \frac{5}{\rho}$. Therefore,

$$T_{grad}(\epsilon) = 5\left( \min\left\{ \left(1 - \frac{1}{\sqrt{2}}\right)\frac{1}{8\kappa_f^2}, \left(1 - \frac{1}{\sqrt{2}}\right)\frac{1}{16(1+\delta)^2}\frac{1}{\kappa_f^2\kappa_g}, \frac{1}{1+\delta}\left(1 - \frac{1}{\sqrt{2}}\right)\frac{1}{4\kappa_f^2} \right\} \right)^{-1}$$

$$\log\left( \frac{\Phi_{0,0}}{\epsilon\min\{M_{x,0},M_{y,0}\}} \right)$$

$$= \mathcal{O}\left( \max\{8\kappa_f^2, 16(1+\delta)^2\kappa_f^2\kappa_g, 4(1+\delta)\kappa_f^2\}\log\left(\frac{\Phi_{0,0}}{\epsilon}\right) \right).$$

## G   Proofs for the Finite Sum Setting

In this section, we prove all results related to convergence analysis of Algorithm 3 in Section 5. For theoretical analysis of Algorithm 3, we make the following smoothness assumptions on $f_{ij}$ particular to the finite sum setting.

**Assumption G.1.** Assume that each $f_{ij}(x, y)$ is $L_{xx}$ smooth in $x$; for every fixed $y$, $\|\nabla_x f_{ij}(x_1, y) - \nabla_x f_{ij}(x_2, y)\| \leq L_{xx}\|x_1 - x_2\|, \forall x_1, x_2 \in \mathbb{R}^{d_x}$

**Assumption G.2.** Assume that each $-f_{ij}(x, y)$ is $L_{yy}$ smooth in $y$ i.e., for every fixed $x$, $\|-\nabla_y f_{ij}(x, y_1) + \nabla_y f_{ij}(x, y_2)\| \leq L_{yy}\|y_1 - y_2\|, \forall y_1, y_2 \in \mathbb{R}^{d_y}$.

**Assumption G.3.** Assume that each $\nabla_x f_{ij}(x, y)$ is $L_{xy}$ Lipschitz in $y$ i.e., for every fixed $x$, $\|\nabla_x f_{ij}(x, y_1) - \nabla_x f_{ij}(x, y_2)\| \leq L_{xy}\|y_1 - y_2\|, \forall y_1, y_2 \in \mathbb{R}^{d_y}$.

**Assumption G.4.** Assume that each $\nabla_y f_{ij}(x, y)$ is $L_{yx}$ Lipschitz in $x$ i.e., for every fixed $y$, $\|\nabla_y f_{ij}(x_1, y) - \nabla_y f_{ij}(x_2, y)\| \leq L_{yx}\|x_1 - x_2\|, \forall x_1, x_2 \in \mathbb{R}^{d_x}$.

We begin with few intermediate results which will help us in getting the final convergence result.

**Lemma G.5.** *Let $\{x_t\}_t, \{y_t\}_t$ be the sequences generated by Algorithm 3 with $\mathcal{G}_t^x$ and $\mathcal{G}_t^y$ obtained from SVRGO. Then, under Assumptions 3.1-3.2 and Assumptions G.1-G.4, the following holds for all*

$t \geq 1$:

$$E\|x_t - \mathbf{1}x^\star - s\mathcal{G}_t^x + s\nabla_x F(\mathbf{1}x^\star, \mathbf{1}y^\star)\|^2 + E\|y_t - \mathbf{1}y^\star + s\mathcal{G}_t^y - s\nabla_y F(\mathbf{1}x^\star, \mathbf{1}y^\star)\|^2$$

$$\leq \left(1 - \mu_x s + \frac{4s^2 L_{yx}^2}{np_{\min}}\right)\|x_t - \mathbf{1}x^\star\|^2 + \left(1 - s\mu_y + \frac{4s^2 L_{xy}^2}{np_{\min}}\right)\|y_t - \mathbf{1}y^\star\|^2$$

$$- \left(2s - \frac{8s^2 L_{xx}}{np_{\min}}\right)\sum_{i=1}^m V_{f_i, y_t^i}(x^\star, x_t^i) - \left(2s - \frac{8s^2 L_{yy}}{np_{\min}}\right)\sum_{i=1}^m V_{-f_i, x_t^i}(y^\star, y_t^i)$$

$$+ \frac{4s^2(L_{xx}^2 + L_{yx}^2)}{np_{\min}}\|\tilde{x}_t - \mathbf{1}x^\star\|^2 + \frac{4s^2(L_{yy}^2 + L_{xy}^2)}{np_{\min}}\|\tilde{y}_t - \mathbf{1}y^\star\|^2, \tag{185}$$

*where* $p_{\min} := \min_{i,j}\{p_{ij}\}$.

## G.1 Proof of Lemma G.5

We begin the proof by bounding the primal ($x$) and dual ($y$) updates on the l.h.s. of (185) separately. In particular, we show that

$$E\|x_t - \mathbf{1}x^\star - s\mathcal{G}_t^x + s\nabla_x F(\mathbf{1}x^\star, \mathbf{1}y^\star)\|^2$$

$$\leq (1 - \mu_x s)\|x_t - \mathbf{1}x^\star\|^2$$

$$+ \frac{2s^2}{n^2 p_{\min}}\sum_{i=1}^m \sum_{j=1}^n \left\|\nabla_x f_{ij}(z_t^i) - \nabla_x f_{ij}(z^\star)\right\|^2 + \frac{2s^2}{n^2 p_{\min}}\sum_{i=1}^m \sum_{j=1}^n \left\|\nabla_x f_{ij}(\tilde{z}_t^i) - \nabla_x f_{ij}(z^\star)\right\|^2$$

$$- 2s\sum_{i=1}^m V_{f_i, y_t^i}(x^\star, x_t^i) + 2s\left(F(\mathbf{1}x^\star, y_t) - F(z_t) + F(x_t, \mathbf{1}y^\star) - F(\mathbf{1}z^\star)\right) \tag{186}$$

and

$$E\|y_t - \mathbf{1}y^\star + s\mathcal{G}_t^y - s\nabla_y F(\mathbf{1}x^\star, \mathbf{1}y^\star)\|^2$$

$$\leq (1 - s\mu_y)\|y_t - \mathbf{1}y^\star\|^2 + 2s\left(-F(x_t, \mathbf{1}y^\star) + F(\mathbf{1}z^\star) - F(\mathbf{1}x^\star, y_t) + F(z_t)\right) - 2s\sum_{i=1}^m V_{-f_i, x_t^i}(y^\star, y_t^i)$$

$$+ \frac{2s^2}{n^2 p_{\min}}\sum_{i=1}^m \sum_{j=1}^n \left\|\nabla_y f_{ij}(z_t^i) - \nabla_y f_{ij}(z^\star)\right\|^2 + \frac{2s^2}{n^2 p_{\min}}\sum_{i=1}^m \sum_{j=1}^n \left\|\nabla_y f_{ij}(\tilde{z}_t^i) - \nabla_y f_{ij}(z^\star)\right\|^2. \tag{187}$$

Observe that (186) and (187) are similar, and we only prove (186) in Section G.1.1 below. Adding (186) and (187), we obtain

$$E\|x_t - \mathbf{1}x^\star - s\mathcal{G}_t^x + s\nabla_x F(\mathbf{1}x^\star, \mathbf{1}y^\star)\|^2 + E\|y_t - \mathbf{1}y^\star + s\mathcal{G}_t^y - s\nabla_y F(\mathbf{1}x^\star, \mathbf{1}y^\star)\|^2$$

$$\leq (1 - \mu_x s)\|x_t - \mathbf{1}x^\star\|^2 + (1 - s\mu_y)\|y_t - \mathbf{1}y^\star\|^2 \tag{188}$$

$$- 2s\sum_{i=1}^m V_{f_i, y_t^i}(x^\star, x_t^i) - 2s\sum_{i=1}^m V_{-f_i, x_t^i}(y^\star, y_t^i)$$

$$+ \frac{2s^2}{n^2 p_{\min}}\sum_{i=1}^m \sum_{j=1}^n \left(\left\|\nabla_x f_{ij}(z_t^i) - \nabla_x f_{ij}(z^\star)\right\|^2 + \left\|\nabla_y f_{ij}(z_t^i) - \nabla_y f_{ij}(z^\star)\right\|^2\right)$$

$$+ \frac{2s^2}{n^2 p_{\min}}\sum_{i=1}^m \sum_{j=1}^n \left(\left\|\nabla_x f_{ij}(\tilde{z}_t^i) - \nabla_x f_{ij}(z^\star)\right\|^2 + \left\|\nabla_y f_{ij}(\tilde{z}_t^i) - \nabla_y f_{ij}(z^\star)\right\|^2\right). \tag{189}$$

To finish the proof of Lemma G.5, we bound the last two terms of (189) as shown in Section G.1.2.

### G.1.1 Proof of (186)

First, consider the primal update term

$$E \left\| x_t - \mathbf{1}x^\star - s\mathcal{G}_t^x + s\nabla_x F(\mathbf{1}x^\star, \mathbf{1}y^\star) \right\|^2$$

$$= \sum_{i=1}^m E \left\| x_t^i - x^\star - s\mathcal{G}_t^{i,x} + s\nabla_x f_i(x^\star, y^\star) \right\|^2$$

$$= \sum_{i=1}^m \left\| x_t^i - x^\star \right\|^2 + s^2 \sum_{i=1}^m E \left\| \mathcal{G}_t^{i,x} - \nabla_x f_i(x^\star, y^\star) \right\|^2$$

$$- 2s \sum_{i=1}^m E \left\langle x_t^i - x^\star, \mathcal{G}_t^{i,x} - \nabla_x f_i(x^\star, y^\star) \right\rangle$$

$$= \sum_{i=1}^m \left\| x_t^i - x^\star \right\|^2 + s^2 \sum_{i=1}^m E \left\| \frac{1}{np_{il}} \left( \nabla_x f_{il}(z_t^i) - \nabla_x f_{il}(\tilde{z}_t^i) \right) + \nabla_x f_i(\tilde{z}_t^i) - \nabla_x f_i(x^\star, y^\star) \right\|^2$$

$$- 2s \sum_{i=1}^m E \left\langle x_t^i - x^\star, \frac{1}{np_{il}} \left( \nabla_x f_{il}(z_t^i) - \nabla_x f_{il}(\tilde{z}_t^i) \right) + \nabla_x f_i(\tilde{z}_t^i) - \nabla_x f_i(x^\star, y^\star) \right\rangle. \quad (190)$$

Observe that

$$E \left[ \frac{1}{np_{il}} \left( \nabla_x f_{il}(z_t^i) - \nabla_x f_{il}(\tilde{z}_t^i) \right) + \nabla_x f_i(\tilde{z}_t^i) - \nabla_x f_i(x^\star, y^\star) \right]$$

$$= \sum_{l=1}^n \frac{\nabla_x f_{il}(z_t^i) - \nabla_x f_{il}(\tilde{z}_t^i)}{np_{il}} \times p_{il} + \nabla_x f_i(\tilde{z}_t^i) - \nabla_x f_i(x^\star, y^\star)$$

$$= \frac{1}{n} \sum_{l=1}^n \nabla_x f_{il}(z_t^i) - \frac{1}{n} \sum_{l=1}^n \nabla_x f_{il}(\tilde{z}_t^i) + \nabla_x f_i(\tilde{z}_t^i) - \nabla_x f_i(z^\star)$$

$$= \nabla_x f_i(z_t^i) - \nabla_x f_i(\tilde{z}_t^i) + \nabla_x f_i(\tilde{z}_t^i) - \nabla_x f_i(z^\star)$$

$$= \nabla_x f_i(z_t^i) - \nabla_x f_i(z^\star), \quad (191)$$

where the first equality and second last equality follows respectively from step (1) of SVRGO and definition of $f_i(x, y)$. Substituting the above in the last term of (190), we see that

$$E \left\| x_t - \mathbf{1}x^\star - s\mathcal{G}_t^x + s\nabla_x F(\mathbf{1}x^\star, \mathbf{1}y^\star) \right\|^2$$

$$\leq \sum_{i=1}^m \left\| x_t^i - x^\star \right\|^2 + s^2 \sum_{i=1}^m E \left\| \frac{1}{np_{il}} \left( \nabla_x f_{il}(z_t^i) - \nabla_x f_{il}(\tilde{z}_t^i) \right) + \nabla_x f_i(\tilde{z}_t^i) - \nabla_x f_i(x^\star, y^\star) \right\|^2$$

$$- 2s \sum_{i=1}^m \left\langle x_t^i - x^\star, \nabla_x f_i(z_t^i) - \nabla_x f_i(z^\star) \right\rangle. \quad (192)$$

Substituting (111) (i.e., Bregman distance) and (113) (i.e., strong convexity of $f$) in (192), we obtain

$$E \left\| x_t - \mathbf{1}x^\star - s\mathcal{G}_t^x + s\nabla_x F(\mathbf{1}x^\star, \mathbf{1}y^\star) \right\|^2$$

$$\leq \sum_{i=1}^m \left\| x_t^i - x^\star \right\|^2 + s^2 \sum_{i=1}^m E \left\| \frac{1}{np_{il}} \left( \nabla_x f_{il}(z_t^i) - \nabla_x f_{il}(\tilde{z}_t^i) \right) + \nabla_x f_i(\tilde{z}_t^i) - \nabla_x f_i(x^\star, y^\star) \right\|^2$$

$$- 2s \sum_{i=1}^m \left( -f_i(x^\star, y_t^i) + f_i(z_t^i) + V_{f_i, y_t^i}(x^\star, x_t^i) \right) + 2s \sum_{i=1}^m \left( f_i(x_t^i, y^\star) - f_i(z^\star) - \frac{\mu_x}{2} \left\| x_t^i - x^\star \right\|^2 \right)$$

$$= \sum_{i=1}^m \left\| x_t^i - x^\star \right\|^2 + s^2 \sum_{i=1}^m E \left\| \frac{1}{np_{il}} \left( \nabla_x f_{il}(z_t^i) - \nabla_x f_{il}(\tilde{z}_t^i) \right) + \nabla_x f_i(\tilde{z}_t^i) - \nabla_x f_i(x^\star, y^\star) \right\|^2$$

$$+ 2s(F(\mathbf{1}x^\star, y_t) - F(z_t)) - 2s \sum_{i=1}^m V_{f_i, y_t^i}(x^\star, x_t^i) + 2s \left( F(x_t, \mathbf{1}y^\star) - F(\mathbf{1}z^\star) \right) - s\mu_x \left\| x_t - \mathbf{1}x^\star \right\|^2$$

$$= (1 - \mu_x s) \left\| x_t - \mathbf{1}x^\star \right\|^2 + s^2 \sum_{i=1}^m E \left\| \frac{1}{np_{il}} \left( \nabla_x f_{il}(z_t^i) - \nabla_x f_{il}(\tilde{z}_t^i) \right) + \nabla_x f_i(\tilde{z}_t^i) - \nabla_x f_i(x^\star, y^\star) \right\|^2$$

$$- 2s \sum_{i=1}^m V_{f_i, y_t^i}(x^\star, x_t^i) + 2s \left( F(\mathbf{1}x^\star, y_t) - F(z_t) + F(x_t, \mathbf{1}y^\star) - F(\mathbf{1}z^\star) \right). \quad (193)$$

Now we bound the second term on the r.h.s. of (193) in terms of $\|x_t - \mathbf{1}x^\star\|^2$ and $\|y_t - \mathbf{1}y^\star\|^2$ as follows:

$$s^2 \sum_{i=1}^{m} E \left\| \frac{1}{np_{il}} \left( \nabla_x f_{il}(z_t^i) - \nabla_x f_{il}(\tilde{z}_t^i) \right) + \nabla_x f_i(\tilde{z}_t^i) - \nabla_x f_i(x^\star, y^\star) \right\|^2$$

$$= s^2 \sum_{i=1}^{m} \sum_{j=1}^{n} p_{ij} \left\| \frac{1}{np_{ij}} \left( \nabla_x f_{ij}(z_t^i) - \nabla_x f_{ij}(\tilde{z}_t^i) \right) + \nabla_x f_i(\tilde{z}_t^i) - \nabla_x f_i(x^\star, y^\star) \right\|^2$$

$$= s^2 \sum_{i=1}^{m} \sum_{j=1}^{n} p_{ij} \left\| \frac{\nabla_x f_{ij}(z_t^i) - \nabla_x f_{ij}(z^\star)}{np_{ij}} + \frac{\nabla_x f_{ij}(z^\star) - \nabla_x f_{ij}(\tilde{z}_t^i)}{np_{ij}} + \nabla_x f_i(\tilde{z}_t^i) - \nabla_x f_i(z^\star) \right\|^2$$

$$\leq 2s^2 \sum_{i=1}^{m} \sum_{j=1}^{n} \frac{p_{ij}}{n^2 p_{ij}^2} \left\| \nabla_x f_{ij}(z_t^i) - \nabla_x f_{ij}(z^\star) \right\|^2$$

$$+ 2s^2 \sum_{i=1}^{m} \sum_{j=1}^{n} p_{ij} \left\| \frac{\nabla_x f_{ij}(z^\star) - \nabla_x f_{ij}(\tilde{z}_t^i)}{np_{ij}} + \nabla_x f_i(\tilde{z}_t^i) - \nabla_x f_i(z^\star) \right\|^2$$

$$= \frac{2s^2}{n^2} \sum_{i=1}^{m} \sum_{j=1}^{n} \frac{1}{p_{ij}} \left\| \nabla_x f_{ij}(z_t^i) - \nabla_x f_{ij}(z^\star) \right\|^2$$

$$+ 2s^2 \sum_{i=1}^{m} \sum_{j=1}^{n} p_{ij} \left\| \frac{\nabla_x f_{ij}(\tilde{z}_t^i) - \nabla_x f_{ij}(z^\star)}{np_{ij}} - \left( \nabla_x f_i(\tilde{z}_t^i) - \nabla_x f_i(z^\star) \right) \right\|^2$$

$$\leq \frac{2s^2}{n^2 p_{\min}} \sum_{i=1}^{m} \sum_{j=1}^{n} \left\| \nabla_x f_{ij}(z_t^i) - \nabla_x f_{ij}(z^\star) \right\|^2$$

$$+ 2s^2 \sum_{i=1}^{m} E \left\| \frac{\nabla_x f_{ij}(\tilde{z}_t^i) - \nabla_x f_{ij}(z^\star)}{np_{ij}} - \left( \nabla_x f_i(\tilde{z}_t^i) - \nabla_x f_i(z^\star) \right) \right\|^2, \qquad (194)$$

where $p_{\min} = \min_{i,j} \{p_{ij}\}$. Let $u_i = \left\{ \frac{\nabla_x f_{il}(\tilde{z}_t^i) - \nabla_x f_{il}(z^\star)}{np_{il}} : l \in \{1, 2, \ldots, n\} \right\}$ be a random variable with probability distribution $\mathcal{P}_i = \{p_{il} : l \in \{1, 2, \ldots, n\}\}$.

$$E[u_i] = E \left[ \frac{\nabla_x f_{il}(\tilde{z}_t^i) - \nabla_x f_{il}(z^\star)}{np_{il}} \right]$$

$$= \sum_{l=1}^{n} \frac{\nabla_x f_{il}(\tilde{z}_t^i) - \nabla_x f_{il}(z^\star)}{np_{il}} p_{il}$$

$$= \frac{1}{n} \sum_{l=1}^{n} \nabla_x f_{il}(\tilde{z}_t^i) - \frac{1}{n} \sum_{l=1}^{n} \nabla_x f_{il}(z^\star)$$

$$= \nabla_x f_i(\tilde{z}_t^i) - \nabla_x f_i(z^\star). \qquad (195)$$

We know that $E\|u_i - Eu_i\|^2 \leq E\|u_i\|^2$. Therefore,

$$E \left\| \frac{\nabla_x f_{ij}(\tilde{z}_t^i) - \nabla_x f_{ij}(z^\star)}{np_{ij}} - \left( \nabla_x f_i(\tilde{z}_t^i) - \nabla_x f_i(z^\star) \right) \right\|^2$$

$$\leq E \left\| \frac{\nabla_x f_{ij}(\tilde{z}_t^i) - \nabla_x f_{ij}(z^\star)}{np_{ij}} \right\|^2$$

$$= \frac{1}{n^2} \sum_{j=1}^{n} \left\| \frac{\nabla_x f_{ij}(\tilde{z}_t^i) - \nabla_x f_{ij}(z^\star)}{p_{ij}} \right\|^2 p_{ij}$$

$$= \frac{1}{n^2} \sum_{j=1}^{n} \frac{1}{p_{ij}} \left\| \nabla_x f_{ij}(\tilde{z}_t^i) - \nabla_x f_{ij}(z^\star) \right\|^2$$

$$\leq \frac{1}{n^2 p_{\min}} \sum_{j=1}^{n} \left\| \nabla_x f_{ij}(\tilde{z}_t^i) - \nabla_x f_{ij}(z^\star) \right\|^2. \qquad (196)$$

By substituting the above inequality in (194), we obtain

$$
s^2 \sum_{i=1}^{m} E \left\| \frac{1}{np_{il}} \left( \nabla_x f_{il}(z_t^i) - \nabla_x f_{il}(\tilde{z}_t^i) \right) + \nabla_x f_i(\tilde{z}_t^i) - \nabla_x f_i(z^\star) \right\|^2
$$

$$
\leq \frac{2s^2}{n^2 p_{\min}} \sum_{i=1}^{m} \sum_{j=1}^{n} \left\| \nabla_x f_{ij}(z_t^i) - \nabla_x f_{ij}(z^\star) \right\|^2 + \frac{2s^2}{n^2 p_{\min}} \sum_{i=1}^{m} \sum_{j=1}^{n} \left\| \nabla_x f_{ij}(\tilde{z}_t^i) - \nabla_x f_{ij}(z^\star) \right\|^2.
$$

(197)

Substituting this inequality in (193) we obtain (186).

### G.1.2 Finishing the Proof of Lemma G.5

We now compute upper bounds on the last two terms present in (189) using smoothness assumptions. First, observe that

$$
\left\| \nabla_x f_{ij}(z_t^i) - \nabla_x f_{ij}(z^\star) \right\|^2 + \left\| \nabla_y f_{ij}(z_t^i) - \nabla_y f_{ij}(z^\star) \right\|^2
$$

$$
= \left\| \nabla_x f_{ij}(z_t^i) - \nabla_x f_{ij}(x^\star, y_t^i) + \nabla_x f_{ij}(x^\star, y_t^i) - \nabla_x f_{ij}(z^\star) \right\|^2
$$

$$
+ \left\| \nabla_y f_{ij}(z_t^i) - \nabla_y f_{ij}(x_t^i, y^\star) + \nabla_y f_{ij}(x_t^i, y^\star) - \nabla_y f_{ij}(z^\star) \right\|^2
$$

$$
\leq 2 \left\| -\nabla_x f_{ij}(z_t^i) + \nabla_x f_{ij}(x^\star, y_t^i) \right\|^2 + 2 \left\| \nabla_x f_{ij}(x^\star, y_t^i) - \nabla_x f_{ij}(z^\star) \right\|^2
$$

$$
+ 2 \left\| \nabla_y f_{ij}(z_t^i) - \nabla_y f_{ij}(x_t^i, y^\star) \right\|^2 + 2 \left\| \nabla_y f_{ij}(x_t^i, y^\star) - \nabla_y f_{ij}(z^\star) \right\|^2
$$

$$
\leq 4 L_{xx} V_{f_{ij}, y_t^i}(x^\star, x_t^i) + 2 L_{xy}^2 \left\| y_t^i - y^\star \right\|^2 + 4 L_{yy} V_{-f_{ij}, x_t^i}(y^\star, y_t^i) + 2 L_{yx}^2 \left\| x_t^i - x^\star \right\|^2, \quad (198)
$$

where the last inequality follows from Proposition C.4, Proposition C.5 and Assumptions G.3-G.4. Adding up the above inequality for $j = 1$ to $n$ and using (27)-(28), we obtain

$$
\sum_{j=1}^{n} \left( \left\| \nabla_x f_{ij}(z_t^i) - \nabla_x f_{ij}(z^\star) \right\|^2 + \left\| \nabla_y f_{ij}(z_t^i) - \nabla_y f_{ij}(z^\star) \right\|^2 \right)
$$

$$
\leq 4 L_{xx} \sum_{j=1}^{n} V_{f_{ij}, y_t^i}(x^\star, x_t^i) + 4 L_{yy} \sum_{j=1}^{n} V_{-f_{ij}, x_t^i}(y^\star, y_t^i) + 2n L_{xy}^2 \left\| y_t^i - y^\star \right\|^2 + 2n L_{yx}^2 \left\| x_t^i - x^\star \right\|^2
$$

$$
= 4 L_{xx} \sum_{j=1}^{n} \left( f_{ij}(x^\star, y_t^i) - f_{ij}(x_t^i, y_t^i) - \left\langle \nabla_x f_{ij}(x_t^i, y_t^i), x^\star - x_t^i \right\rangle \right)
$$

$$
+ 4 L_{yy} \sum_{j=1}^{n} \left( -f_{ij}(x_t^i, y^\star) + f_{ij}(x_t^i, y_t^i) - \left\langle -\nabla_y f_{ij}(x_t^i, y_t^i), y^\star - y_t^i \right\rangle \right)
$$

$$
+ 2n L_{xy}^2 \left\| y_t^i - y^\star \right\|^2 + 2n L_{yx}^2 \left\| x_t^i - x^\star \right\|^2
$$

$$
= 4 L_{xx} \left( n f_i(x^\star, y_t^i) - n f_i(x_t^i, y_t^i) - \left\langle n \nabla_x f_i(x_t^i, y_t^i), x^\star - x_t^i \right\rangle \right)
$$

$$
+ 4 L_{yy} \left( -n f_i(x_t^i, y^\star) + n f_i(x_t^i, y_t^i) - \left\langle -n \nabla_y f_i(x_t^i, y_t^i), y^\star - y_t^i \right\rangle \right) + 2n L_{xy}^2 \left\| y_t^i - y^\star \right\|^2 + 2n L_{yx}^2 \left\| x_t^i - x^\star \right\|^2
$$

$$
= 4n L_{xx} V_{f_i, y_t^i}(x^\star, x_t^i) + 4n L_{yy} V_{-f_i, x_t^i}(y^\star, y_t^i) + 2n L_{xy}^2 \left\| y_t^i - y^\star \right\|^2 + 2n L_{yx}^2 \left\| x_t^i - x^\star \right\|^2,
$$

(199)

where the second last step follows from the structure of $f_i(x, y) = \frac{1}{n} \sum_{j=1}^{n} f_{ij}(x, y)$. Therefore,

$$
\frac{2s^2}{n^2 p_{\min}} \sum_{i=1}^{m} \sum_{j=1}^{n} \left( \left\| \nabla_x f_{ij}(z_t^i) - \nabla_x f_{ij}(z^\star) \right\|^2 + \left\| \nabla_y f_{ij}(z_t^i) - \nabla_y f_{ij}(z^\star) \right\|^2 \right)
$$

$$
\leq \frac{8s^2 L_{xx}}{n p_{\min}} \sum_{i=1}^{m} V_{f_i, y_t^i}(x^\star, x_t^i) + \frac{8s^2 L_{yy}}{n p_{\min}} \sum_{i=1}^{m} V_{-f_i, x_t^i}(y^\star, y_t^i) + \frac{4s^2 L_{xy}^2}{n p_{\min}} \left\| y_t - \mathbf{1} y^\star \right\|^2 + \frac{4s^2 L_{yx}^2}{n p_{\min}} \left\| x_t - \mathbf{1} x^\star \right\|^2.
$$

(200)

Similarly, we bound the last term of (189) as

$$
\frac{2s^2}{n^2 p_{\min}} \sum_{i=1}^{m} \sum_{j=1}^{n} \left( \left\| \nabla_x f_{ij}(\tilde{z}_t^i) - \nabla_x f_{ij}(z^\star) \right\|^2 + \left\| \nabla_y f_{ij}(\tilde{z}_t^i) - \nabla_y f_{ij}(z^\star) \right\|^2 \right)
$$

$$
\leq \frac{4s^2 (L_{xx}^2 + L_{yx}^2)}{n p_{\min}} \left\| \tilde{x}_t - \mathbf{1} x^\star \right\|^2 + \frac{4s^2 (L_{yy}^2 + L_{xy}^2)}{n p_{\min}} \left\| \tilde{y}_t - \mathbf{1} y^\star \right\|^2.
$$

(201)

On substituting (200) and (201) in (189), we obtain

$$E \left\| x_t - \mathbf{1}x^\star - s\mathcal{G}_t^x + s\nabla_x F(\mathbf{1}x^\star, \mathbf{1}y^\star) \right\|^2 + E \left\| y_t - \mathbf{1}y^\star + s\mathcal{G}_t^y - s\nabla_y F(\mathbf{1}x^\star, \mathbf{1}y^\star) \right\|^2$$

$$\leq \left( 1 - \mu_x s + \frac{4s^2 L_{yx}^2}{np_{\min}} \right) \left\| x_t - \mathbf{1}x^\star \right\|^2 + \left( 1 - s\mu_y + \frac{4s^2 L_{xy}^2}{np_{\min}} \right) \left\| y_t - \mathbf{1}y^\star \right\|^2$$

$$- \left( 2s - \frac{8s^2 L_{xx}}{np_{\min}} \right) \sum_{i=1}^m V_{f_i, y_t^i}(x^\star, x_t^i) - \left( 2s - \frac{8s^2 L_{yy}}{np_{\min}} \right) \sum_{i=1}^m V_{-f_i, x_t^i}(y^\star, y_t^i)$$

$$+ \frac{4s^2(L_{xx}^2 + L_{yx}^2)}{np_{\min}} \left\| \tilde{x}_t - \mathbf{1}x^\star \right\|^2 + \frac{4s^2(L_{yy}^2 + L_{xy}^2)}{np_{\min}} \left\| \tilde{y}_t - \mathbf{1}y^\star \right\|^2, \tag{185}$$

proving Lemma G.5.
We now have the following corollary.

**Corollary G.6.** *Let* $s = \frac{\mu n p_{\min}}{24L^2}$. *Then under the settings of Lemma G.5,*

$$E \left\| x_t - \mathbf{1}x^\star - s\mathcal{G}_t^x + s\nabla_x F(\mathbf{1}x^\star, \mathbf{1}y^\star) \right\|^2 + E \left\| y_t - \mathbf{1}y^\star + s\mathcal{G}_t^y - s\nabla_y F(\mathbf{1}x^\star, \mathbf{1}y^\star) \right\|^2 \tag{202}$$

$$\leq \left( 1 - \mu_x s + \frac{4s^2 L_{yx}^2}{np_{\min}} \right) \left\| x_t - \mathbf{1}x^\star \right\|^2 + \left( 1 - s\mu_y + \frac{4s^2 L_{xy}^2}{np_{\min}} \right) \left\| y_t - \mathbf{1}y^\star \right\|^2 \tag{203}$$

$$+ \frac{4s^2(L_{xx}^2 + L_{yx}^2)}{np_{\min}} \left\| \tilde{x}_t - \mathbf{1}x^\star \right\|^2 + \frac{4s^2(L_{yy}^2 + L_{xy}^2)}{np_{\min}} \left\| \tilde{y}_t - \mathbf{1}y^\star \right\|^2. \tag{204}$$

## G.2 Proof of Corollary G.6

From the statement of the corollary, we have $s = \frac{\mu n p_{\min}}{24L^2} \leq \frac{np_{\min}}{24L\kappa} < \frac{np_{\min}}{4L} \leq \frac{np_{\min}}{4L_{xx}}$. This implies that

$$\frac{4sL_{xx}}{np_{\min}} \leq 1 \text{ i.e., } \frac{8s^2 L_{xx}}{np_{\min}} \leq 2s. \tag{205}$$

Notice that $V_{f_i, y_t^i}(x^\star, x_t^i) \geq 0$. Therefore, $\left( 2s - \frac{8s^2 L_{xx}}{np_{\min}} \right) \sum_{i=1}^m V_{f_i, y_t^i}(x^\star, x_t^i) \geq 0$. We also have $s \leq \frac{np_{\min}}{4L_{yy}}$ because $L = \max\{L_{xx}, L_{yy}, L_{xy}, L_{yx}\}$. Therefore, $\frac{8s^2 L_{yy}}{np_{\min}} \leq 2s$. Due to the concavity of $f_i(x,y)$ in $y$, $V_{-f_i, x_t^i}(y^\star, y_t^i)$ is nonnegative. Therefore, $\left( 2s - \frac{8s^2 L_{yy}}{np_{\min}} \right) \sum_{i=1}^m V_{-f_i, x_t^i}(y^\star, y_t^i) \geq 0$.
By substituting these lower bounds in (185), we get the desired result.

**Parameters setup**
Let $p_{\min} = \min_{i,j}\{p_{ij}\}$. We define the following quantities which are instrumental in simplifying the bounds and in Algorithm 3 implementation.

$$\tilde{c}_x := \frac{8s^2(L_{xx}^2 + L_{yx}^2)}{np_{\min}p}, \ \tilde{c}_y := \frac{8s^2(L_{yy}^2 + L_{xy}^2)}{np_{\min}p}, \tag{206}$$

$$b_x := s\mu_x - \frac{4s^2 L_{yx}^2}{np_{\min}} - \tilde{c}_x p, b_y := s\mu_y - \frac{4s^2 L_{xy}^2}{np_{\min}} - \tilde{c}_y p, \tag{207}$$

$$\alpha_x := \frac{b_x}{(1+\delta)}, \ \alpha_y := \frac{b_y}{(1+\delta)}, \tag{208}$$

$$\gamma_x := \min\left\{ \frac{b_x}{4\sqrt{\delta}(1+\delta)\lambda_{\max}(I-W)}, \frac{1}{4(1+\delta)\lambda_{\max}(I-W)} \right\}, \tag{209}$$

$$\gamma_y := \min\left\{ \frac{b_y}{4\sqrt{\delta}(1+\delta)\lambda_{\max}(I-W)}, \frac{1}{4(1+\delta)\lambda_{\max}(I-W)} \right\}, \tag{210}$$

$$\tag{211}$$

$$\Phi_t := M_x \|x_t - \mathbf{1}x^\star\|^2 + \frac{2s^2}{\gamma_x} \|D_t^x - D_x^\star\|_{(I-W)^\dagger}^2 + \sqrt{\delta} \|H_t^x - H_x^\star\|^2 \tag{212}$$

$$+ M_y \|y_t - \mathbf{1}y^\star\|^2 + \frac{2s^2}{\gamma_y} \|D_t^y - D_y^\star\|_{(I-W)^\dagger}^2 + \sqrt{\delta} \|H_t^y - H_y^\star\|^2, \tag{213}$$

$$\rho = \max \left\{ \frac{1-b_x}{M_x}, \frac{1-b_y}{M_y}, 1 - \frac{\gamma_x}{2}\lambda_{m-1}(I-W), 1 - \frac{\gamma_y}{2}\lambda_{m-1}(I-W), 1 - \alpha_x, 1 - \alpha_y, 1 - \frac{p}{2} \right\} \tag{214}$$

$$\tilde{\Phi}_t = \Phi_t + \tilde{c}_x \|\tilde{x}_t - \mathbf{1}x^\star\|^2 + \tilde{c}_y \|\tilde{y}_t - \mathbf{1}y^\star\|^2. \tag{215}$$

It is worth mentioning that $\gamma_x$ and $\gamma_y$ are well defined for $\delta = 0$.

**Lemma G.7.** *Parameters Feasibility The parameters defined in* (207), (208), (209) *and* (210) *satisfy the followings:*

$$b_x \in (0,1), \ b_y \in (0,1), \tag{216}$$

$$\alpha_x < \min \left\{ \frac{b_x}{\sqrt{\delta}}, \frac{1}{1+\delta} \right\}, \ \alpha_y < \min \left\{ \frac{b_y}{\sqrt{\delta}}, \frac{1}{1+\delta} \right\} \tag{217}$$

$$\gamma_x \in \left( 0, \min \left\{ \frac{2 - 2\sqrt{\delta}\alpha_x}{\lambda_{\max}(I-W)}, \frac{\alpha_x - (1+\delta)\alpha_x^2}{\sqrt{\delta}\lambda_{\max}(I-W)} \right\} \right), \tag{218}$$

$$\gamma_y \in \left( 0, \min \left\{ \frac{2 - 2\sqrt{\delta}\alpha_y}{\lambda_{\max}(I-W)}, \frac{\alpha_y - (1+\delta)\alpha_y^2}{\sqrt{\delta}\lambda_{\max}(I-W)} \right\} \right), \tag{219}$$

$$\frac{\gamma_x}{2}\lambda_{m-1}(I-W) \in (0,1), \ \frac{\gamma_y}{2}\lambda_{m-1}(I-W) \in (0,1), \tag{220}$$

$$M_x \in (0,1), \ M_y \in (0,1), \tag{221}$$

$$\frac{1-b_x}{M_x} \in (0,1), \ \frac{1-b_y}{M_y} \in (0,1). \tag{222}$$

*Moreover,*

$$M_x \geq 1 - \frac{8b_x\sqrt{\delta}}{7(1+\delta)} \geq 1 - \frac{4b_x}{7}, \tag{223}$$

$$M_y \geq 1 - \frac{8b_y\sqrt{\delta}}{7(1+\delta)} \geq 1 - \frac{4b_y}{7}, \tag{224}$$

$$\frac{1-b_x}{M_x} < 1 - \frac{3b_x}{7}, \ \frac{1-b_y}{M_y} < 1 - \frac{3b_y}{7}, \tag{225}$$

$$1 - \frac{\gamma_x}{2}\lambda_{m-1}(I-W) = \begin{cases} 1 - \frac{b_x}{8\sqrt{\delta}(1+\delta)\kappa_g} \ ; \ if \, b_x \leq \sqrt{\delta} \\ 1 - \frac{1}{8(1+\delta)\kappa_g} \ ; \ if \, b_x > \sqrt{\delta} \end{cases} . \tag{226}$$

## G.3 Proof of Lemma G.7

In this section, we show that the chosen parameters $\alpha_x\alpha_y, M_x, M_y, \gamma_x$ and $\gamma_y$ satisfy the following conditions given in Lemma G.7:

$$\alpha_x < \min \left\{ \frac{b_x}{\sqrt{\delta}}, \frac{1}{1+\delta} \right\}, \ \alpha_y < \min \left\{ \frac{b_y}{\sqrt{\delta}}, \frac{1}{1+\delta} \right\}$$

$$\gamma_x \in \left( 0, \min \left\{ \frac{2 - 2\sqrt{\delta}\alpha_x}{\lambda_{\max}(I-W)}, \frac{\alpha_x - (1+\delta)\alpha_x^2}{\sqrt{\delta}\lambda_{\max}(I-W)} \right\} \right), \gamma_y \in \left( 0, \min \left\{ \frac{2 - 2\sqrt{\delta}\alpha_y}{\lambda_{\max}(I-W)}, \frac{\alpha_y - (1+\delta)\alpha_y^2}{\sqrt{\delta}\lambda_{\max}(I-W)} \right\} \right),$$

$$\frac{\gamma_x}{2}\lambda_{m-1}(I-W) \in (0,1), \ \frac{\gamma_y}{2}\lambda_{m-1}(I-W) \in (0,1), M_x \in (0,1), \ M_y \in (0,1),$$

$$\frac{1-b_x}{M_x} \in (0,1), \ \frac{1-b_y}{M_y} \in (0,1). \tag{227}$$

We first show that $b_x \in (0,1)$ and $b_y \in (0,1)$. From definition,

$$b_x = s\mu_x - \frac{4s^2 L_{yx}^2}{np_{\min}} - \tilde{c}_x p < s\mu_x = \frac{\mu np_{\min}\mu_x}{24L^2} \leq \frac{np_{\min}\mu_x^2}{24L_{xx}^2} = \frac{np_{\min}}{24\kappa_x^2} \leq \frac{1}{24} < 1. \tag{228}$$

Similarly,

$$b_y = s\mu_y - \frac{4s^2 L_{xy}^2}{np_{\min}} - \tilde{c}_y p < s\mu_y = \frac{\mu np_{\min}\mu_y}{24L^2} \leq \frac{np_{\min}\mu_y^2}{24L_{yy}^2} = \frac{np_{\min}}{24\kappa_y^2} \leq \frac{1}{24} < 1. \tag{229}$$

We now focus on the lower bound on $b_x$ and $b_y$.

$$\begin{aligned}
b_x &= s\mu_x - \frac{4s^2 L_{yx}^2}{np_{\min}} - \tilde{c}_x p \\
&= s\mu_x - \frac{4s^2 L_{yx}^2}{np_{\min}} - \frac{8s^2(L_{xx}^2 + L_{yx}^2)}{np_{\min}} \\
&= s\mu_x - \frac{12s^2 L_{yx}^2}{np_{\min}} - \frac{8s^2 L_{xx}^2}{np_{\min}} \\
&\geq s\mu - \frac{12s^2 L^2}{np_{\min}} - \frac{8s^2 L^2}{np_{\min}} \\
&= s\mu - \frac{20s^2 L^2}{np_{\min}} \\
&= \frac{\mu^2 np_{\min}}{24L^2} - \frac{\mu^2 n^2 p_{\min}^2}{576L^4}\frac{20L^2}{np_{\min}} \\
&= \frac{np_{\min}}{24\kappa_f^2} - \frac{20np_{\min}}{576\kappa_f^2} \\
&= \frac{np_{\min}}{144\kappa_f^2} \\
&> 0.
\end{aligned} \tag{230}$$

$$> 0. \tag{231}$$

In a similar fashion, we get $b_y < 1$ and

$$b_y \geq \frac{np_{\min}}{144\kappa_f^2} > 0. \tag{232}$$

**Feasibility of $\alpha_{\mathbf{x}}$ and $\alpha_y$.**

We have, $0 < b_x < 1$. Therefore, $\alpha_x < \frac{1}{1+\delta}$. Moreover, $\frac{\sqrt{\delta}}{1+\delta} \leq 1/2$. Therefore, $\alpha_x \leq \frac{b_x}{2\sqrt{\delta}} < b_x/\sqrt{\delta}$.
Hence, $\alpha_x < \min\left\{\frac{b_x}{\sqrt{\delta}}, \frac{1}{1+\delta}\right\}$. Similarly, $\alpha_y < \min\left\{\frac{b_y}{\sqrt{\delta}}, \frac{1}{1+\delta}\right\}$ because $b_y \in (0,1)$.

**Feasibility of $\gamma_{x,k}$ and $\gamma_{y,k}$.** We consider two cases to verify the feasibility of $\gamma_x$ and $\gamma_y$.

**Case I:** $b_x \leq \sqrt{\delta}$.
This gives $\gamma_x = \frac{b_x}{4\sqrt{\delta}(1+\delta)\lambda_{\max}(I-W)}$. Consider

$$\frac{\alpha_x - (1+\delta)\alpha_x^2}{\sqrt{\delta}\lambda_{\max}(I-W)} = \frac{b_x - b_x^2}{\sqrt{\delta}(1+\delta)\lambda_{\max}(I-W)}. \tag{233}$$

Using (228), we have $b_x \leq \frac{1}{24\kappa_x^2} < 0.75$. This allows us to use the inequality $2x - 2x^2 \geq x/2$ for all $0 \leq x \leq 0.75$. Therefore,

$$\begin{aligned}
\frac{\alpha_x - (1+\delta)\alpha_x^2}{\sqrt{\delta}\lambda_{\max}(I-W)} &> \frac{b_x}{4\sqrt{\delta}(1+\delta)\lambda_{\max}(I-W)} \\
&= \gamma_{x,k}.
\end{aligned} \tag{234}$$

We also have

$$\begin{aligned}
\frac{2 - 2\sqrt{\delta}\alpha_x}{\lambda_{\max}(I-W)} &= \left(2 - \frac{2\sqrt{\delta}b_x}{1+\delta}\right)\frac{1}{\lambda_{\max}(I-W)} \geq \left(2 - \frac{2\sqrt{\delta}}{1+\delta}\right)\frac{1}{\lambda_{\max}(I-W)} \\
&\geq \frac{1}{\lambda_{\max}(I-W)} > \frac{1}{4(1+\delta)\lambda_{\max}(I-W)} \\
&> \frac{b_x}{4\sqrt{\delta}(1+\delta)\lambda_{\max}(I-W)} \\
&= \gamma_{x,k},
\end{aligned} \tag{235}$$

where the second last inequality uses $b_x \leq \sqrt{\delta}$. We know that $b_y \in (0,1)$. Therefore, by following similar steps, the chosen $\gamma_y$ is also feasible.

**Case II:** $b_x > \sqrt{\delta}$

This give $\gamma_x = \frac{1}{4(1+\delta)\lambda_{\max}(I-W)}$.

$$\frac{\alpha_x - (1+\delta)\alpha_x^2}{\sqrt{\delta}\lambda_{\max}(I-W)} = \frac{b_x - b_x^2}{\sqrt{\delta}(1+\delta)\lambda_{\max}(I-W)}$$
$$\geq \frac{b_x}{4\sqrt{\delta}(1+\delta)\lambda_{\max}(I-W)}$$
$$> \frac{1}{4(1+\delta)\lambda_{\max}(I-W)}$$
$$= \gamma_x. \tag{236}$$

Consider

$$\frac{2 - 2\sqrt{\delta}\alpha_x}{\lambda_{\max}(I-W)} = \left(2 - \frac{2\sqrt{\delta}b_x}{1+\delta}\right)\frac{1}{\lambda_{\max}(I-W)}$$
$$\geq \left(2 - \frac{2\sqrt{\delta}}{1+\delta}\right)\frac{1}{\lambda_{\max}(I-W)}$$
$$\geq \frac{1}{\lambda_{\max}(I-W)}$$
$$> \frac{1}{4(1+\delta)\lambda_{\max}(I-W)}$$
$$= \gamma_x. \tag{237}$$

Therefore, $\gamma_x < \min\left\{\frac{\alpha_x - (1+\delta)\alpha_x^2}{\sqrt{\delta}\lambda_{\max}(I-W)}, \frac{2-2\sqrt{\delta}\alpha_x}{\lambda_{\max}(I-W)}\right\}$.

As $\gamma_x < \frac{2-2\sqrt{\delta}\alpha_x}{\lambda_{\max}(I-W)} < \frac{2}{\lambda_{\max}(I-W)}$. Notice that $\lambda_{m-1}(I-W) < \lambda_{\max}(I-W)$ Therefore,

$$\frac{\gamma_x}{2}\lambda_{m-1}(I-W) < \frac{\gamma_x}{2}\lambda_{\max}(I-W) < 1. \tag{238}$$

Similarly, $\frac{\gamma_y}{2}\lambda_{m-1}(I-W) < 1$.

**Feasibility of $M_x$ and $M_y$.**

Recall $M_x = 1 - \frac{\sqrt{\delta}\alpha_x}{1 - \frac{\gamma_x}{2}\lambda_{\max}(I-W)}$ and $M_y = 1 - \frac{\sqrt{\delta}\alpha_y}{1 - \frac{\gamma_y}{2}\lambda_{\max}(I-W)}$. We have

$$\gamma_x < \frac{2 - 2\sqrt{\delta}\alpha_x}{\lambda_{\max}(I-W)}$$
$$\frac{\gamma_x\lambda_{\max}(I-W)}{2} < 1 - \sqrt{\delta}\alpha_x$$
$$1 - \frac{\gamma_x\lambda_{\max}(I-W)}{2} > \sqrt{\delta}\alpha_x$$
$$\frac{\sqrt{\delta}\alpha_x}{1 - \frac{\gamma_x\lambda_{\max}(I-W)}{2}} < 1. \tag{239}$$

Moreover, $\frac{\sqrt{\delta}\alpha_x}{1 - \frac{\gamma_x\lambda_{\max}(I-W)}{2}} > 0$. Therefore, $M_x \in (0, 1)$. Similar steps follow to prove the feasibility of $M_y$.

**Feasibility of $\frac{1-b_x}{M_x}$ and $\frac{1-b_y}{M_y}$.**

We derive upper bounds on $\frac{1-b_x}{M_x}$ and $\frac{1-b_y}{M_y}$ to verify the feasibility. We divide the derivation into two cases.

**Case I:** $b_x \leq \sqrt{\delta}$

This implies that

$$\gamma_x = \frac{b_x}{4\sqrt{\delta}(1+\delta)\lambda_{\max}(I-W)} \tag{240}$$

$$\frac{\gamma_x}{2}\lambda_{\max}(I-W) = \frac{b_x}{8\sqrt{\delta}(1+\delta)}. \tag{241}$$

Recall $M_x$:

$$M_x = 1 - \frac{\sqrt{\delta}\alpha_x}{1 - \frac{\gamma_x}{2}\lambda_{\max}(I - W)}$$

$$= 1 - \frac{\frac{\sqrt{\delta}b_x}{1+\delta}}{1 - \frac{b_x}{8\sqrt{\delta}(1+\delta)}}$$

$$= 1 - \frac{\sqrt{\delta}b_x \times 8\sqrt{\delta}(1 + \delta)}{(1+\delta)\left(8\sqrt{\delta}(1+\delta) - b_x\right)}$$

$$= 1 - \frac{8\delta b_x}{\left(8\sqrt{\delta}(1+\delta) - b_x\right)}$$

$$= 1 - \frac{8\delta}{\frac{8\sqrt{\delta}(1+\delta)}{b_x} - 1}. \tag{242}$$

We know that $\frac{\sqrt{\delta}}{b_x} \geq 1$. Therefore, $\frac{\sqrt{\delta}(1+\delta)}{b_x} > 1$ which in turn implies that

$$\frac{8\sqrt{\delta}(1+\delta)}{b_x} - 1 > \frac{8\sqrt{\delta}(1+\delta)}{b_x} - \frac{\sqrt{\delta}(1+\delta)}{b_x}$$

$$= \frac{7\sqrt{\delta}(1+\delta)}{b_x}$$

$$\frac{1}{\frac{8\sqrt{\delta}(1+\delta)}{b_x} - 1} < \frac{b_x}{7\sqrt{\delta}(1+\delta)}. \tag{243}$$

By using above relation in (242), we obtain

$$M_x \geq 1 - \frac{8\delta b_x}{7\sqrt{\delta}(1+\delta)} = 1 - \frac{8b_x\sqrt{\delta}}{7(1+\delta)} \tag{244}$$

$$\geq 1 - \frac{8b_x}{7}\frac{1}{2} = 1 - \frac{4b_x}{7}, \tag{245}$$

where the second last inequality uses $\frac{\sqrt{\delta}}{1+\delta} \leq \frac{1}{2}$.

$$\frac{1 - b_x}{M_x} = 1 + \frac{1 - b_x}{M_x} - 1 \leq 1 + \frac{1 - b_x}{1 - \frac{8b_x\sqrt{\delta}}{7(1+\delta)}} - 1$$

$$= 1 + \frac{1 - b_x - 1 + \frac{8b_x\sqrt{\delta}}{7(1+\delta)}}{1 - \frac{8b_x\sqrt{\delta}}{7(1+\delta)}} = 1 - \frac{b_x - \frac{8b_x\sqrt{\delta}}{7(1+\delta)}}{1 - \frac{8b_x\sqrt{\delta}}{7(1+\delta)}}$$

$$= 1 - \frac{7b_x(1+\delta) - 8b_x\sqrt{\delta}}{7(1+\delta) - 8b_x\sqrt{\delta}} = 1 - \frac{7(1+\delta) - 8\sqrt{\delta}}{\frac{7(1+\delta)}{b_x} - 8\sqrt{\delta}}$$

$$\leq 1 - \frac{7(1+\delta) - \frac{8(1+\delta)}{2}}{\frac{7(1+\delta)}{b_x} - 8\sqrt{\delta}} = 1 - \frac{3(1+\delta)}{\frac{7(1+\delta)}{b_x} - 8\sqrt{\delta}}$$

$$< 1 - \frac{3(1+\delta)}{\frac{7(1+\delta)}{b_x}}$$

$$= 1 - \frac{3b_x}{7}$$

$$< 1. \tag{246}$$

Similarly, we obtain

$$M_y \geq 1 - \frac{8b_y\sqrt{\delta}}{7(1+\delta)} \geq 1 - \frac{4b_y}{7} \text{ and,} \tag{247}$$

$$\frac{1 - b_y}{M_y} < 1 - \frac{3b_y}{7}. \tag{248}$$

**Case II:** $b_x > \sqrt{\delta}$ .

$$\gamma_x = \frac{1}{4(1+\delta)\lambda_{\max}(I-W)}. \tag{249}$$

We have

$$M_x = 1 - \frac{\sqrt{\delta}\alpha_x}{1 - \frac{\gamma_x}{2}\lambda_{\max}(I-W)}$$

$$= 1 - \frac{\sqrt{\delta}\alpha_x}{1 - \frac{1}{8(1+\delta)}}$$

$$= 1 - \frac{\frac{\sqrt{\delta}b_x}{1+\delta}}{1 - \frac{1}{8(1+\delta)}}$$

$$= 1 - \frac{\sqrt{\delta}b_x \times 8(1+\delta)}{(1+\delta)(8(1+\delta)-1)}$$

$$= 1 - \frac{8\sqrt{\delta}b_x}{8(1+\delta)-1}. \tag{250}$$

As $8(1+\delta) - 1 > 8(1+\delta) - 1 - \delta = 7(1+\delta)$. Therefore,

$$M_x \geq 1 - \frac{8\sqrt{\delta}b_x}{7(1+\delta)}. \tag{251}$$

Notice that above lower bound matches with lower bound in (244). Therefore, by following steps similar to Case I, we obtain

$$\frac{1-b_x}{M_x} < 1 - \frac{3b_x}{7}, \text{ and} \tag{252}$$

$$\frac{1-b_y}{M_y} < 1 - \frac{3b_y}{7}. \tag{253}$$

**Feasibility of** $1 - \frac{\gamma_x}{2}\lambda_{m-1}(I-W)$ **and** $1 - \frac{\gamma_y}{2}\lambda_{m-1}(I-W)$**.**
If $b_x \leq \sqrt{\delta}$, then $\gamma_x = \frac{b_x}{4\sqrt{\delta}(1+\delta)\lambda_{\max}(I-W)}$.

$$1 - \frac{\gamma_x}{2}\lambda_{m-1}(I-W) = 1 - \frac{b_x}{8\sqrt{\delta}(1+\delta)\lambda_{\max}(I-W)}\lambda_{m-1}(I-W)$$

$$= 1 - \frac{b_x}{8\sqrt{\delta}(1+\delta)\kappa_g}. \tag{254}$$

If $b_x > \sqrt{\delta}$, then $\gamma_x = \frac{1}{4(1+\delta)\lambda_{\max}(I-W)}$.

$$1 - \frac{\gamma_x}{2}\lambda_{m-1}(I-W) = 1 - \frac{1}{8(1+\delta)\lambda_{\max}(I-W)}\lambda_{m-1}(I-W) \tag{255}$$

$$= 1 - \frac{1}{8(1+\delta)\kappa_g}. \tag{256}$$

We now have a result on recursive relationship on $E\left[\tilde{\Phi}_t\right]$.

**Lemma G.8.** *Let* $\{x_t\}_t, \{y_t\}_t$ *be the sequences generated by Algorithm 3. Suppose Assumptions 3.1-3.5 and Assumptions G.1-G.4 hold. Then for every* $t \geq 1$*:*

$$E\left[\tilde{\Phi}_t\right] \leq \rho^t\tilde{\Phi}_0, \tag{257}$$

*where* $\rho$ *is defined in equation* (214).

### G.4 Proof of Lemma G.8

Iterates $\{x_t, y_t\}$ of Algorithm 3 are obtained by calling Algorithm 1 at iterate $t-1$. Therefore, Lemma D.1 and Lemma D.2 also holds for Algorithm 3. Adding inequalities (53) and (103) (Lemma

D.1 and Lemma D.2), we have

$$M_x E \left\| x_{t+1} - \mathbf{1}x^\star \right\|^2 + \frac{2s^2}{\gamma_x} E \left\| D^x_{t+1} - D^\star_x \right\|^2_{(I-W)^\dagger} + \sqrt{\delta} E \left\| H^x_{t+1} - H^\star_x \right\|^2 +$$

$$+ M_y E \left\| y_{t+1} - \mathbf{1}y^\star \right\|^2 + \frac{2s^2}{\gamma_y} E \left\| D^y_{t+1} - D^\star_y \right\|^2_{(I-W)^\dagger} + \sqrt{\delta} E \left\| H^y_{t+1} - H^\star_y \right\|^2$$

$$\leq \left\| x_t - \mathbf{1}x^\star - s\mathcal{G}^x_t + s\nabla_x F(\mathbf{1}z^\star) \right\|^2 + \frac{2s^2}{\gamma_x} \left( 1 - \frac{\gamma_y}{2}\lambda_{m-1}(I-W) \right) \left\| D^x_t - D^\star_x \right\|^2_{(I-W)^\dagger}$$

$$+ \sqrt{\delta}(1-\alpha_x) \left\| H^x_t - H^\star_x \right\|^2$$

$$+ \left\| y_t - \mathbf{1}y^\star + s\mathcal{G}^y_t - s\nabla_y F(\mathbf{1}z^\star) \right\|^2 + \frac{2s^2}{\gamma_y} \left( 1 - \frac{\gamma_y}{2}\lambda_{m-1}(I-W) \right) \left\| D^y_t - D^\star_y \right\|^2_{(I-W)^\dagger}$$

$$+ \sqrt{\delta}(1-\alpha_y) \left\| H^y_t - H^\star_y \right\|^2. \tag{258}$$

By the definition of $\Phi_t$, the above inequality can be rewritten as

$$E\left[\Phi_{t+1}\right]$$

$$\leq \left\| x_t - \mathbf{1}x^\star - s\mathcal{G}^x_t + s\nabla_x F(\mathbf{1}z^\star) \right\|^2 + \frac{2s^2}{\gamma_x} \left( 1 - \frac{\gamma_y}{2}\lambda_{m-1}(I-W) \right) \left\| D^x_t - D^\star_x \right\|^2_{(I-W)^\dagger}$$

$$+ \sqrt{\delta}(1-\alpha_x) \left\| H^x_t - H^\star_x \right\|^2$$

$$+ \left\| y_t - \mathbf{1}y^\star + s\mathcal{G}^y_t - s\nabla_y F(\mathbf{1}z^\star) \right\|^2 + \frac{2s^2}{\gamma_y} \left( 1 - \frac{\gamma_y}{2}\lambda_{m-1}(I-W) \right) \left\| D^y_t - D^\star_y \right\|^2_{(I-W)^\dagger}$$

$$+ \sqrt{\delta}(1-\alpha_y) \left\| H^y_t - H^\star_y \right\|^2. \tag{259}$$

Taking conditional expectation on stochastic gradient at $t$-th step on both sides of above inequality and applying Tower property, we obtain

$$E\left[\Phi_{t+1}\right]$$

$$\leq E \left\| x_t - \mathbf{1}x^\star - s\mathcal{G}^x_t + s\nabla_x F(\mathbf{1}z^\star) \right\|^2 + \frac{2s^2}{\gamma_x} \left( 1 - \frac{\gamma_x}{2}\lambda_{m-1}(I-W) \right) \left\| D^x_t - D^\star_x \right\|^2_{(I-W)^\dagger}$$

$$+ \sqrt{\delta}(1-\alpha_x) \left\| H^x_t - H^\star_x \right\|^2$$

$$+ E \left\| y_t - \mathbf{1}y^\star + s\mathcal{G}^y_t - s\nabla_y F(\mathbf{1}z^\star) \right\|^2 + \frac{2s^2}{\gamma_y} \left( 1 - \frac{\gamma_y}{2}\lambda_{m-1}(I-W) \right) \left\| D^y_t - D^\star_y \right\|^2_{(I-W)^\dagger}$$

$$+ \sqrt{\delta}(1-\alpha_y) \left\| H^y_t - H^\star_y \right\|^2$$

$$\leq \left( 1 - \mu_x s + \frac{4s^2 L^2_{yx}}{np_{\min}} \right) \left\| x_t - \mathbf{1}x^\star \right\|^2 + \left( 1 - s\mu_y + \frac{4s^2 L^2_{xy}}{np_{\min}} \right) \left\| y_t - \mathbf{1}y^\star \right\|^2$$

$$+ \frac{4s^2(L^2_{xx} + L^2_{yx})}{np_{\min}} \left\| \tilde{x}_t - \mathbf{1}x^\star \right\|^2 + \frac{4s^2(L^2_{yy} + L^2_{xy})}{np_{\min}} \left\| \tilde{y}_t - \mathbf{1}y^\star \right\|^2$$

$$+ \frac{2s^2}{\gamma_x} \left( 1 - \frac{\gamma_x}{2}\lambda_{m-1}(I-W) \right) \left\| D^x_t - D^\star_x \right\|^2_{(I-W)^\dagger} + \frac{2s^2}{\gamma_y} \left( 1 - \frac{\gamma_y}{2}\lambda_{m-1}(I-W) \right) \left\| D^y_t - D^\star_y \right\|^2_{(I-W)^\dagger}$$

$$+ \sqrt{\delta}(1-\alpha_x) \left\| H^x_t - H^\star_x \right\|^2 + \sqrt{\delta}(1-\alpha_y) \left\| H^y_t - H^\star_y \right\|^2. \tag{260}$$

The last step holds due to Corollary G.6. From SVRG oracle, we have

$$\left\| \tilde{x}^i_{t+1} - x^\star \right\|^2 = \begin{cases} \left\| x^i_t - x^\star \right\|^2 & \text{with probability } p \\ \left\| \tilde{x}^i_t - x^\star \right\|^2 & \text{with probability } 1-p \end{cases}, \text{ and}$$

$$\left\| \tilde{y}^i_{t+1} - y^\star \right\|^2 = \begin{cases} \left\| y^i_t - y^\star \right\|^2 & \text{with probability } p \\ \left\| \tilde{y}^i_t - y^\star \right\|^2 & \text{with probability } 1-p \end{cases}.$$

Therefore,
$$E \left\| \tilde{x}_{t+1} - \mathbf{1}x^\star \right\|^2 + E \left\| \tilde{y}_{t+1} - \mathbf{1}y^\star \right\|^2$$

$$= \sum_{i=1}^{m} E \left\| \tilde{x}_{t+1}^i - x^\star \right\|^2 + \sum_{i=1}^{m} E \left\| \tilde{y}_{t+1}^i - y^\star \right\|^2$$

$$= \sum_{i=1}^{m} \left( p \left\| x_t^i - x^\star \right\|^2 + (1-p) \left\| \tilde{x}_t^i - x^\star \right\|^2 \right) + \sum_{i=1}^{m} \left( p \left\| y_t^i - y^\star \right\|^2 + (1-p) \left\| \tilde{y}_t^i - y^\star \right\|^2 \right)$$

$$= p \left\| x_t - \mathbf{1}x^\star \right\|^2 + (1-p) \left\| \tilde{x}_t - \mathbf{1}x^\star \right\|^2 + p \left\| y_t - \mathbf{1}y^\star \right\|^2 + (1-p) \left\| \tilde{y}_t - \mathbf{1}y^\star \right\|^2. \qquad (261)$$

Using above equality and (260), we obtain
$$E \left[ \Phi_{t+1} \right] + \tilde{c}_x E \left\| \tilde{x}_{t+1} - \mathbf{1}x^\star \right\|^2 + \tilde{c}_y E \left\| \tilde{y}_{t+1} - \mathbf{1}y^\star \right\|^2$$

$$\leq \left( 1 - \mu_x s + \frac{4s^2 L_{yx}^2}{np_{\min}} + \tilde{c}_x p \right) \left\| x_t - \mathbf{1}x^\star \right\|^2 + \left( 1 - s\mu_y + \frac{4s^2 L_{xy}^2}{np_{\min}} + \tilde{c}_y p \right) \left\| y_t - \mathbf{1}y^\star \right\|^2$$

$$+ \left( \tilde{c}_x (1-p) + \frac{4s^2 (L_{xx}^2 + L_{yx}^2)}{np_{\min}} \right) \left\| \tilde{x}_t - \mathbf{1}x^\star \right\|^2 + \left( \tilde{c}_y (1-p) + \frac{4s^2 (L_{yy}^2 + L_{xy}^2)}{np_{\min}} \right) \left\| \tilde{y}_t - \mathbf{1}y^\star \right\|^2$$

$$+ \frac{2s^2}{\gamma_x} \left( 1 - \frac{\gamma_x}{2} \lambda_{m-1}(I-W) \right) \left\| D_t^x - D_x^\star \right\|_{(I-W)^\dagger}^2 + \frac{2s^2}{\gamma_y} \left( 1 - \frac{\gamma_y}{2} \lambda_{m-1}(I-W) \right) \left\| D_t^y - D_y^\star \right\|_{(I-W)^\dagger}^2$$

$$+ \sqrt{\delta}(1 - \alpha_x) \left\| H_t^x - H_x^\star \right\|^2 + \sqrt{\delta}(1 - \alpha_y) \left\| H_t^y - H_y^\star \right\|^2. \qquad (262)$$

We have $\tilde{c}_x = \frac{8s^2 (L_{xx}^2 + L_{yx}^2)}{np_{\min} p}$. The coefficient of $\left\| \tilde{x}_t - \mathbf{1}x^\star \right\|^2$ in (262) is

$$\tilde{c}_x (1-p) + \frac{4s^2 (L_{xx}^2 + L_{yx}^2)}{np_{\min}} = \tilde{c}_x \left( 1 - p + \frac{4s^2 (L_{xx}^2 + L_{yx}^2)}{np_{\min} \tilde{c}_x} \right)$$

$$= \tilde{c}_x \left( 1 - p + \frac{4s^2 (L_{xx}^2 + L_{yx}^2)}{np_{\min}} \frac{np_{\min} p}{8s^2 (L_{xx}^2 + L_{yx}^2)} \right)$$

$$= \tilde{c}_x \left( 1 - p + \frac{p}{2} \right)$$

$$= \tilde{c}_x \left( 1 - \frac{p}{2} \right), \qquad (263)$$

and the coefficient of $\left\| \tilde{y}_t - \mathbf{1}y^\star \right\|^2$ in (262) is

$$\tilde{c}_y (1-p) + \frac{4s^2 (L_{yy}^2 + L_{xy}^2)}{np_{\min}} = \tilde{c}_y \left( 1 - p + \frac{4s^2 (L_{yy}^2 + L_{xy}^2)}{np_{\min} \tilde{c}_y} \right)$$

$$= \tilde{c}_y \left( 1 - p + \frac{4s^2 (L_{yy}^2 + L_{xy}^2)}{np_{\min}} \frac{np_{\min} p}{8s^2 (L_{yy}^2 + L_{xy}^2)} \right)$$

$$= \tilde{c}_y \left( 1 - p + \frac{p}{2} \right)$$

$$= \tilde{c}_y \left( 1 - \frac{p}{2} \right). \qquad (264)$$

Substituting the above simplified coefficients into (262), we see that
$$E \left[ \Phi_{t+1} \right] + \tilde{c}_x E \left\| \tilde{x}_{t+1} - \mathbf{1}x^\star \right\|^2 + \tilde{c}_y E \left\| \tilde{y}_{t+1} - \mathbf{1}y^\star \right\|^2$$

$$\leq \left( 1 - \mu_x s + \frac{4s^2 L_{yx}^2}{np_{\min}} + \tilde{c}_x p \right) \left\| x_t - \mathbf{1}x^\star \right\|^2 + \left( 1 - s\mu_y + \frac{4s^2 L_{xy}^2}{np_{\min}} + \tilde{c}_y p \right) \left\| y_t - \mathbf{1}y^\star \right\|^2$$

$$+ \tilde{c}_x \left( 1 - \frac{p}{2} \right) \left\| \tilde{x}_t - \mathbf{1}x^\star \right\|^2 + \tilde{c}_y \left( 1 - \frac{p}{2} \right) \left\| \tilde{y}_t - \mathbf{1}y^\star \right\|^2$$

$$+ \frac{2s^2}{\gamma_x} \left( 1 - \frac{\gamma_x}{2} \lambda_{m-1}(I-W) \right) \left\| D_t^x - D_x^\star \right\|_{(I-W)^\dagger}^2 + \frac{2s^2}{\gamma_y} \left( 1 - \frac{\gamma_y}{2} \lambda_{m-1}(I-W) \right) \left\| D_t^y - D_y^\star \right\|_{(I-W)^\dagger}^2$$

$$+ \sqrt{\delta}(1 - \alpha_x) \left\| H_t^x - H_x^\star \right\|^2 + \sqrt{\delta}(1 - \alpha_y) \left\| H_t^y - H_y^\star \right\|^2. \qquad (265)$$

By taking total expectation on both sides and using the definition of $b_x$ and $b_y$, we obtain

$$E\left[\Phi_{t+1}\right] + \tilde{c}_x E\left\|\tilde{x}_{t+1} - \mathbf{1}x^\star\right\|^2 + \tilde{c}_y E\left\|\tilde{y}_{t+1} - \mathbf{1}y^\star\right\|^2$$

$$\leq (1 - b_x) E\left\|x_t - \mathbf{1}x^\star\right\|^2 + (1 - b_y) E\left\|y_t - \mathbf{1}y^\star\right\|^2$$

$$+ \tilde{c}_x \left(1 - \frac{p}{2}\right) E\left\|\tilde{x}_t - \mathbf{1}x^\star\right\|^2 + \tilde{c}_y \left(1 - \frac{p}{2}\right) E\left\|\tilde{y}_t - \mathbf{1}y^\star\right\|^2$$

$$+ \frac{2s^2}{\gamma_x}\left(1 - \frac{\gamma_x}{2}\lambda_{m-1}(I - W)\right) E\left\|D_t^x - D_x^\star\right\|^2_{(I-W)^\dagger} + \frac{2s^2}{\gamma_y}\left(1 - \frac{\gamma_y}{2}\lambda_{m-1}(I - W)\right) E\left\|D_t^y - D_y^\star\right\|^2_{(I-W)^\dagger}$$

$$+ \sqrt{\delta}(1 - \alpha_x)E\left\|H_t^x - H_x^\star\right\|^2 + \sqrt{\delta}(1 - \alpha_y)E\left\|H_t^y - H_y^\star\right\|^2$$

$$= \left(\frac{1 - b_x}{M_x}\right) M_x E\left\|x_t - \mathbf{1}x^\star\right\|^2 + \left(\frac{1 - b_y}{M_y}\right) M_y E\left\|y_t - \mathbf{1}y^\star\right\|^2$$

$$+ \tilde{c}_x \left(1 - \frac{p}{2}\right) E\left\|\tilde{x}_t - \mathbf{1}x^\star\right\|^2 + \tilde{c}_y \left(1 - \frac{p}{2}\right) E\left\|\tilde{y}_t - \mathbf{1}y^\star\right\|^2$$

$$+ \frac{2s^2}{\gamma_x}\left(1 - \frac{\gamma_x}{2}\lambda_{m-1}(I - W)\right) E\left\|D_t^x - D_x^\star\right\|^2_{(I-W)^\dagger} + \frac{2s^2}{\gamma_y}\left(1 - \frac{\gamma_y}{2}\lambda_{m-1}(I - W)\right) E\left\|D_t^y - D_y^\star\right\|^2_{(I-W)^\dagger}$$

$$+ \sqrt{\delta}(1 - \alpha_x)E\left\|H_t^x - H_x^\star\right\|^2 + \sqrt{\delta}(1 - \alpha_y)E\left\|H_t^y - H_y^\star\right\|^2 \tag{266}$$

$$\leq \max\left\{\frac{1 - b_x}{M_x}, \frac{1 - b_y}{M_y}, 1 - \frac{\gamma_x}{2}\lambda_{m-1}(I - W), 1 - \frac{\gamma_y}{2}\lambda_{m-1}(I - W), 1 - \alpha_x, 1 - \alpha_y, 1 - \frac{p}{2}\right\}$$

$$\times \left(E\left[\Phi_t\right] + \tilde{c}_x E\left\|\tilde{x}_t - \mathbf{1}x^\star\right\|^2 + \tilde{c}_y E\left\|\tilde{y}_t - \mathbf{1}y^\star\right\|^2\right)$$

$$= \rho\left(E\left[\Phi_t\right] + \tilde{c}_x E\left\|\tilde{x}_t - \mathbf{1}x^\star\right\|^2 + \tilde{c}_y E\left\|\tilde{y}_t - \mathbf{1}y^\star\right\|^2\right), \tag{267}$$

where

$$\rho = \max\left\{\frac{1 - b_x}{M_x}, \frac{1 - b_y}{M_y}, 1 - \frac{\gamma_x}{2}\lambda_{m-1}(I - W), 1 - \frac{\gamma_y}{2}\lambda_{m-1}(I - W), 1 - \alpha_x, 1 - \alpha_y, 1 - \frac{p}{2}\right\}. \tag{268}$$

By the definition of $\tilde{\Phi}_t = \Phi_t + \tilde{c}_x \left\|\tilde{x}_t - \mathbf{1}x^\star\right\|^2 + \tilde{c}_y \left\|\tilde{y}_t - \mathbf{1}y^\star\right\|^2$, (267) reduces to

$$E\left[\tilde{\Phi}_{t+1}\right] \leq \rho E\left[\tilde{\Phi}_t\right]. \tag{269}$$

Therefore, $E\left[\tilde{\Phi}_{t+1}\right] \leq \rho^{t+1}\tilde{\Phi}_0$.

## H   Proof of Theorem 5.1

This proof is based on several intermediate results proved in Appendices C-E. Hence it would be useful to refer to those results in order to appreciate the proof of Theorem 5.1.

Observe that

$$E\left\|x_{t+1} - \mathbf{1}x^\star\right\|^2 + E\left\|y_{t+1} - \mathbf{1}y^\star\right\|^2 \leq \frac{1}{\min\{M_x, M_y\}}\left(M_x E\left\|x_{t+1} - \mathbf{1}x^\star\right\|^2 + M_y E\left\|y_{t+1} - \mathbf{1}y^\star\right\|^2\right)$$

$$\leq \frac{1}{\min\{M_x, M_y\}}E\left[\tilde{\Phi}_{t+1}\right] \tag{270}$$

$$\leq \frac{1}{M}\rho^{t+1}\tilde{\Phi}_0, \tag{271}$$

where last inequality follows from Lemma G.8 and $M := \min\{M_x, M_y\}$. Hence,

$$E\left\|x_{T(\epsilon)} - \mathbf{1}x^\star\right\|^2 + E\left\|y_{T(\epsilon)} - \mathbf{1}y^\star\right\|^2 \leq \epsilon, \tag{272}$$

for $T(\epsilon) = \frac{1}{-\log\rho}\log\left(\frac{\tilde{\Phi}_0}{M\epsilon}\right)$.

### H.0.1   Gradient Computation Complexity

Recall

$$\rho = \max\left\{\frac{1 - b_x}{M_x}, \frac{1 - b_y}{M_y}, 1 - \frac{\gamma_x}{2}\lambda_{m-1}(I - W), 1 - \frac{\gamma_y}{2}\lambda_{m-1}(I - W), 1 - \alpha_x, 1 - \alpha_y, 1 - \frac{p}{2}\right\}. \tag{273}$$

Using Lemma G.7, $\rho$ can be upper bounded as

$$\rho \leq \max \left\{ 1 - \frac{3b_x}{7}, 1 - \frac{3b_y}{7}, 1 - \frac{b_x}{8\sqrt{\delta}(1+\delta)\kappa_g}, 1 - \frac{1}{8(1+\delta)\kappa_g}, 1 - \frac{b_y}{8\sqrt{\delta}(1+\delta)\kappa_g}, \right. \tag{274}$$

$$\left. 1 - \frac{1}{8(1+\delta)\kappa_g}, 1 - \frac{b_x}{1+\delta}, 1 - \frac{b_y}{1+\delta}, 1 - \frac{p}{2} \right\}. \tag{275}$$

Using (231) and (232), we have

$$1 - \frac{3b_x}{7} \leq 1 - \frac{3}{7}\frac{np_{\min}}{144\kappa_f^2} = 1 - \frac{np_{\min}}{336\kappa_f^2}, \quad 1 - \frac{3b_y}{7} \leq 1 - \frac{3}{7}\frac{np_{\min}}{144\kappa_f^2} = 1 - \frac{np_{\min}}{336\kappa_f^2}$$

$$1 - \frac{b_x}{1+\delta} \leq 1 - \frac{np_{\min}}{144(1+\delta)\kappa_f^2}, \quad 1 - \frac{b_y}{1+\delta} \leq 1 - \frac{np_{\min}}{144(1+\delta)\kappa_f^2} \tag{276}$$

$$1 - \frac{b_x}{8\sqrt{\delta}(1+\delta)\kappa_g} \leq 1 - \frac{np_{\min}}{1152\sqrt{\delta}(1+\delta)\kappa_g\kappa_f^2}, \quad 1 - \frac{b_y}{8\sqrt{\delta}(1+\delta)\kappa_g} \leq 1 - \frac{np_{\min}}{1152\sqrt{\delta}(1+\delta)\kappa_g\kappa_f^2}. \tag{277}$$

Therefore,

$$\rho \leq \max \left\{ 1 - \frac{np_{\min}}{336\kappa_f^2}, 1 - \frac{np_{\min}}{1152\sqrt{\delta}(1+\delta)\kappa_g\kappa_f^2}, 1 - \frac{1}{8(1+\delta)\kappa_g}, 1 - \frac{np_{\min}}{144(1+\delta)\kappa_f^2}, 1 - \frac{p}{2} \right\} \tag{278}$$

$$= 1 - \min \left\{ \frac{np_{\min}}{336\kappa_f^2}, \frac{np_{\min}}{1152\sqrt{\delta}(1+\delta)\kappa_g\kappa_f^2}, \frac{1}{8(1+\delta)\kappa_g}, \frac{np_{\min}}{144(1+\delta)\kappa_f^2}, \frac{p}{2} \right\} \tag{279}$$

$$=: 1 - \tilde{C}. \tag{280}$$

By taking log on both sides, we obtain

$$\log \rho \leq \log(1 - \tilde{C})$$

$$-\log \rho \geq -\log(1 - \tilde{C})$$

$$\frac{1}{-\log \rho} \leq \frac{1}{-\log(1 - \tilde{C})}$$

$$\leq \frac{5}{\tilde{C}}$$

$$= 5 \left( \min \left\{ \frac{np_{\min}}{336\kappa_f^2}, \frac{np_{\min}}{1152\sqrt{\delta}(1+\delta)\kappa_g\kappa_f^2}, \frac{1}{8(1+\delta)\kappa_g}, \frac{np_{\min}}{144(1+\delta)\kappa_f^2}, \frac{p}{2} \right\} \right)^{-1}$$

$$= 5 \max \left\{ \frac{336\kappa_f^2}{np_{\min}}, \frac{1152\sqrt{\delta}(1+\delta)\kappa_g\kappa_f^2}{np_{\min}}, 8(1+\delta)\kappa_g, \frac{144(1+\delta)\kappa_f^2}{np_{\min}}, \frac{2}{p} \right\}, \tag{281}$$

where the fourth inequality uses the fact that $(1/-\log(1 - x)) \leq 5/x$ for all $0 < x < 1$. Using Lemma G.7, we have $M_x \geq 1 - \frac{4b_x}{7}$. Therefore, $M_x > 1 - \frac{4}{7} = \frac{3}{7}$ because $0 < b_x < 1$. Moreover, $M_y > \frac{3}{7}$ as $0 < b_y < 1$. Therefore, $\log\left(\frac{\tilde{\Phi}_0}{M\epsilon}\right) \leq \log\left(\frac{7\tilde{\Phi}_0}{3\epsilon}\right)$. Hence,

$$T(\epsilon) = \mathcal{O}\left( \max \left\{ \frac{\kappa_f^2}{np_{\min}}, \frac{\sqrt{\delta}(1+\delta)\kappa_g\kappa_f^2}{np_{\min}}, (1+\delta)\kappa_g, \frac{(1+\delta)\kappa_f^2}{np_{\min}}, \frac{2}{p} \right\} \log\left(\frac{\tilde{\Phi}_0}{\epsilon}\right) \right). \tag{282}$$

## H.1 Discussion on the analysis techniques

In this section, we discuss and compare the analysis techniques of our work with those in existing works. In [20] a convex composite minimization problem is studied and inexact PDHG method is applied to its saddle point formulation. In this work, we study a different problem (1) where a smooth function depends jointly on primal and dual variables. We prove that it is equivalent to study unconstrained saddle point problem (3) to get the solution of (1). However, [20] uses a well known equivalence between a convex minimization problem and its Lagrangian formulation [19]. We define additional quantities $D_y^\star$, $H_y^\star$ and Bregman distance functions $V_{f_i,y}(x_1, x_2)$, $V_{-f_i,x}(y_1, y_2)$ in Appendix C to obtain appropriate bounds.

**Algorithm 2 analysis:** In contrast to [20], we get complicated upper bounds depending on primal and dual iterates in Lemma E.3 which yields different set of parameters. We prove the feasibility of these parameters and derive useful lower and upper bounds in Appendix E.4. [20] uses diminishing step size and an induction approach to prove the convergence to exact solution. However, we use a different method summarized below to derive the convergence rate of Algorithm (2). We first derive a relation which connects iterate $t$ information with the restart iterates (see Lemma E.6). Then by appropriately choosing the restart iterates $x_{k,0}, y_{k,0}, D^x_{k,0}, D^y_{k,0}, H^x_{k,0}$ and $H^y_{k,0}$ and inner iterates $t_k$, we get the complexity result in Appendix F. It is worth noting that proof techniques of Lemma E.6 and Theorem 4.1 are different from those in [20] due to different algorithm structure involving a restart scheme, complicated bounds, and different sets of parameters.

**Algorithm 3 analysis:** Using smoothness, strong convexity strong concavity assumptions, and definitions of $V_{f_i,y}(x_1, x_2)$ and $V_{-f_i,x}(y_1, y_2)$, we upper bound $E \|x_t - \mathbf{1}x^\star - s\mathcal{G}^x_t + s\nabla_x F(\mathbf{1}x^\star, \mathbf{1}y^\star)\|^2 + E \|y_t - \mathbf{1}y^\star + s\mathcal{G}^y_t - s\nabla_y F(\mathbf{1}x^\star, \mathbf{1}y^\star)\|^2$ in terms of $x_t, y_t, \tilde{x}_t, \tilde{y}_t, V_{f_i,y}(x_1, x_2)$ and $V_{-f_i,x}(y_1, y_2)$ in Lemma G.5. Note that the upper bound in Lemma G.5 is complicated and different from that of [20] because we have additional terms contributed by dual variable $y$ with different coefficients and terms containing square norms dependent on the reference points. This intermediate result generates different bounds and sets of parameters in the subsequent analysis. We carefully set the step size and choose algorithm parameters with proven feasibility in Lemma G.7. We rigorously compute lower and upper bounds on chosen parameters in terms of $\kappa_f, \kappa_g$ and $\delta$ in Lemma G.7 and Appendix H. In our work, these derivations are more involved in comparison to [20].

Analysis methods of [36] and [23] are based on averaging quantities; for example average of iterates and gradients. The analysis in [36] and [23] requires separate bounds for consensus error and gradient estimation errors and depends in addition on the smoothness of saddle point problem. In contrast to [36] and [23], our analysis does not demand any separate bound on consensus error and gradient estimation error and handles non-smooth functions as well. Unlike our compression based communication scheme, the analysis in [5] bounds errors using an accelerated gossip scheme and approximate solution obtained by solving an inner saddle point problem at every node.

# I    Numerical Experiments

We evaluate the effectiveness of proposed algorithms on robust logistic regression problem

$$\min_{x \in \mathcal{X}} \max_{y \in \mathcal{Y}} \Psi(x, y) = \frac{1}{N} \sum_{i=1}^{N} \log\left(1 + exp\left(-b_i x^\top(a_i + y)\right)\right) + \frac{\lambda}{2}\|x\|_2^2 - \frac{\beta}{2}\|y\|_2^2 \qquad (283)$$

over a binary classification data set $\mathcal{D} = \{(a_i, b_i)\}_{i=1}^N$. We consider constraint sets $\mathcal{X}$ and $\mathcal{Y}$ as $\ell_2$ ball of radius 100 and 1 respectively. We compute smoothness parameters $L_{xx}, L_{yy}, L_{xy}$ and $L_{yx}$ using Hessian information of the objective function (see Appendix I.5) and set strong convexity and strong concavity parameters to $\lambda$ and $\beta$ respectively. Unless stated otherwise, we set $\lambda = \beta = 10$, number of nodes to $m = 20$ and number of batches to $n = 20$ in all our experiments. The initial points $x_0, y_0$ are generated randomly and $D_x, D_y$ are set to 0. We set up the step size of proposed methods and baseline methods using the theoretically values provided in the respective papers. We implement all the experiments in Python Programming Language.

## I.1   Experimental Setup:

**Datasets:** We rely on four binary classification datasets namely, a4a, phishing and ijcnn1 from `https://www.csie.ntu.edu.tw/~cjlin/libsvmtools/datasets/` and sido data from `http://www.causality.inf.ethz.ch/data/SIDO.html`. The characteristics of these datasets are reported in Table 2. We distribute the samples across 20 nodes and create 20 mini batches of local samples for all datasets.

**Network Setting:** We conduct the experiments for 2D torus topology and ring topology. We generate weight matrix $W$ with $W_{ij} = 1/5$ and $W_{ij} = 1/3$ for all $(i, j) \in \mathcal{E} \cup \{(i, i)\}$ for 2D torus topology and ring topology respectively.

**Compression Operator:** We consider an unbiased $b$-bits quantization operator $Q_\infty(x)$ [24] throughout our empirical study.

$$Q_\infty(x) = \left(\|x\|_\infty 2^{-(b-1)} sign(x)\right) \cdot \left\lfloor \frac{2^{b-1}|x|}{\|x\|_\infty} + u \right\rfloor, \qquad (284)$$

Table 2: Data Sets used for experiments. $N$ and $d$ denote respectively the number of samples and number of features.

| Data set | $N$ | $d$ |
|---|---|---|
| a4a | 4781 | 122 |
| phishing | 11,055 | 68 |
| ijcnn1 | 49,990 | 22 |
| sido | 2536 | 4932 |

where $\cdot$ represents Hadamard product, $|x|$ denotes elementwise absolute value and $u$ is a random vector uniformly distributed in $[0,1]^d$. Theorem 3 in [24] shows that $Q_\infty(x)$ satisfies assumption 3.4 with $\delta = \sup_x \frac{\left\|sign(x)2^{-(b-1)}\right\|^2 \|x\|_\infty^2}{4\|x\|_2^2}$. We know that $\|x\|_\infty \le \|x\|_2$ for all $x \in \mathbb{R}^d$. Using this inequality, we can upper bound $\delta$ as follows:

$$\delta \le \sup_x \frac{\left\|sign(x)2^{-(b-1)}\right\|^2 \|x\|_2^2}{4\|x\|_2^2} = \sup_x \frac{\left\|sign(x)2^{-(b-1)}\right\|^2}{4} \le \frac{d}{4(2^{b-1})^2}. \tag{285}$$

The above bound depends can be made independent of $d$ by choosing $b = 1 + \log_2 \sqrt{d}$. We use six different bits value from the set $\{1 + \log_2 \sqrt{d}, 2, 4, 8, 16, 32\}$ to evaluate the behavior of Algorithm 2 and Algorithm 3 with number of bits.

### I.2 Baseline methods

We compare the performance of proposed algorithms C-RDPSG and C-DPSVRG with three non-compression based baseline algorithms: (1) Distributed Min-Max Data similarity algorithm [5] (2) Decentralized Parallel Optimistic Stochastic Gradient (DPOSG) algorithm [23] and, (3) Decentralized Minimax Hybrid Stochastic Gradient Descent (DM-HSGD) algorithm [36].

**Distributed Min-Max data similarity:** This algorithm is based on accelerated gossip scheme employed on model updates and gradient vectors [5]. The number of iterates in accelerated gossip scheme and the step size are computed according to the theoretical details provided in [5]. This algorithm requires approximate solution of an inner saddle point problem at every iterate. We run extragradient method [16] to solve the inner saddle point problem with a desired precision accuracy provided in [5]. Throughout this section, we use the shorthand notation for Distributed Min-Max data similarity algorithm as Min-Max similarity.

**Decentralized Parallel Optimistic Stochastic Gradient (DPOSG):** DPOSG [23] is a two step algorithm with local model averaging designed for solving unconstrained saddle-point problems in a decentralized fashion. We include the projection steps to both update sequences of DPOSG as we are solving constrained problem (283). The step size and the number of local model averaging steps are tuned according to Theorem 1 in [23].

**Decentralized Minimax Hybrid Stochastic Gradient Descent (DM-HSGD):** DM-HSGD [36] is a gradient tracking based algorithm designed for solving saddle point problems with a constraint set on dual variable. We incorporate projection step to the model update of primal variable. We use grid search to find the best step sizes for primal and dual variable updates. Other parameters like initial large batch size and parameters involved in gradient tracking update sequence are chosen according to the experimental setting in [36].

### I.3 Benchmark Quantities

We run the centralized and uncompressed version of C-DPSVRG for $50,000$ iterations to find saddle point solution $z^\star = (x^\star, y^\star)$ of problem (283). The performance of all the methods is measured using $\sum_{i=1}^m \left\|z_t^i - z^\star\right\|^2$.

**Number of gradient computations and communications:** We calculate the total number of gradient computations according to the number of samples used in the gradient computation at a given iterate $t$. The number of communications per iterate are computed as the number of times a node exchanges information with its neighbors.

**Number of bits transmitted:** We set number of bits $b = 4$ in compression operator $Q_\infty(x)$ for C-RDPSG and C-DPSVRG. Similar to [15], we assume that on an average 5 bits (1 bit for sign and 4

bits for quantization level) are transmitted at every iterate for C-RDPSG and C-DPSVRG. We assume that on an average 32 bits are transmitted per communication for DPOSG, DM-HSGD and Min-Max similarity algorithm.

## I.4 Observations

**Comparison to baselines:** C-RDPSG converges faster in the beginning and slows down after reaching approximately $10^{-8}$ accuracy as depicted in Figure 1 and Figure 2. C-DPSVRG converges faster than other baseline methods. DPOSG and DM-HSGD converge only to a neighbourhood of the saddle point solution and start oscillating after some time. The restart scheme's inclusion in C-RDPSG helps mitigate the flat and oscillatory behaviour at the later iterations, unlike DPOSG and DM-HSGD. C-RDPSG is faster than C-DPSVRG, DPOSG, DM-HSGD and Min-Max similarity in terms of gradient computations, communications and bits transmitted to achieve saddle points of moderate accuracy. The performance of C-RDPSG is competitive with DPOSG and DM-HSGD in the long run as demonstrated in Figure 1 and Figure 2.

**Compression effect:** Plots in Figure 1 depict that C-DPSVRG is 1000 times faster than Min-Max similarity, DPOSG algorithm and 10 times faster than DM-HSGD in terms of transmitted bits. We observe that C-RDPSG is also 1000 times faster than Min-max similarity and DPOSG for obtaining saddle point solutions of moderate accuracy, in terms of transmitted bits.

**Communication efficiency:** The one-time communication at every iterate in C-DPSVRG speeds up communication and makes C-DPSVRG to be 100 times faster than Min-Max similarity and DPOSG methods as shown in Figure 1. C-RDPSG is 10 times faster than DM-HSGD and 100 times faster than DPOSG and Min-Max similarity in terms of communications at the initial stages of the algorithm.

**Different choices of reference probabilities:** The full batch gradient computations in C-DPSVRG depends on the reference probability parameter $p$. Motivated from [18], we run C-DPSVRG with five different reference probabilities as $1/n, 1/(\kappa n^3)^{1/4}, 1/(\kappa n)^{1/2}, 1/(\kappa^3 n)^{1/4}$ and $1/\kappa$. From Figure 3, we observe that setting $p = 1/n$ requires the least number of gradient computations as it corresponds to the less frequent computation of full batch gradients.

**Impact of topology:** From Figure 2, we observe that the convergence behaviour of all methods is similar to 2D torus topology. We also note that ring topology requires large number of communications compared to 2D torus due to its sparse connectivity.

**Compression error:** We plot compression error $\|Q(\nu^x) - \nu^x\|^2 + \|Q(\nu^y) - \nu^y\|^2$ against number of transmitted bits for C-RDPSG and C-DPSVRG as shown in Figure 4. We observe that C-DPSVRG with $O(\log d)$ bits achieves compression error $10^{-25}$ in less than 20,000 transmitted bits. It shows a clear advantage of using $O(\log d)$ bits in C-DPSVRG while maintaining low compression error. In C-RDPSG, larger the number of bits used in the quantization operator, smaller the compression error. There are sharp jumps in the decay of compression error during a restart of C-RDPSG as depicted in Figure 4.

**Number of bits transmitted:** As demonstrated in Figure 5, C-DPSVRG transmits less number of bits when $b = 1 + \log_2 \sqrt{d}$ to achieve highly accurate solution. We can observe that the convergence behavior of C-DPSVRG is affected by number of bits less than $1 + \log_2 \sqrt{d}$. For example, the convergence of C-DPSVRG becomes slow for sido data with $b = 2, 4 < 1 + \log_2 \sqrt{d} \approx 7$ as shown in Figure 5. It shows that $Q_\infty(x)$ provides better performance for $b = \mathcal{O}(\log_2 d)$ especially for high-dimensional data points. The behavior of C-RDPSG is less affected by varying the number of bits as shown in Figure 6. In the long term, C-RDPSG behavior is almost identical for all chosen values of number of bits except $b = 2$.

**Impact of number of nodes:** As the number of nodes increases, C-RDPSG requires fewer gradient computations in the initial phase. However, smaller number of nodes gives faster convergence at the later iterations for 2D torus and ring topology, as depicted in Figure 8 and Figure 10. C-DPSVRG also requires small number of gradient computations in 2D torus with larger number of nodes because it assigns smaller batch sizes to every node. As depicted in Figure 7, C-DPSVRG performance does not change much in terms of the number of communications and bits transmitted for 2D torus. The sparsity level of ring topology is higher than that of 2D torus and increases with number of nodes. It leads C-DPSVRG to achieve fast convergence eventually in terms of gradient computations with a smaller number of nodes, as demonstrated in Figure 9. In contrast to the performance of C-DPSVRG in terms of communications in 2D torus (Figure 7), C-DPSVRG requires more communications for large number of nodes in a ring topology, as shown in Figure 9. DM-HSGD and C-RDPSG are competitive in terms of gradient computations and communications with larger number of nodes as demonstrated in Figures 11 and 12. However it is to be noted that DM-HSGD exhibits an oscillatory behavior in the saddle point solutions, after reaching a moderate solution accuracy.

## I.5 Estimating Lipschitz parameters

In this section, we estimate Lipschitz parameters $L_{xx}, L_{yy}, L_{xy}, L_{yy}$ of robust logistic regression problem (283). Assume that each node $i$ has $N_i$ number of local samples such that $\sum_{i=1}^{m} N_i = N$. Recall objective function $\Psi(x, y)$ in equation (283):

$$
\begin{aligned}
\Psi(x, y) &= \frac{1}{N} \sum_{i=1}^{N} \log\left(1 + exp\left(-b_i x^\top (a_i + y)\right)\right) + \frac{\lambda}{2} \|x\|_2^2 - \frac{\beta}{2} \|y\|_2^2 \\
&= \frac{1}{N} \sum_{i=1}^{m} \sum_{l=1}^{N_i} \log\left(1 + exp\left(-b_{il} x^\top (a_{il} + y)\right)\right) + \frac{\lambda}{2} \|x\|_2^2 - \frac{\beta}{2} \|y\|_2^2 \\
&= \sum_{i=1}^{m} \left( \frac{1}{N} \sum_{l=1}^{N_i} \log\left(1 + exp\left(-b_{il} x^\top (a_{il} + y)\right)\right) + \frac{\lambda}{2m} \|x\|_2^2 - \frac{\beta}{2m} \|y\|_2^2 \right) \\
&= \sum_{i=1}^{m} f_i(x, y),
\end{aligned}
$$

where $f_i(x, y) = \frac{1}{N} \sum_{l=1}^{N_i} \log\left(1 + exp\left(-b_{il} x^\top (a_{il} + y)\right)\right) + \frac{\lambda}{2m} \|x\|_2^2 - \frac{\beta}{2m} \|y\|_2^2$. Gradients of $f_i(x, y)$ with respect to $x$ and $y$ are given by

$$
\nabla_x f_i(x, y) = \frac{1}{N} \sum_{l=1}^{N_i} \frac{-b_{il}(a_{il} + y)}{1 + exp\left(b_{il} x^\top (a_{il} + y)\right)} + \frac{\lambda}{m} x
$$

$$
\nabla_y f_i(x, y) = \frac{1}{N} \sum_{l=1}^{N_i} \frac{-b_{il} x}{1 + exp\left(b_{il} x^\top (a_{il} + y)\right)} - \frac{\beta}{m} y.
$$

We create $n$ batches $\{N_{i1}, \ldots, N_{in}\}$ of local samples $N_i$ and write $f_i(x, y)$ in the form of $\frac{1}{n} f_{ij}(x, y)$.

$$
\begin{aligned}
f_i(x, y) &= \frac{1}{N} \sum_{l=1}^{N_i} \log\left(1 + exp\left(-b_{il} x^\top (a_{il} + y)\right)\right) + \frac{\lambda}{2m} \|x\|_2^2 - \frac{\beta}{2m} \|y\|_2^2 \\
&= \frac{1}{N} \sum_{j=1}^{n} \sum_{l=1}^{N_{ij}} \log\left(1 + exp\left(-b_{il}^j x^\top (a_{il}^j + y)\right)\right) + \frac{\lambda}{2m} \|x\|_2^2 - \frac{\beta}{2m} \|y\|_2^2 \\
&= \sum_{j=1}^{n} \left( \frac{1}{N} \sum_{l=1}^{N_{ij}} \log\left(1 + exp\left(-b_{il}^j x^\top (a_{il}^j + y)\right)\right) + \frac{\lambda}{2mn} \|x\|_2^2 - \frac{\beta}{2mn} \|y\|_2^2 \right) \\
&= \frac{1}{n} \sum_{j=1}^{n} \left( \frac{n}{N} \sum_{l=1}^{N_{ij}} \log\left(1 + exp\left(-b_{il}^j x^\top (a_{il}^j + y)\right)\right) + \frac{\lambda}{2m} \|x\|_2^2 - \frac{\beta}{2m} \|y\|_2^2 \right) \\
&= \frac{1}{n} \sum_{j=1}^{n} f_{ij}(x, y),
\end{aligned}
$$

where $f_{ij}(x, y) = \frac{n}{N} \sum_{l=1}^{N_{ij}} \log\left(1 + exp\left(-b_{il}^j x^\top (a_{il}^j + y)\right)\right) + \frac{\lambda}{2m} \|x\|_2^2 - \frac{\beta}{2m} \|y\|_2^2$. We are now ready to find required Lipschitz parameters.

**Computing $L_{xx}^{ij}$:**

$$
\nabla_{xx}^2 f_{ij}(x, y) = \frac{n}{N} \sum_{l=1}^{N_{ij}} \frac{(a_{il}^j + y)(a_{il}^j + y)^\top exp\left(b_{il}^j x^\top (a_{il}^j + y)\right)}{(1 + exp(b_{il}^j x^\top (a_{il}^j + y))^2} + \frac{\lambda}{m} I
$$

$$
\implies \left\| \nabla_{xx}^2 f_{ij}(x, y) \right\|_2 \leq \frac{n}{4N} \sum_{l=1}^{N_{ij}} (2\|a_{il}^j\|_2^2 + 2R_y^2) + \frac{\lambda}{m}
$$

$$
= \frac{n}{2N} \sum_{l=1}^{N_{ij}} \|a_{il}^j\|_2^2 + \frac{n N_{ij} R_y^2}{2N} + \frac{\lambda}{m} =: L_{xx}^{ij}.
$$

**Computing $L_{yy}^{ij}$:**

$$\nabla_{yy}^2 f_{ij}(x,y) = \frac{n}{N} \sum_{l=1}^{N_{ij}} \frac{exp\left(b_{il}^j x^\top (a_{il}^j + y)\right)(b_{il}^j)^2 xx^\top}{\left(1 + exp\left(b_{il}^j x^\top (a_{il}^j + y)\right)\right)^2} - \frac{\beta}{m}I$$

$$\implies \left\|\nabla_{xx}^2 f_{ij}(x,y)\right\|_2 \le \frac{n}{N} \sum_{l=1}^{N_{ij}} \frac{\left\|xx^\top\right\|_2}{4} + \frac{\beta}{m}$$

$$\le \frac{nN_{ij}R_x^2}{4N} + \frac{\beta}{m} =: L_{yy}^{ij}.$$

**Computing $L_{xy}^{ij}$:**

$$\nabla_y(\nabla_x f_{ij}(x,y)) = \frac{n}{N} \sum_{l=1}^{N_{ij}} \left( \frac{-b_{il}^j I}{1 + exp\left(b_{il}^j x^\top (a_{il}^j + y)\right)} + (b_{il}^j)^2(a_{il}^j + y)x^\top \frac{exp\left(b_{il}^j x^\top (a_{il}^j + y)\right)}{(1 + exp(b_{il}^j x^\top (a_{il}^j + y))^2} \right)$$

Hence we have

$$\left\|\nabla_{xy}^2 f_{ij}(x,y)\right\|_2 \le \frac{n}{N} \sum_{l=1}^{N_{ij}} \left(1 + \frac{1}{4}\left\|(a_{il}^j + y)x^\top\right\|_2\right)$$

$$\le \frac{n}{N} \sum_{l=1}^{N_{ij}} \left(1 + \frac{R_x}{4}\|(a_{il}^j + y)\|_2\right)$$

$$\le \frac{n}{N} \sum_{l=1}^{N_{ij}} \left(1 + \frac{R_x}{4}(\|a_{il}^j\|_2 + R_y)\right)$$

$$= \frac{n}{N} \left(\left(1 + \frac{R_x R_y}{4}\right) N_{ij} + \frac{R_x}{4} \sum_{l=1}^{N_{ij}} \|a_{il}^j\|_2\right) =: L_{xy}^{ij}.$$

We set $L_{xx} = \max_{i,j}\{L_{xx}^{ij}\}$, $L_{yy} = \max_{i,j}\{L_{yy}^{ij}\}$ and $L_{xy} = L_{yx} = \max_{i,j}\{L_{xy}^{ij}\}$. The strong convexity and strong concavity parameters are respectively set to $\lambda$ and $\beta$.

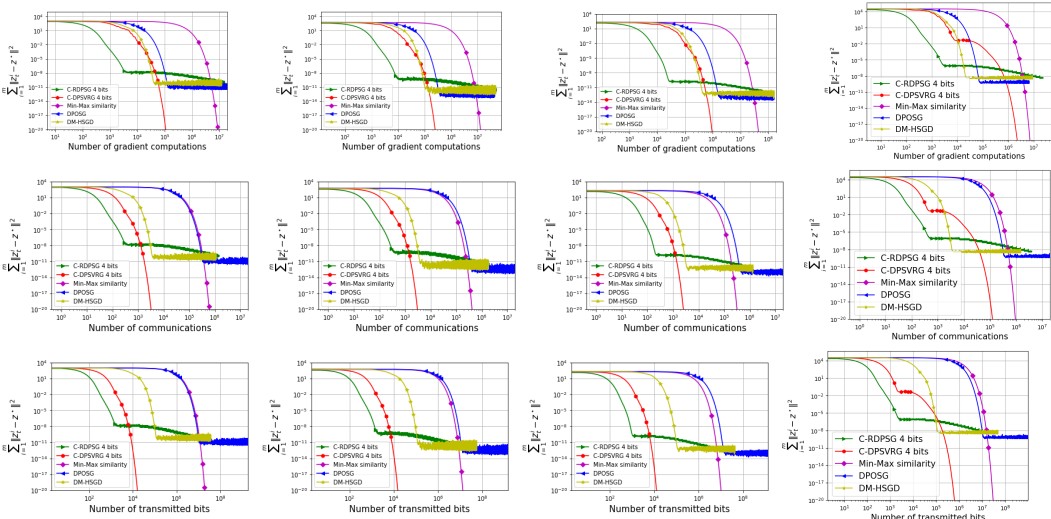

Figure 1: Convergence behavior of iterates to saddle point vs. Gradient computations (Row 1), Communications (Row 2), Number of bits transmitted (Row 3) for different algorithms in 2D torus topology. a4a, phishing, ijcnn, sido datasets are in Columns 1,2,3,4 respectively.

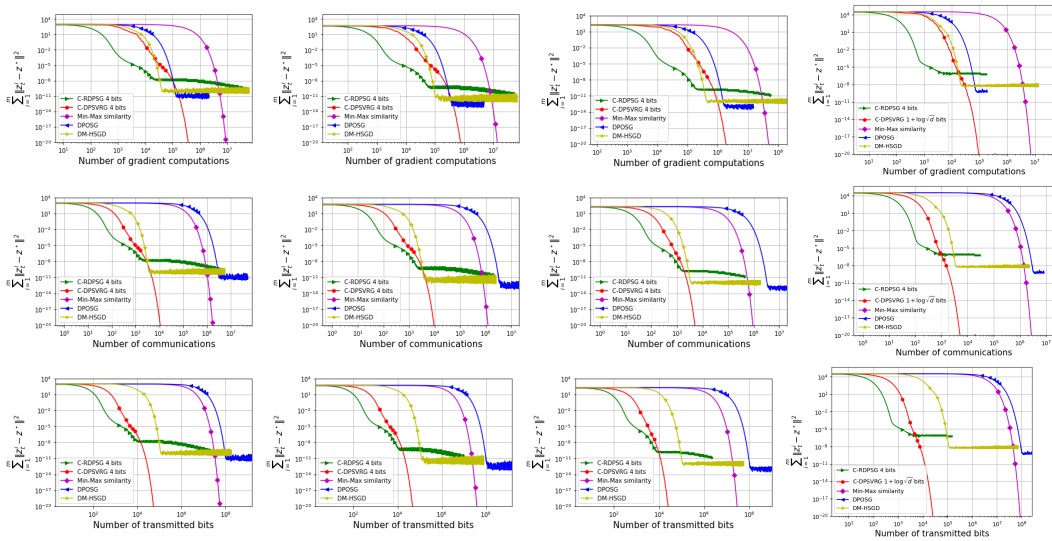

Figure 2: Convergence behavior of iterates to saddle point vs. Gradient computations (Row 1), Communications (Row 2), Number of bits transmitted (Row 3) for different algorithms in ring topology. a4a, phishing, ijcnn, sido datasets are in Columns 1,2,3,4 respectively.

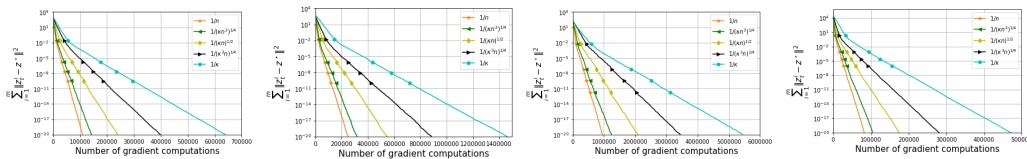

Figure 3: Performance of C-DPSVRG with different reference probabilities in 2D torus. a4a, phishing, ijcnn, sido datasets are in Columns 1,2,3,4 respectively.

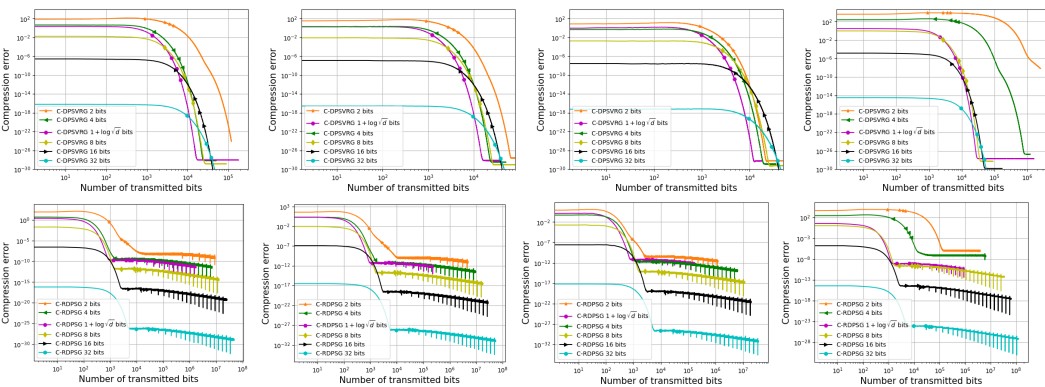

Figure 4: Compression error with different number of bits. Row 1: C-DPSVRG, Row 2: C-RDPSG. a4a, phishing, ijcnn, sido datasets are in Columns 1,2,3,4 respectively.

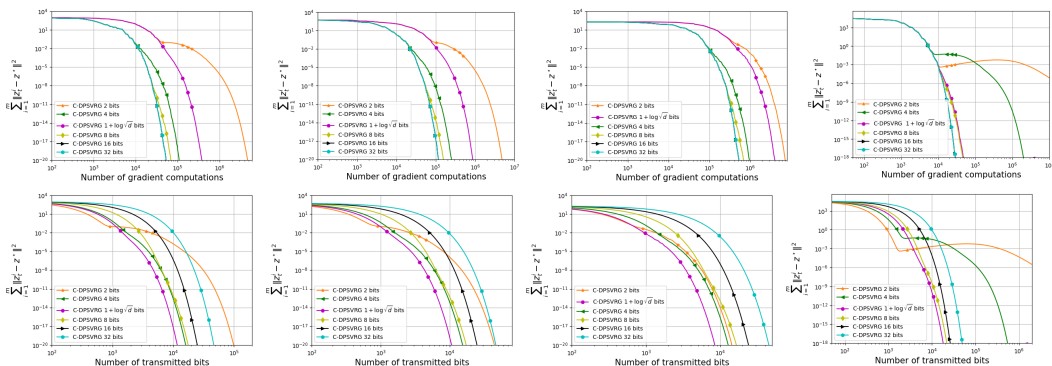

Figure 5: Convergence behavior of iterates to saddle point vs. Gradient computations (Row 1), Number of bits transmitted (Row 2) for C-DPSVRG behavior with **different number of bits** in 2D torus topology. a4a, phishing, ijcnn, sido datasets are in Columns 1,2,3,4 respectively. Number of bits $1 + \log \sqrt{d}$ for a4a, phishing, ijcnn1 and sido datasets are $4.465, 4.043, 3.22$ and $7.13$ respectively.

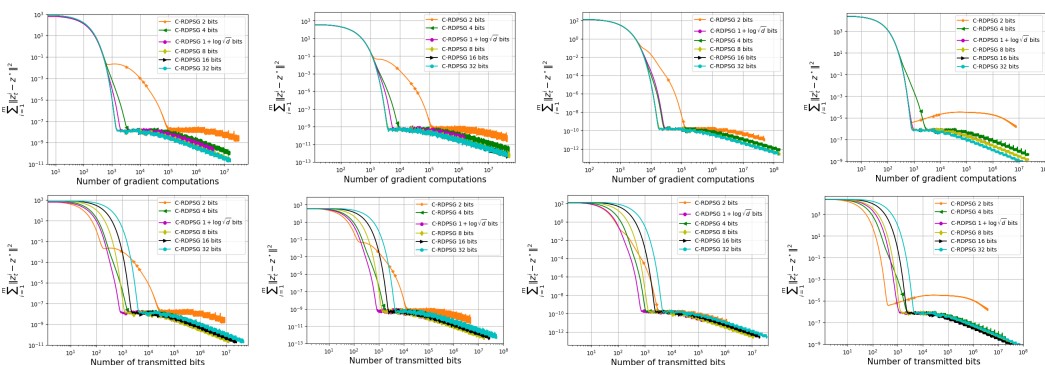

Figure 6: Convergence of iterates to saddle point vs. Gradient computations (Row 1), Number of bits transmitted (Row 2) for C-RDPSG behavior with **different number of bits** in 2D torus topology. a4a, phishing, ijcnn, sido datasets are in Columns 1,2,3,4 respectively. Number of bits $1 + \log \sqrt{d}$ for a4a, phishing, ijcnn1 and sido datasets are $4.465, 4.043, 3.22$ and $7.13$ respectively.

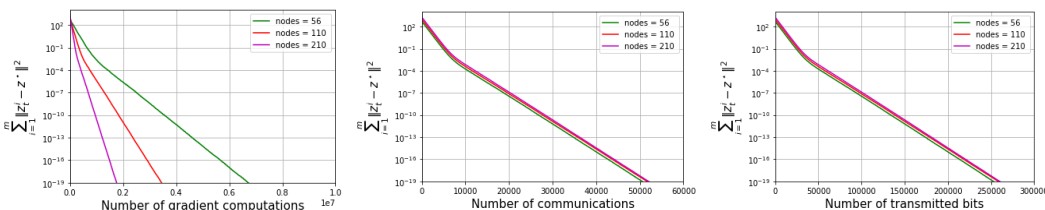

Figure 7: Performance of C-DPSVRG with different number of nodes on 2D torus topology with ijcnn data.

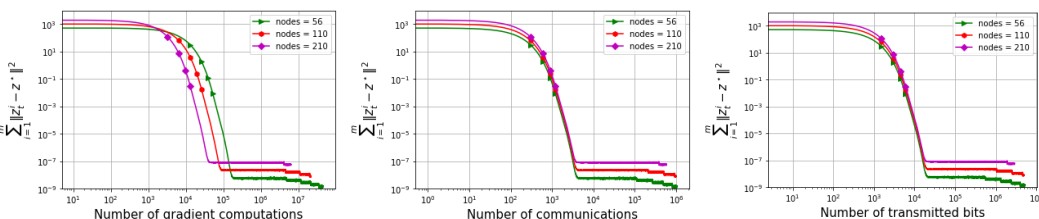

Figure 8: Performance of C-RDPSG with different number of nodes on 2D torus topology with ijcnn data.

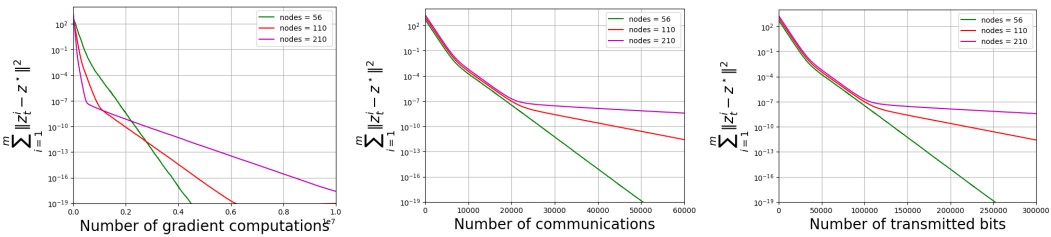

Figure 9: Performance of C-DPSVRG with different number of nodes on ring topology with ijcnn data.

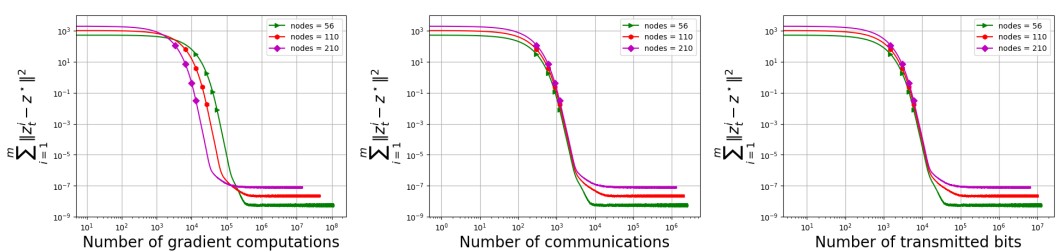

Figure 10: Performance of C-RDPSG with different number of nodes on ring topology with ijcnn data.

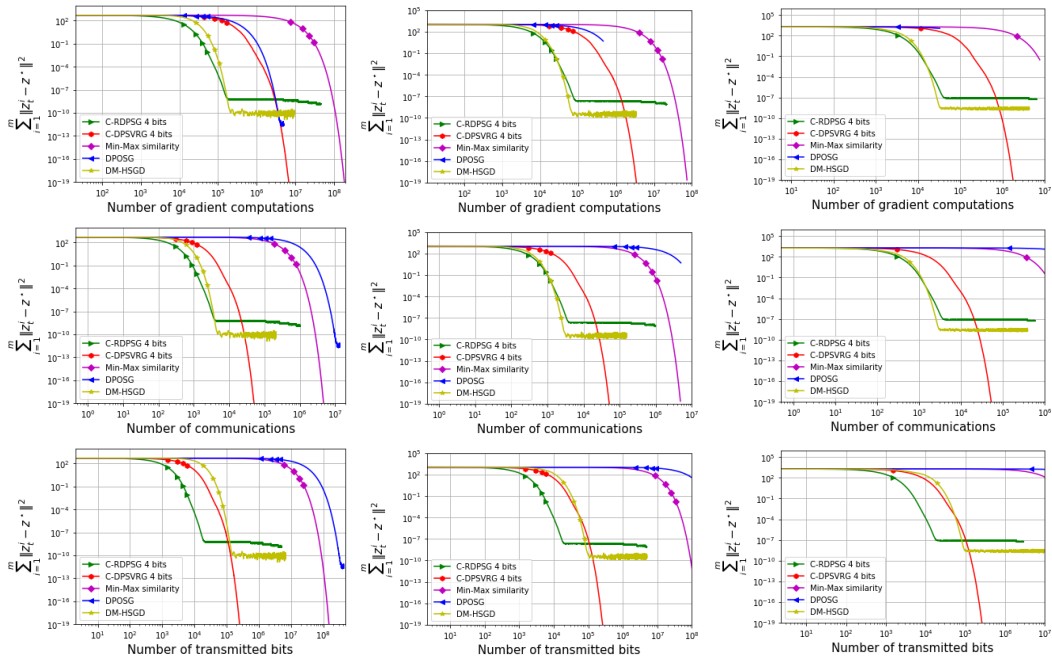

Figure 11: Comparison with baselines with different number of nodes on a 2D torus topology with ijcnn data. Column 1: 56 nodes, Column 2: 110 nodes, Column 3: 210 nodes.

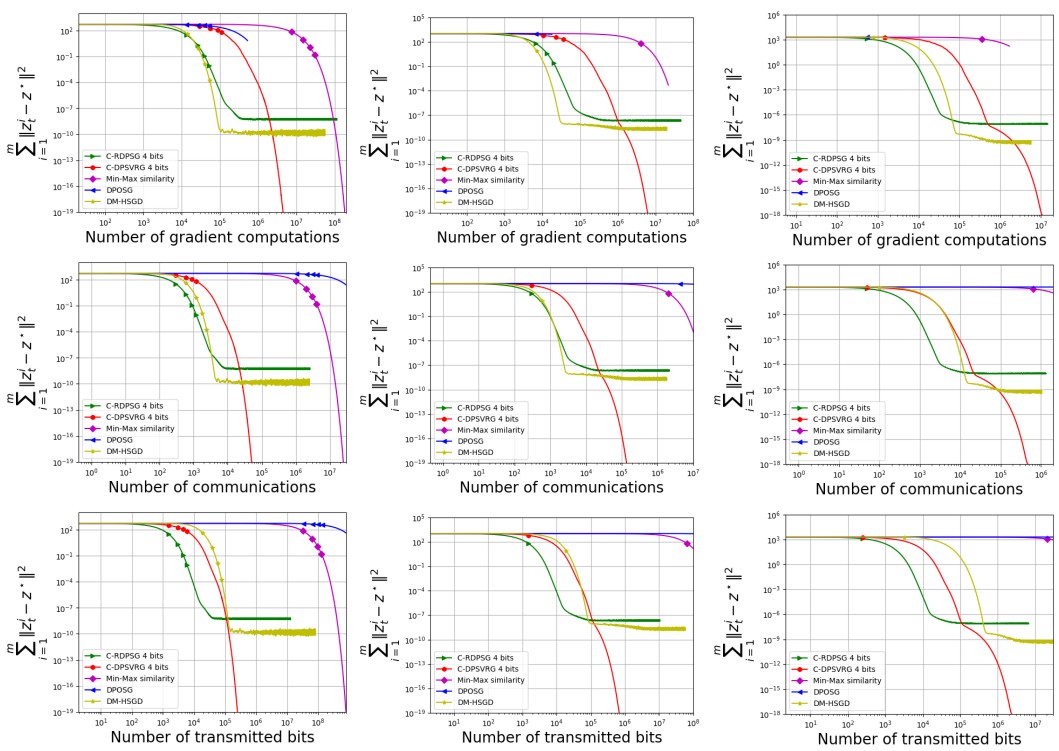

Figure 12: Comparison with baselines with different number of nodes on a ring topology with ijcnn data. Column 1: 56 nodes, Column 2: 110 nodes, Column 3: 210 nodes.

