# OpenReview forum: "Stochastic Gradient Methods with Compressed Communication for Decentralized Saddle Point Problems"
_NeurIPS.cc/2022/Workshop/Federated_Learning — FL-NeurIPS 2022 Poster_

### Official Review · Reviewer_u25Z · 2022-10-05
**Technical contribution on decentralized min-max optimization**

This paper develops two algorithms to solve decentralized non-smooth strongly convex-strongly concave saddle-point problems over networks and with communication compression. The two algorithms address the general stochastic setting and finite-sum optimization (where variance reduction can be applied). A rigorous complexity analysis is provided for both algorithms, as well as experiments are presented (in the appendix) on a logistic regression problem.

I would say that this contribution is rather on the edge of the scope for this workshop (more a core optimization paper) but could be seen as a contribution to "decentralized FL" as announced in the call.

- Although it is a very technical and rigorous work, it is partly difficult for the reader to understand what the main novel contributions are (without checking the mentioned references in detail). For example, several elements have existed before (variance reduction, compression of differences) and it is not clear from the text whether the method is a mix-and-patch of existing tools, or whether new seminal approaches are presented. Maybe this could be highlighted a bit better in the final version.
- Furthermore, I would have enjoyed a brief discussion of the convergence rates (is the dependency on the problem parameters tight)?

---

### Official Review · Reviewer_dXi3 · 2022-10-18
**The paper presents two algorithms for decentralized minimax optimization problems via compression. The authors show extensive numerical results as well as convergence analysis for their proposed methods.**

The authors present two compression-based stochastic gradient algorithms for solving saddle-point problems in a decentralized setting. The first algorithm is a restart-based decentralized proximal stochastic gradient method with compression for general stochastic settings. The second algorithm is a decentralized proximal stochastic variance-reduced gradient algorithm with compression for finite sum settings. The authors show a sublinear convergence rate for the first algorithm and a linear convergence for the second algorithm depending on the condition number of the function, graph, and compression ratio.

Please elaborate on the tightness of the bounds based on the dependence on the compression ratio and graph condition number.

---

### Decision · Program_Chairs · 2022-10-20

Accept (Poster)